# Offline RL with Smooth OOD Generalization in Convex Hull and its Neighborhood

**Qingmao Yao**[1], **Zhichao Lei**[2], **Tianyuan Chen**[3,4,6], **Ziyue Yuan**[1], **Xuefan Chen**[1], **Jianxiang Liu**[3,4], **Faguo Wu**[3,4,5,6,*], **Xiao Zhang**[1,4,5,6,7*]

[1]School of Mathematical Sciences, Beihang University
[2]School of Physics, Beihang University
[3]School of Artificial Intelligence, Beihang University
[4]Key Laboratory of Mathematics, Informatics and Behavioral Semantics, MoE, Beihang University
[5]Beijing Advanced Innovation Center for Future Blockchain and Privacy Computing, Beihang University
[6]Zhongguancun Laboratory
[7]Hangzhou International Innovation Institute of Beihang University

## Abstract

Offline Reinforcement Learning (RL) struggles with distributional shifts, leading to the $Q$-value overestimation for out-of-distribution (OOD) actions. Existing methods address this issue by imposing constraints; however, they often become overly conservative when evaluating OOD regions, which constrains the $Q$-function generalization. This over-constraint issue results in poor $Q$-value estimation and hinders policy improvement. In this paper, we introduce a novel approach to achieve better $Q$-value estimation by enhancing $Q$-function generalization in OOD regions within Convex Hull and its Neighborhood (CHN). Under the safety generalization guarantees of the CHN, we propose the Smooth Bellman Operator (SBO), which updates OOD $Q$-values by smoothing them with neighboring in-sample $Q$-values. We theoretically show that SBO approximates true $Q$-values for both in-sample and OOD actions within the CHN. Our practical algorithm, Smooth Q-function OOD Generalization (SQOG), empirically alleviates the over-constraint issue, achieving near-accurate $Q$-value estimation. On the D4RL benchmarks, SQOG outperforms existing state-of-the-art methods in both performance and computational efficiency. Code is available at https://github.com/yqpqry/SQOG.

## 1 Introduction

Reinforcement Learning (RL) offers a powerful framework for control tasks, underpinned by rigorous mathematical principles. In online RL, an agent learns optimal strategies through direct interaction with the environment. However, in many real-world domains (e.g., robotics, healthcare, autonomous driving), such interactions are infeasible or impractical. Offline RL, in contrast, enables the agent to learn optimal policies from pre-collected datasets, eliminating the need for online interaction. Combining this data-driven paradigm with deep neural networks (DNNs) is anticipated to produce robust and generalizable decision-making engines. However, the primary challenge in offline RL lies in the distribution shift between the learned policy and the behavior policy, which leads to overestimation of OOD actions. Incorrect evaluation of OOD actions results in extrapolation errors, which are further exacerbated by bootstrapping, ultimately causing severe value function approximation errors and hindering the agent from learning an optimal policy.

Recent model-free offline RL algorithms addresses this challenge through the following methods: 1) policy constraints, where constraints are added during policy updates to ensure the learned policies remain close to the behavior policy (Fujimoto et al., 2019; Wu et al., 2019; Kumar et al., 2019; Fujimoto & Gu, 2021; Kostrikov et al., 2021a; Ran et al., 2023; Li et al., 2022; Huang et al., 2023). 2) value penalization, where constraints are incorporated during value updates to enforce conservatism in the value function (Wu et al., 2019; Kostrikov et al., 2021a; Kumar et al., 2020; Lyu et al., 2022; Yang et al., 2022). 3) in-sample learning, where the value function is learned only within the dataset

---
*Corresponding Authors. Emails: {faguo,xiao.zh}@buaa.edu.cn.

to avoid evaluating OOD samples (Wang et al., 2018; Chen et al., 2020; Kostrikov et al., 2021b; Xu et al., 2023; Garg et al., 2023). Dataset OOD regions are typically regarded as information-deficient and potentially hazardous. Avoiding evaluations in these regions helps mitigate the overestimation issue. However, many existing methods tend to be overly conservative in handling dataset OOD regions, which limits the $Q$-function's ability to generalize effectively. This phenomenon, termed the over-constraint issue, results in poor $Q$-value estimation. This raises an important question: *Can we achieve better Q-value estimation by enhancing Q-function generalization in dataset OOD regions?*

To address this question, we first introduce the CHN to define a boundary for safety generalization. By providing two safety guarantees, we demonstrate that generalizing the $Q$-function to OOD regions within the CHN is safe, while doing so outside the CHN is risky. Therefore, we focus on $Q$-function generalization within the CHN. To enhance this generalization, we propose the SBO, which incorporates a smooth generalization term into the empirical Bellman operator. The key idea is to adjust biased OOD $Q$-values using neighboring in-sample $Q$-values that closely approximate the true values. We provide a theoretical justification for the SBO, showing that the smooth generalization term is appropriate. Additionally, we analyze its effects on both in-sample and OOD evaluations and establish its convergence properties. Applying the SBO allows the $Q$-function to gradually approximate the true OOD $Q$-values, while minimally affecting in-sample evaluations. In theory, SBO yields a more accurate $Q$-function for policy evaluation.

Building on SBO, we develop a computationally efficient offline RL algorithm: Smooth Q-function OOD Generalization (SQOG). Empirically, we demonstrate that compared to TD3+BC (Fujimoto & Gu, 2021), SQOG achieves more accurate $Q$-value estimation, particularly in OOD regions within the CHN, thereby alleviating the over-constraint issue of the $Q$-function. Finally, on the D4RL benchmarks, SQOG shows superior performance and computational efficiency compared to existing state-of-the-art methods.

To summarize, our contributions are as follows:

- Under the safety guarantees of the CHN, we propose the Smooth Bellman Operator (SBO), which enhances $Q$-function generalization in OOD regions and approximates the true $Q$-values.
- Building on SBO, we design an effective algorithm, SQOG, which alleviates the over-constraint issue and obtains SOTA results on benchmark datasets.

## 2 PRELIMINARIES

RL is typically modeled as a Markov Decision Process (MDP) (Sutton & Barto, 2018), defined as $M = (S, A, T, d_0, r, \gamma)$. $S$ represents the state space, $A$ denotes the action space, and $T$ describes the conditional probability of state transitions $T(s_{t+1}|s_t, a_t)$ (simply denoted as $T(s'|s, a)$). The initial state distribution is defined by $d_0(s_0)$, the reward function is $r : S \times A \to \mathbb{R}$, and $\gamma \in [0, 1)$ is the discount factor. The objective of RL is to learn an optimal policy $\pi$ that maximizes the cumulative expected reward $J(\pi) = \mathbb{E}_{s_0 \sim d, a_t \sim \pi(\cdot|s_t), s_{t+1} \sim T}[\sum_{t=0}^{\infty} \gamma^t r(s_t, a_t)]$. The state-action value function $Q^\pi(s, a)$ quantifies the discounted return of a trajectory starting from state $s$ and action $a$, following the policy $\pi$. The reward function is bounded, i.e. $|r(s, a)| \leq r_{\max}$. Given a policy $\pi$, the Bellman operator for the $Q$ function's iteration is defined as: $\mathcal{B}^\pi Q(s, a) = r(s, a) + \gamma \mathbb{E}_{s' \sim T, a' \sim \pi(\cdot|s')}[Q(s', a')]$.

Offline RL algorithms based on dynamic programming maintain a parametric $Q$-function $Q_\theta(s, a)$ and optionally a parametric policy $\pi_\phi(a|s)$. The dataset is typically defined as $\mathcal{D} = \{(s_i, a_i, r_i, s'_i, d_i)\}_{i=1}^{N}$, where $d_i \in \{0, 1\}$ is the done flag. The dataset is generated according to the behavior policy $\mu(\cdot|s)$. Given state $s' \in \mathcal{D}$, the empirical behavior policy is defined as: $\hat{\mu}(a'|s') := \frac{\sum_{(s,a) \in \mathcal{D}} \mathbf{1}[s=s', a=a']}{\sum_{s \in \mathcal{D}} \mathbf{1}[s=s']}$. The Actor-Critic algorithm is widely used in RL, consisting of policy evaluation in Eq. (1) and policy improvement in Eq. (2).

$$\mathcal{L}_{critic}(\theta) = \mathbb{E}_{(s,a,r,s') \sim \mathcal{D}}[(Q_\theta(s, a) - (r + \gamma \mathbb{E}_{a' \sim \pi_\phi(\cdot|\mathbf{s}')}[\hat{Q}_{\theta'}(s', a')]))^2] \tag{1}$$

$$\mathcal{J}_{actor}(\phi) = -\mathbb{E}_{s \sim \mathcal{D}, a \sim \pi_\phi(\cdot|\mathbf{s}')}[Q_\theta(s, a)] \tag{2}$$

Since $\mathcal{D}$ typically does not contain all possible transitions $(s, a, s')$, the policy evaluation step uses an empirical Bellman operator $\hat{\mathcal{B}}^\pi Q_\theta(s, a) = r + \gamma \mathbb{E}_{s' \sim D, a' \sim \pi_\phi(\cdot|\mathbf{s}')}[Q_{\theta'}(s', a')]$ that only backs up a

single sample. The empirical Bellman operator relies on $a'$ sampled from learned policy $\pi_\phi(\cdot|s')$. In offline RL, $a'$ may not correspond to any action in the dataset, typically when $\hat{\mu}(a'|s') = 0$. We refer to such actions as OOD actions[1], which are usually overestimated (Kumar et al., 2019). Most existing methods introduce a new over-constraint issue when addressing the overestimation of OOD actions. We alleviate this issue by enhancing $Q$-function generalization in OOD regions within the CHN.

## 3 $Q$-LEARNING WITH SMOOTH OOD GENERALIZATION IN CHN

In this section, we first formally define the CHN and outline its safety guarantees for distinguishing the safer OOD regions (Section 3.1). Then, we construct the SBO to improve the $Q$ generalization within the CHN. Theoretically, we provide the justification for using SBO, as well as discuss its in-sample and OOD effects (Section 3.2). Finally, we propose our practical algorithm SQOG with a low-computational-cost implementation (Section 3.3).

### 3.1 CONVEX HULL AND ITS NEIGHBORHOOD

**Definition 1** (CHN, Convex Hull and its Neighborhood). *For a given dataset $\mathcal{D}$, we define the in-sample state-action set $(S, A)_\mathcal{D} = \{(s, a)|(s, a) \in \mathcal{D}\}$[2]. CHN[3] is defined as the union of the convex hull and its neighborhood of $(S, A)_\mathcal{D}$: $CHN(\mathcal{D}) = Conv(\mathcal{D}) \cup N(Conv(\mathcal{D}))$, where $Conv(\mathcal{D}) = \{\sum_{i=1}^n \lambda_i x_i | \lambda_i \geq 0, \sum_{i=1}^n \lambda_i = 1, x_i \in (S, A)_\mathcal{D}\}$ is convex hull, and $N(Conv(\mathcal{D})) = \{x \in (S, A) \mid x \notin Conv(\mathcal{D}), \min_{y \in Conv(\mathcal{D})} \|x - y\| \leq r\}$[4] is the external neighborhood. The radius $r$ is always chosen to be smaller than or equal to the diameter of $Conv(\mathcal{D})$.*

Definition 1 presents the formal mathematical definition of *CHN*, which possesses uniqueness, compactness and connectivity. We then demonstrate two safety guarantees of *CHN*.

**Proposition 1** (Safety guarantee 1: $Q$-value difference is controlled within *CHN*). *Under the NTK regime, given a dataset $\mathcal{D}$, $x_1 \in Conv(\mathcal{D})$, $x_2 \in N(Conv(\mathcal{D}))$, $x_3 \in (S, A) - CHN(\mathcal{D})$. We have,*

$$\|Q_\theta(x_1) - Q_\theta(\text{Proj}_\mathcal{D}(x_1))\| \leq C_1(\sqrt{\min(\|x_1\|, \|\text{Proj}_\mathcal{D}(x_1)\|)}\sqrt{d_1} + 2d_1) \leq M_1 \tag{3}$$

$$\|Q_\theta(x_2) - Q_\theta(\text{Proj}_\mathcal{D}(x_2))\| \leq C_1(\sqrt{\min(\|x_2\|, \|\text{Proj}_\mathcal{D}(x_2)\|)}\sqrt{d_2} + 2d_2) \leq M_2 \tag{4}$$

$$\|Q_\theta(x_3) - Q_\theta(\text{Proj}_\mathcal{D}(x_3))\| \leq C_1(\sqrt{\min(\|x_3\|, \|\text{Proj}_\mathcal{D}(x_3)\|)}\sqrt{d_3} + 2d_3) \tag{5}$$

*where $\text{Proj}_\mathcal{D}(x) := \text{argmin}_{x_i \in \mathcal{D}}\|x - x_i\|$ is the projection to the dataset. $d_1, d_2, d_3$ are the point-to-dataset distances. Both $d_1 = \|x_1 - \text{Proj}_\mathcal{D}(x_1)\| \leq \max_{x' \in \mathcal{D}}\|x_1 - x'\| \leq B$ and $d_2 = \|x_2 - \text{Proj}_\mathcal{D}(x_2)\| \leq r \leq B$ are bounded. $d_3 = \|x_3 - \text{Proj}_\mathcal{D}(x_3)\| > r$, where $r$ is the external neighborhood radius and $B = \sup\{\|x - y\| \mid x, y \in Conv(\mathcal{D})\}$ is the diameter of the convex hull. let $r = \max_{x \in Conv(\mathcal{D})}\|x - \text{Proj}_\mathcal{D}(x)\|$, then $r \leq B$. $C_1, M_1, M_2$ are constants.*

We generalize Proposition 1 from the analysis of DOGE (Li et al., 2022) under the NTK regime (see Appendix A). The *external neighborhood* is a crucial augmentation that significantly broadens the scope of safety generalization. For any state-action pair $x_1$ (inside the convex hull) or $x_2$ (in the external neighborhood), the difference between its $Q$-value and the in-sample $Q$-value $Q_\theta(\text{Proj}_\mathcal{D}(x))$ can be controlled by the point-to-dataset distance. Due to the uniqueness of *CHN*, this distance $d_i = \|x_i - \text{Proj}_\mathcal{D}(x_i)\|, i = 1, 2$ can be strictly controlled by the longest diameter $B$ of the convex hull. Assuming that deep $Q$-function is a continuous mapping, keeping the compactness (bounded and closed) and connectivity of the set, then $Q$ is bounded within *CHN*. Building upon Proposition 1, we can quantify the bound: $\forall x_{in} \in CHN$, $\|Q_\theta(x_{in})\| \leq \sup_{x_i \in \mathcal{D}}\|Q_\theta(x_i)\| + \max\{M_1, M_2\}$.

**Proposition 2** (Safety guarantee 2: $Q$-function is uniformly continuous within *CHN*). *Assuming that deep $Q$-function is continuous, then deep $Q$-function defined on CHN is uniformly continuous:*

$$\forall \varepsilon > 0, \ \exists \delta > 0, \ s.t. \ \forall x_i, x_j \in CHN(\mathcal{D}), \ if \ \|x_i - x_j\| < \delta, \ then \ \|Q_\theta(x_i) - Q_\theta(x_j)\| < \varepsilon.$$

---

[1] In practice, actions with $\hat{\mu}(a'|s') \approx 0$ are often treated as OOD actions due to their negligible frequency.

[2] Here, $(S, A)_\mathcal{D}$ represents the set of state-action pairs $(s, a)$ extracted from the dataset $\mathcal{D}$. The dataset $\mathcal{D}$ consists of tuples $(s, a, r, s', d)$, but only the $(s, a)$ pairs are included in $(S, A)_\mathcal{D}$, while $(r, s', d)$ are ignored.

[3] We use both *CHN* and the CHN in this paper. The italicized form emphasizes the mathematical structure and properties, while the regular font highlights its conceptual and intuitive meaning.

[4] In this paper, unless otherwise specified, the $\|\cdot\|$ norm refers to the $\mathcal{L}_2$ norm $\|\cdot\|_2$.

Proposition 2 ensures that small input changes will not lead to drastic changes in the output $Q$-value, which means that the $Q$-function of OOD actions within *CHN* is easy to learn from the neighbor $Q$-function of in-sample actions which is more accurate (proved in Section 3.2). Through the safety guarantees in Proposition 1 and 2, we can make it clear that the generalization of the $Q$-function in the OOD regions within *CHN* is safer and more reliable, without producing excessive high estimates. In the following sections, we only focus on the $Q$ generalization in the OOD regions within *CHN*.

### 3.2 Smooth Bellman Operator with OOD generalization

In this section, we introduce a method to actively improve $Q$ generalization in OOD regions within the *CHN*. We start by defining the SBO and providing its theoretical justification through Theorem 1 and Proposition 3. Subsequently, we present its in-sample and OOD effects in Theorems 2 and 3.

**Definition 2.** *Given policy $\pi$, the Smooth Bellman Operator (SBO) is defined as*

$$\widetilde{\mathcal{B}}^\pi Q(s,a) = (\mathcal{G}_1 \hat{\mathcal{B}}_2^\pi)Q(s,a) \tag{6}$$

*where $\mathcal{G}_1$ is the smooth generalization operator:*

$$\mathcal{G}_1 Q(s,a) = \begin{cases} Q(s,a), & \hat{\mu}(a|s) > 0 \\ Q(s,a_{neighbor}^{in}), & \hat{\mu}(a|s) = 0 \text{ and } (s,a) \in CHN \end{cases} \tag{7}$$

*and $\hat{\mathcal{B}}_2$ is the base Bellman operator:*

$$\hat{\mathcal{B}}_2^\pi Q(s,a) = \begin{cases} \hat{\mathcal{B}}^\pi Q(s,a), & \hat{\mu}(a|s) > 0 \\ Q(s,a), & \hat{\mu}(a|s) = 0 \text{ and } (s,a) \in CHN \end{cases} \tag{8}$$

*where $\hat{\mu}(a|s) = 0$ and $(s,a) \in CHN$ implies that $a$ is an OOD action within CHN, $a_{neighbor}^{in}$ denotes a dataset action which is in the neighborhood of the OOD action $a$, i.e., $a_{neighbor}^{in} \in \mathcal{D}$ and $\|a_{neighbor}^{in} - a\| \leq \delta$. $\hat{\mathcal{B}}^\pi Q(s,a)$ denotes the wildly used empirical Bellman operator $\hat{\mathcal{B}}^\pi Q = \mathbb{E}_{s,a,r,s'\sim\mathcal{D}}[r + \gamma \mathbb{E}_{a'\sim\pi(\cdot|\mathbf{s}')}[Q(s',a')]]$. For simplicity, we omit the network parameters $\theta$, $\theta'$ and $\phi$.*

In the SBO, in-sample $Q$-values are updated using the empirical Bellman backup in $\hat{\mathcal{B}}_2^\pi Q_\theta$, while OOD $Q$-values within *CHN* are updated using the neighboring in-sample $Q$-values $Q(s, a_{neighbor}^{in})$ in $\mathcal{G}_1 Q_\theta$. Inspired by the MCB operator in MCQ (Lyu et al., 2022), we decompose the operator into $\mathcal{G}_1 Q_\theta$ and $\hat{\mathcal{B}}_2^\pi Q_\theta$ to address the potential OOD actions generated by $\pi(\cdot|s')$. The smooth generalization operator $\mathcal{G}_1$ conveys the key contribution of the SBO. Through the following Theorem 1 and Proposition 3, we will provide theoretical justification for $\mathcal{G}_1$.

**Theorem 1** (The empirical Bellman operator $\hat{\mathcal{B}}^\pi Q_\theta$ is close to $\mathcal{B}^\pi Q_\theta$). *Suppose there exist a policy constraint offline RL algorithm such that the KL-divergence of learned policy $\pi$ and the behavior policy $\mu$ is optimized to guarantee $\max(KL(\pi,\mu), KL(\mu,\pi)) \leq \epsilon$. Then, under the NTK regime, for all $(s,a) \in \mathcal{D}$, with high probability $\geq 1 - \delta$, $\delta \in (0,1)$.*

$$\|\hat{\mathcal{B}}^\pi Q_\theta - \mathcal{B}^\pi Q_\theta\| \leq \underbrace{\frac{C_{r,T,\delta}}{\sqrt{|\mathcal{D}(s,a)|}}}_{\text{sampling error bound}} + \zeta \cdot C \cdot \underbrace{\max_{s'}\left[\sqrt{\min(\mathbb{E}_{a'\sim\pi}\|(s',a')\|, \mathbb{E}_{a'\sim\mu}\|(s',a')\|)}\sqrt{d} + 2d\right]}_{\text{OOD overestimation error bound}}$$

$$\tag{9}$$

*where $C_{r,T,\delta} = C_{r,\delta} + \gamma C_{T,\delta} R_{max}/(1-\gamma)$, $\zeta = \frac{\gamma C_{T,\delta}}{\sqrt{|\mathcal{D}(s,a)|}}$ and $d \leq \frac{\|a_{min}\|^2 + \|a_{max}\|^2}{2}\sqrt{\frac{\epsilon}{2}}$. Here, $C_{r,\delta}$ and $C_{T,\delta}$ are constants dependent on the concentration properties of $r(s,a)$ and $T(s'|s,a)$, $|\mathcal{D}(s,a)|$ is the dataset size, $a_{min}$ and $a_{max}$ denote the minimum and maximum actions, and $C$ is a constant.*

*Proof sketch.* The proof consists of considering two main sources of error: the sampling error (arising from $r$ and $\hat{T}$), and the OOD overestimation error (generated from $\mathbb{E}_{a'\sim\pi(\cdot|\mathbf{s}')}[Q_{\theta'}(s',a')]$). Since $\hat{\mathcal{B}}^\mu Q_\theta$ has low OOD overestimation error and $\mu$ is close to $\pi$, we first analyze the sampling error through $\|\hat{\mathcal{B}}^\mu Q_\theta - \mathcal{B}^\mu Q_\theta\|$, which can be bounded by $\frac{C_{r,T,\delta}}{\sqrt{|\mathcal{D}(s,a)|}}$. The OOD overestimation error is then examined as the difference between $\mathbb{E}_{a'\sim\pi(\cdot|\mathbf{s}')}[Q_{\theta'}(s',a')]$ and $\mathbb{E}_{a'\sim\mu(\cdot|\mathbf{s}')}[Q_{\theta'}(s',a')]$. Under

the NTK regime, this difference is controlled by the distance $d$. Given that both $\frac{1}{\sqrt{|\mathcal{D}(s,a)|}}$ and $d$ are small, the difference between $\hat{\mathcal{B}}^\pi Q_\theta$ and $\mathcal{B}^\pi Q_\theta$ is expected to be small. See Appendix A. $\qquad\square$

Theorem 1 shows that under the policy constraint algorithm framework, for in-sample $(s, a^{in})$, the empirical $\hat{Q}^\pi_\theta(s, a^{in})$ can closely approximate $Q^\pi_\theta(s, a^{in})$[5] by applying the empirical Bellman operator $\hat{\mathcal{B}}^\pi$. Meanwhile, $Q^\pi_\theta(s, a^{in})$ is close to the true non-parametric $Q$-value $Q^\pi(s, a^{in})$ (Ran et al., 2023). Then, $\hat{Q}^\pi_\theta(s, a^{in})$ is a near-accurate estimation for the in-sample $(s, a^{in})$, i.e. $\hat{Q}^\pi_\theta(s, a^{in}) \approx Q^\pi(s, a^{in})$. However, the OOD $Q$-value $\hat{Q}^\pi_\theta(s, a^{ood})$ may suffer from *underestimation* due to the over-constraint issue, i.e., for some $(s, a^{ood}) \in CHN$, $\hat{Q}^\pi_\theta(s, a^{ood}) < Q^\pi(s, a^{ood})$.

In Eq. (7), the operator $\mathcal{G}_1$ is introduced to approximate the true OOD $Q$-value within the CHN. Although the true OOD $Q^\pi(s, a^{ood})$ and the exact OOD reward $r(s, a^{ood})$ are unattainable, we already obtain the nearly accurate in-sample $\hat{Q}^\pi_\theta(s, a^{in})$. In Proposition 3, we show that $\hat{Q}^\pi_\theta(s, a^{in})$ can serve as a *appropriate* OOD target when combined with the neighboring condition.

**Proposition 3** ($\hat{Q}^\pi_\theta(s, a^{in}_{neighbor})$ is appropriate). *Suppose there exist $\varepsilon$ such that $\|\hat{Q}^\pi_\theta(s, a) - Q^\pi(s, a)\| < \varepsilon/2$, for all $(s, a) \in \mathcal{D}$. For any OOD actions $a^{ood}$ within CHN, by Proposition 2, there exist a small $\delta$, if $\|a^{ood} - a^{in}_{neighbor}\| < \delta$, then $\|Q^\pi(s, a^{ood}) - Q^\pi(s, a^{in}_{neighbor})\| < \varepsilon/2$, we have,*

$$\|Q^\pi(s, a^{ood}) - \hat{Q}^\pi_\theta(s, a^{in}_{neighbor})\| < \varepsilon \tag{10}$$

Proposition 3 can be proved directly using the triangle inequality. Subsequently, we propose Theorem 2 and 3 to illustrate the *effects* of the SBO.

**Theorem 2** (Effects on in-sample evaluation). *For the in-sample evaluation, $\mathcal{G}_1$ introduces negligible changes to the empirical Bellman operator $\hat{\mathcal{B}}^\pi Q_\theta$. Under the NTK regime, **given** $(s, a) \in \mathcal{D}$, assuming that $\forall a' \sim \pi(\cdot|s'), \exists a^{in}_{neighbor}, s.t. \|a' - a^{in}_{neighbor}\| < \delta$, we have,*

$$\|\widetilde{\mathcal{B}}^\pi Q_\theta(s, a) - \hat{\mathcal{B}}^\pi Q_\theta(s, a)\| \le C \cdot \gamma \mathbb{E}_{s'}\left[\sqrt{\min(x, y)}\sqrt{\delta} + 2\delta\right] \tag{11}$$

*where $x = \mathbb{E}_{a' \sim \pi, \|a' - a^{in}_{neighbor}\| < \delta}\|(s', a^{in}_{neighbor})\|$, $y = \mathbb{E}_{a' \sim \pi}\|(s', a')\|$, $C$ is a constant.*

The proof of Theorem 2 is similar to Theorem 1. See Appendix A.

**Theorem 3** (Effects on OOD evaluation). *For the OOD evaluation, $\mathcal{G}_1$ helps mitigate underestimation and overestimation. Assuming that for all $(s, a) \in \mathcal{D}$, $Q^k(s, a) \approx Q^\pi(s, a)$. **Given** $(s, a) \notin \mathcal{D}$ and $(s, a) \in CHN$, assuming that $\|a - a^{in}_{neighbor}\| \le \delta$ and $\|Q^\pi(s, a) - Q^k(s, a^{in}_{neighbor})\| < \varepsilon$, if $\|Q^k(s, a) - Q^k(s, a^{in}_{neighbor})\| > \varepsilon$ (underestimation or overestimation), by applying the SBO for gradient descent updates (with infinitesimally small learning rate), we have,*

$$\|Q^\pi(s, a) - Q^{k+1}(s, a)\| < \|Q^\pi(s, a) - Q^k(s, a)\| \tag{12}$$

*If $\|Q^k(s, a) - Q^k(s, a^{in}_{neighbor})\| \le \varepsilon$, then $\|Q^{k+1}(s, a) - Q^\pi(s, a)\| < 2\varepsilon$.*

*Proof sketch.* If $Q^k(s, a) < Q^k(s, a^{in}_{neighbor}) - \varepsilon$ (underestimation), then by applying the gradient descent method, it follows that $Q^k(s, a) < Q^{k+1}(s, a) \le Q^\pi(s, a)$. Similarly, the overestimation case can be addressed. The final result can be established using a similar approach combined with the triangle inequality. See Appendix A. $\qquad\square$

From Theorem 2 and 3, we observe that applying the SBO, $\widetilde{\mathcal{B}}^\pi$, enables the $Q$-function to gradually approximate the true OOD $Q$-values within the CHN, while incurring negligible side effects on the in-sample evaluation. Finally, we present the convergence of the SBO with its proof in Appendix A.

It is worth noting that the SBO is designed specifically to handle OOD $Q$-values within the CHN, adhering to the safety guarantees. Extending this approach to learn $Q$-values for faraway OOD actions, which fall outside the CHN, remains an open challenge in the field.

---

[5]Assuming that parametric in-sample $Q$-value $Q^\pi_\theta(s, a^{in})$ is converged, i.e. $Q^\pi_\theta(s, a^{in}) \approx \mathcal{B}^\pi Q^\pi_\theta(s, a^{in})$.

---

**Algorithm 1 S**mooth **Q**-function **O**OD **G**eneralization (SQOG)

---

1: **Initialize:** Actor network parameter $\phi$, critic network parameters $\theta_1$ and $\theta_2$, dataset $\mathcal{D}$, target parameters $\phi', \theta_1', \theta_2'$, training step $T$, smoothing parameter $\tau$, actor update frequency $m$.
2: **for** step $t = 1$ to $T$ **do**
3:     Sample a mini-batch of transitions $\{(s, a, r, s', d)\}$ from $\mathcal{D}$.
4:     Update $\theta_1, \theta_2$ via minimizing the critic loss Eq. (14).
5:     **if** $t$ mod $m = 0$ **then**
6:         Update $\phi$ via minimizing the actor loss Eq. (15).
7:         Update target network parameters by $\phi' \leftarrow (1 - \tau)\phi' + \tau\phi, \theta_i' \leftarrow (1 - \tau)\theta_i' + \tau\theta_i, i = 1, 2$
8:     **end if**
9: **end for**

---

### 3.3 PRACTICAL ALGORITHM

Based on the smooth generalization operator $\mathcal{G}_1$ in SBO, we design an OOD generalization loss $\mathcal{L}_{OG}$ in Eq. (13), which can be easily integrated into the objective function of critic network Eq. (1):

$$\mathcal{L}_{OG}(\theta) = \mathbb{E}_{s \sim \mathcal{D}, a^{ood}} \left[ \left( Q_\theta(s, a^{ood}) - Q(s, a^{in}_{neighbor}) \right)^2 \right] \tag{13}$$

In practice, we aim to devise a low-computational-cost implementation for $a^{ood}$ and $a^{in}_{neighbor}$. Notably, during each training loop, we randomly sample a batch of state-action pairs from the dataset, which *naturally* yields an in-sample action $a^{in}$. By adding noise $\eta$ to $a^{in}$, we generate a neighboring action $a^{ood} = a^{in} + \eta$[6] ensuring that $\|a^{in} - a^{ood}\| \leq \delta$ and $(s, a^{ood}) \in$ *CHN* are satisfied by appropriately controlling the noise scale. This approach allows us to sample pairs of $a^{in}_{neighbor}$ and $a^{ood}$ with minimal computational cost. Consequently, by combining with Eq. (1) and (13), we achieve a practical $Q$-learning objective function of critic networks with low-computational-cost:

$$\mathcal{L}_{SQOG}(\theta_i) = \mathbb{E}_{(s,a,r,s') \sim \mathcal{D}} \left[ \left( Q_{\theta_i}(s, a) - \left( r + \gamma \min_i \hat{Q}_{\theta_i'}(s', a') \right) \right)^2 \right]$$
$$+ \beta \mathbb{E}_{(s,a) \sim \mathcal{D}} \left[ \left( Q_{\theta_i}(s, a + \eta) - \bar{Q}_{\theta_i}(s, a) \right)^2 \right] \tag{14}$$

where $\hat{Q}_{\theta_i'}(s', a')$ represents the $Q$ target network outputs, $a' = \pi_\phi(s)$, $\bar{Q}_{\theta_i}(s, a)$ is the $Q$ network output with the gradient detached, $i \in \{1, 2\}$. Similar to TD3+BC (Fujimoto & Gu, 2021), a representative offline Actor-Critic algorithm, we set the objective function of actor network as:

$$\mathcal{J}(\phi) = -\mathbb{E}_{(s,a) \sim \mathcal{D}}[\lambda Q_{\theta_1}(s, \pi_\phi(s)) - (\pi_\phi(s) - a)^2] \tag{15}$$

where $\lambda = \alpha N / \sum_{s_i, a_i} Q(s_i, a_i)$, $\alpha$ is a hyperparameter, $N$ is the batch-size. The pseudo-code of SQOG is in Algorithm 1. Further discussions on SQOG are provided in Appendix D and E.

## 4 EXPERIMENTS

In this section, we first empirically show that compared to TD3+BC, SQOG alleviates the over-constraint issue, leading to more accurate $Q$-value estimation. Second, we highlight the advantages of our algorithm on the D4RL benchmarks (Fu et al., 2021), where SQOG demonstrates superior performance and computational efficiency. Finally, we present an ablation study to analyze the contributions of the key components in our approach.

**Sanity check: alleviation of the over-constraint issue.** To demonstrate the alleviation of the over-constraint issue, we construct a dataset using the Mujoco environment "Inverted Double Pendulum", chosen for its one-dimensional action space and appropriate task complexity (see Appendix B.2 for wider evidence on high-dimensional tasks). The dataset is generated by training a policy online using Soft Actor-Critic (Haarnoja et al., 2018) and subsequently collecting 1 million samples from the trained policy. We select two key states (the most frequently occurring ones in the dataset) to

---

[6]Note that the superscript ood is used to distinguish from the in-dataset real actions. If $a^{ood}$ is in-sample, the added noise will provide robustness for training the in-sample $Q$. Similar to (Lyu et al., 2022).

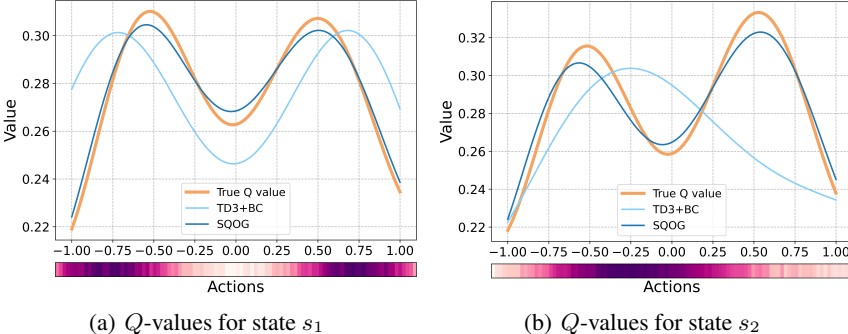

(a) $Q$-values for state $s_1$                    (b) $Q$-values for state $s_2$

Figure 1: $Q$-values estimation for two key states. The color bars show the density of different actions. Higher density actions correspond to darker colors. With the tight constraints of the behavior policy, the $Q$-values of TD3+BC are overly constrained within [-0.50, 0.50] as shown in Figure 1(a), while in Figure 1(b), the $Q$-values are overly constrained within [-0.80, -0.40] and [0.25, 1.00]. However, SQOG consistently achieves accurate estimation of $Q$-values in most cases.

Table 1: Normalized average score comparison of SQOG against baseline methods on D4RL benchmarks over the final 10 evaluations and 4 random seeds. We bold the highest scores.

| Dataset | BC | TD3+BC | CQL | IQL | DOGE | MCQ | SQOG |
|---|---|---|---|---|---|---|---|
| halfcheetah-r | 2.2±0.0 | 11.0±1.1 | 17.5±1.5 | 13.1±1.3 | 17.8±1.2 | 23.6±0.8 | **25.6±0.4** |
| hopper-r | 3.7±0.6 | 8.5±0.6 | 7.9±0.4 | 7.9±0.2 | 21.1±12.6 | **31.0±1.7** | 15.6±3.3 |
| walker2d-r | 1.3±0.1 | 1.6±1.7 | 5.1±1.3 | 5.4±1.2 | 0.9±2.4 | 10.3±6.8 | **17.7±3.5** |
| halfcheetah-m | 43.2±0.6 | 48.3±0.3 | 47.0±0.5 | 47.4±0.2 | 45.3±0.6 | 58.3±1.3 | **59.2±2.4** |
| hopper-m | 54.1±3.8 | 59.3±4.2 | 53.0±28.5 | 66.2±5.7 | 98.6±2.1 | 73.6±10.3 | **100.6±0.7** |
| walker2d-m | 70.9±11.0 | 83.7±2.1 | 73.3±17.7 | 78.3±8.7 | 86.8±0.8 | **88.4±1.3** | 82.9±0.8 |
| halfcheetah-m-r | 37.6±2.1 | 44.6±0.5 | 45.5±0.7 | 44.2±1.2 | 42.8±0.6 | **51.5±0.2** | 46.4±1.2 |
| hopper-m-r | 16.6±4.8 | 60.9±18.8 | 88.7±12.9 | 94.7±8.6 | 76.2±17.7 | 99.5±1.7 | **100.9±5.1** |
| walker2d-m-r | 20.3±9.8 | 81.8±5.5 | 81.8±2.7 | 73.8±7.1 | 87.3±2.3 | 83.3±1.9 | **88.3±3.5** |
| halfcheetah-m-e | 44.0±1.6 | 90.7±4.3 | 75.6±25.7 | 86.7±5.3 | 78.7±8.4 | 85.4±3.4 | **92.6±0.4** |
| hopper-m-e | 53.9±4.7 | 98.0±9.4 | 105.6±12.9 | 91.5±14.3 | 102.7±5.2 | 106.1±2.3 | **109.2±2.8** |
| walker2d-m-e | 90.1±13.2 | 110.1±0.5 | 107.9±1.6 | 109.6±1.0 | **110.4±1.5** | 110.3±0.1 | 109.0±0.3 |
| Mujoco Average | 36.5 | 58.2 | 61.8 | 59.9 | 64.1 | 68.4 | **70.7** |
| Maze2d Average | -2.0 | 35.0 | 19.6 | 37.2 | - | 102.2 | **124.7** |
| Adroit Total | 93.9 | 0.0 | 93.6 | 110.7 | - | 123.3 | **149.6** |
| Runtime (h) | 0.3 | 0.4 | 10.8 | 0.4 | 0.9 | 8.0 | 0.4 |

illustrate the estimation of $Q$-values and use TD3+BC to highlight the over-constraint issue. For each state, we compute the $Q$-values for every 0.01 increment within the action range [-1.0, 1.0], using the critic networks of TD3+BC and SQOG. The true $Q$-values are obtained by a Monte Carlo method, where the discounted return is computed for the same state-action pairs under the same policy. To facilitate comparison, we smooth the values using cubic spline interpolation.

Based on the color bars in Figure 1, we can identify the OOD regions. In Figure 1(a), the highest true value occurs within the range [-0.50, 0.50], corresponding to OOD regions inside the **convex hull**. In Figure 1(b), the highest true value is located within [0.30, 1.00], representing OOD regions in the **neighborhood** of the convex hull. However, TD3+BC struggles with the over-constraint issue in these OOD regions, failing to accurately estimate $Q$-values for policy evaluation. In contrast, SQOG successfully estimates $Q$-values by smoothly generalizing in the OOD regions within the CHN (inside the convex hull in 1(a) and its neighborhood in 1(b)). These results from our sanity check demonstrate that improving $Q$-value generalization in the OOD regions within the CHN leads to better policy evaluation, reinforcing our theoretical analysis.

**Results on D4RL benchmarks.** We evaluate our proposed approach on the D4RL benchmarks of OpenAI gym Mujoco locomotion tasks (Brockman et al., 2016; Todorov et al., 2012). For baselines, we choose representative offline model-free algorithms of different categories including BC, TD3+BC

(Fujimoto & Gu, 2021), CQL (Kumar et al., 2020), IQL (Kostrikov et al., 2021b), as well as including DOGE (Li et al., 2022), MCQ (Lyu et al., 2022) due to their high performance. For fairness, we choose four types of the "-v2" datasets (r:random, m:medium, m-r:medium-replay, m-e:medium-expert) for all methods, yielding a total of 12 datasets. We conduct additional experiments on 4 Maze2d "-v1" datasets and 8 Adroit (Rajeswaran et al., 2017) "-v1" datasets (see Appendix B). The results of BC, TD3+BC, CQL, IQL and MCQ are obtained from MCQ paper (Lyu et al., 2022), DOGE (Li et al., 2022) is obtained from its own paper. All methods are run for 1 M gradient steps.

In Table 1, we present the Mujoco results alongside the average scores for Maze2d and total scores for Adroit tasks. For detailed results, please refer to Appendix B. As demonstrated, SQOG consistently attains the highest scores on most datasets and achieves the highest average scores across the Mujoco, Maze2d, and Adroit tasks. The second-ranking MCQ algorithm approaches ours in average score but runs **20 times** slower. Compared to the representative method TD3+BC, SQOG demonstrates a **70%** performance improvement on hopper-medium-v2 dataset and significant improvements in average scores, with minimal increase in computational cost. Based on the benchmark results, SQOG exhibits performance improvement, which aligns with our theoretical analysis indicating that SQOG achieves better evaluation.

**Computational cost.** Time complexity is a significant challenge in offline RL. We evaluate run time of training each offline RL algorithms for 1 million time steps, using the author-provided implementations or the re-implementations of the source code using JAX. The results are reported in Figure 2. In contrast to MCQ and CQL, our approach significantly reduces computational costs by avoiding the use of a generative model for behavior modeling (Figure 3). Compared to TD3+BC, the supplementary OOD generalization term is computationally-free due to the low-computational-cost implementation of our OOD sampling methods.

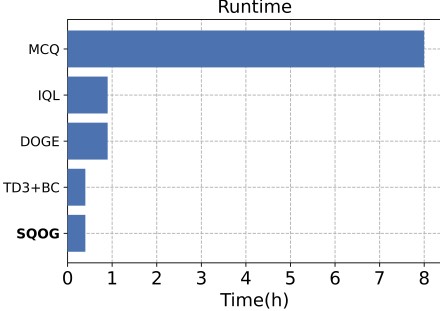

Figure 2: Average run time on Mujoco locomotion tasks.

| Key Features | MCQ | SQOG |
|---|---|---|
| Loss Modification | Critic | Critic |
| Generative Model | CVAE | None |
| OOD Sampling | From $\pi$ | Add noise |

Figure 3: The key features of SQOG and MCQ (Lyu et al., 2022). The use of CVAE makes MCQ time consuming. In contrast, SQOG avoids the use of any generative model, achieving SOTA results with low computational cost.

**Ablation study.** We conduct ablation studies on hyperparameter $\beta$ and the noise type. The hyperparameter $\beta$ controls the significance of the OOD generalization term in $Q$-learning, specifically balancing the learning weight between the OOD $Q$-values and the in-sample $Q$-values. We investigate the effects of four different values of $\beta$ to understand its impact. Our findings (Figure 4(a), 4(b), 4(c)) indicate that $\beta = 0.5$ generally yields optimal performance. Larger values of $\beta$ make it difficult to achieve accurate in-sample $Q$-values, while smaller values of $\beta$ hinder the learning of OOD $Q$-values, leading to reduced performance. Additional study on hyperparameter $\alpha$ is provided in Appendix B.3.

Additionally, we examined two commonly used types of noise, both constrained to the same range [-0.5, 0.5]. As shown in Figures 4(d), 4(e) and 4(f), the normal noise with clipping appears to be a straightforward and effective choice for dataset noise. In contrast, uniform noise tends to generate overly random OOD samples that are sparsely and evenly distributed, which may limit its ability to focus on critical OOD regions. Further details, including an additional study on noise clipping, are provided in Appendix B.4 and B.5.

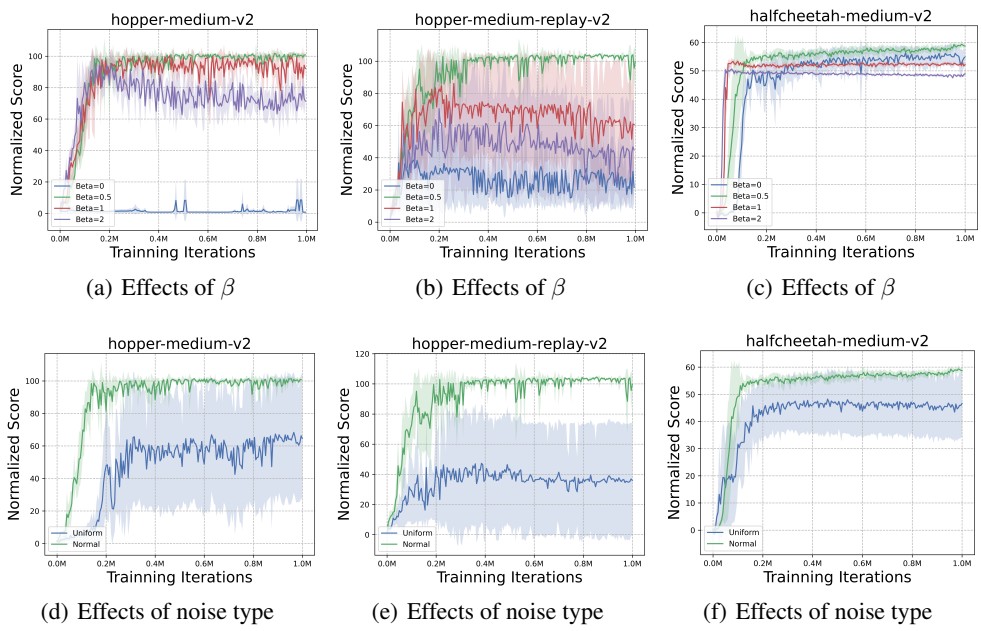

Figure 4: Hyperparameter study and noise study on hopper-medium-v2, hopper-medium-replay-v2, halfcheetah-medium-v2. The experiments are run for 1M gradient steps over 4 random seeds.

## 5 RELATED WORK

**Dataset structure.** Manifold learning (Roweis & Saul, 2000; Tenenbaum et al., 2000; Coifman & Lafon, 2006; Belkin & Niyogi, 2001; Wang et al., 2004; Barannikov et al., 2021; Hegde et al., 2007; Brehmer & Cranmer, 2020) is a kind of dataset structure learning method aiming to uncover the underlying structure of high-dimensional data. However, manifold learning faces challenges when applied to offline RL datasets due to the complexity of the data and the trajectory information in the high dimensional state-action space. Convex hull is another dataset structure used in dataset analysis and algorithm designation in supervised learning settings such as classification and regression (Fung et al., 2005; Khosravani et al., 2016; Xiong et al., 2014; Nemirko & Dulá, 2021; Xu et al., 2021). Understanding the structure of RL datasets facilitates the design of superior algorithms, enhancing their performance, stability, and generalization capability (Wang et al., 2020; Schweighofer et al., 2022; Hong et al., 2023). DOGE (Li et al., 2022) was the first to apply the convex hull in the design of offline RL algorithms. SEABO (Lyu et al., 2024), PRDC (Ran et al., 2023) and (Sun et al., 2023) utilize nearest neighbor techniques to address practical challenges. Building upon these previous works, we propose a new dataset structure called CHN and analyze its safety guarantees for generalization.

**Model-free offline RL.** Offline RL algorithms address the challenge of distribution shift by employing different strategies. Model-free offline RL can be broadly categorized into policy-based approaches (Wang et al., 2018; Fujimoto et al., 2019; Peng et al., 2019; Ran et al., 2023; Wu et al., 2019; Fujimoto & Gu, 2021; Kostrikov et al., 2021b;a; Kumar et al., 2019; Li et al., 2022; Mao et al., 2024) and value-based approaches (Kumar et al., 2020; Kostrikov et al., 2021b; Lyu et al., 2022; An et al., 2021; Ghasemipour et al., 2022; Xu et al., 2023; Zhang et al., 2023; Yang et al., 2024; Lee et al., 2024; Geng et al., 2024). Policy-based approaches like BCQ (Fujimoto et al., 2019), BRAC (Wu et al., 2019) and TD3+BC (Fujimoto & Gu, 2021) solely rely on the distribution of behavior policy, leading to overly constrained learned policies. PRDC (Ran et al., 2023) relaxes policy constraints by utilizing the entire dataset, while DOGE (Li et al., 2022) introduces a novel policy constraint that enables exploitation in OOD areas within the dataset convex hull. Instead of directly modifying the constraint term during policy improvement, we focus on policy evaluation. Our method, MQOG, achieves better policy evaluation, which indirectly addresses the over-constraint issue arising from policy improvement. Compared to value-based approaches like IQL (Kostrikov et al., 2021b) and MCQ (Lyu et al., 2022), which approximate the optimal in-sample $Q$ for $Q$-learning, MQOG aims to approximate the true $Q$ for policy evaluation. Our theoretical and empirical results demonstrate that improved evaluation leads to enhanced performance.

## 6   CONCLUSION

In this paper, we introduce Smooth Q-function OOD Generalization (SQOG) to achieve better policy evaluation by improving $Q$ generalization in OOD regions within the CHN. We provide the safety guarantees for the CHN, indicating that our approach to OOD generalization is unlikely to adversely affect evaluation. SQOG trains OOD $Q$-values by constructing appropriate OOD target values with guidance from the Smooth Bellman Operator (SBO). Specifically, we use the neighboring in-sample $Q$-values to update OOD $Q$-values in the SBO. Theoretically, we demonstrate that the SBO is appropriate and enables the $Q$-function to approximate the true $Q$-values within the CHN. Furthermore, we conduct a sanity check to show that SQOG achieve better $Q$ estimation by improving the generalization in OOD regions, thereby alleviating the over-constraint issue. Experiments on the D4RL benchmarks demonstrate that SQOG outperforms baseline methods across most datasets, highlighting the importance of accurate evaluation in OOD regions within the CHN. We anticipate that our work will draw greater attention to $Q$-value estimation in OOD regions and provide new insights for the offline RL community.

Finally, it is crucial to emphasize that precisely solving the CHN and identifying all OOD regions within CHN for each dataset is impractical and unnecessary. However, various ingenious approaches can be employed to improve the estimation of the OOD $Q$-values within the CHN.

ACKNOWLEDGEMENT

This work was supported by the National Science and Technology Major Project under Grant 2022ZD0116401, the National Natural Science Foundation of China under Grant 62141605, and the Fundamental Research Funds for the Central Universities, China. We acknowledge the support of high-performance computing resources provided by the School of Mathematical Sciences of Beihang University. We would like to thank Xiaotie Deng, Xu Chu, Yurong Chen, Zhijian Duan and the anonymous reviewers for their insightful comments on improving the paper.

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

## A  ADDITIONAL THEORETICAL RESULTS AND MISSING PROOFS

Before providing the missing proofs, we briefly introduce the Neural Tangent Kernel (NTK) (Jacot et al., 2018). NTK is widely used in the analysis of the generalization, the convergence and optimality of deep RL. In this paper, we complete our proof of Proposition 1 and Theorem 1 and 3 under the NTK regime.

Following DOGE (Li et al., 2022), we introduce Assumption 1 and Lemma 1 and 2.

**Assumption 1.** *We assume the function approximators discussed in our paper are two-layer fully-connected ReLU neural networks with infinity width and are trained with infinitesimally small learning rate unless otherwise specified.*

**Lemma 1** (Smoothness of the kernel map of two-layer ReLU networks). *Let $\phi$ be the kernel map of the neural tangent kernel induced by a two-layer ReLU neural network, $x$ and $y$ be two inputs, then $\phi$ satisfies the following smoothness property.*

$$\|\phi(x) - \phi(y)\| \leq \sqrt{min(\|x\|, \|y\|)\|x - y\|} + 2\|x - y\| \tag{16}$$

For the proof of Lemma 1, we refer the reader to (Li et al., 2022).

**Lemma 2** (Smoothness for deep $Q$-function). *Given two inputs $x$ and $x'$, the distance between these two data points is $d = \|x - x'\|$, $C$ is a finite constant. Then the difference between the output at $x$ and the output at $x'$ can be bounded by:*

$$\|Q_\theta(x) - Q_\theta(x')\| \leq C\sqrt{min(\|x\|, \|x'\|)}\sqrt{d} + 2d \tag{17}$$

For the proof of Lemma 2, we refer the reader to (Bietti & Mairal, 2019).

**Proof of Proposition 1.**   We obtain this theorem by generalizing DOGE's (Li et al., 2022) Theorem 1, which is proved under Neural Tangent Kernel (NTK) regime. We recall the main results in our theorem: Given a point within convex hull $x_1 \in Conv(\mathcal{D})$, a point in the convex hull neighborhood $x_2 \in N(Conv(\mathcal{D}))$, and an external point $x_3 \in (S, A) - CHN(\mathcal{D})$, we have,

$$\|Q_\theta(x_1) - Q_\theta(\text{Proj}_\mathcal{D}(x_1))\| \leq C_1(\sqrt{min(\|x_1\|, \|\text{Proj}_\mathcal{D}(x_1)\|)}\sqrt{d_1} + 2d_1) \leq M_1 \tag{18}$$

$$\|Q_\theta(x_2) - Q_\theta(\text{Proj}_\mathcal{D}(x_2))\| \leq C_1(\sqrt{min(\|x_2\|, \|\text{Proj}_\mathcal{D}(x_2)\|)}\sqrt{d_2} + 2d_2) \leq M_2 \tag{19}$$

$$\|Q_\theta(x_3) - Q_\theta(\text{Proj}_\mathcal{D}(x_3))\| \leq C_1(\sqrt{min(\|x_3\|, \|\text{Proj}_\mathcal{D}(x_3)\|)}\sqrt{d_3} + 2d_3) \tag{20}$$

where $\text{Proj}_\mathcal{D}(x) := \text{argmin}_{x_i \in \mathcal{D}}\|x - x_i\|$ is the projection to the dataset. $d_1, d_2, d_3$ are the point-to-dataset distances. Both $d_1 = \|x_1 - \text{Proj}_\mathcal{D}(x_1)\| \leq max_{x' \in \mathcal{D}}\|x_1 - x'\| \leq B$ and $d_2 = \|x_2 - \text{Proj}_\mathcal{D}(x_2)\| \leq r \leq B$ are bounded. $d_3 = \|x_3 - \text{Proj}_\mathcal{D}(x_3)\| > r$, where $r$ is the neighborhood radius and $B = \sup\{\|x - y\| \mid x, y \in Conv(\mathcal{D})\}$ is the diameter of the convex hull. let $r = max_{x \in Conv(\mathcal{D})}\|x - \text{Proj}_\mathcal{D}(x)\|$, then $r \leq B$. $C_1, M_1, M_2$ are constants.

DOGE classify data points into $x_{in}$ (the interpolate data in convex hull) and $x_{out}$ (the extrapolate data outside convex hull). However, $x_{out}$ can be further divided into $x_2$ (the data in the neighborhood of convex hull) and $x_3$ (the data outside *CHN*).

With Lemma 2, we can derive the following bound:

$$\|Q_\theta(x_1) - Q_\theta(\text{Proj}_\mathcal{D}(x_1))\| \leq C_1(\sqrt{min(\|x_1\|, \|\text{Proj}_\mathcal{D}(x_1)\|)}\sqrt{d_1} + 2d_1)$$
$$\leq C_1(\sqrt{min(\|x_1\|, \|\text{Proj}_\mathcal{D}(x_1)\|)}\sqrt{B} + 2B) \leq M_1$$

Similarly, we have:

$$\|Q_\theta(x_2) - Q_\theta(\text{Proj}_\mathcal{D}(x_2))\| \leq C_1(\sqrt{min(\|x_2\|, \|\text{Proj}_\mathcal{D}(x_2)\|)}\sqrt{d_2} + 2d_2)$$
$$\leq C_1(\sqrt{min(\|x_2\|, \|\text{Proj}_\mathcal{D}(x_2)\|)}\sqrt{r} + 2r) \leq M_2$$

Since $d_3 = \|x_3 - \text{Proj}_D(x_3)\| > r$, Eq. (20) is no longer bounded, indicating that the error grows uncontrollably for data points far outside the neighborhood of the convex hull. However, we demonstrate that the approximation error within the convex hull's neighborhood can still be effectively controlled. To sum up, we expand the safe generalization boundary of $Q$-function.

**Proof of Proposition 2.** The $Q$-function defined on *CHN* is uniformly continuous:

$$\forall \varepsilon > 0, \; \exists \delta > 0, \; s.t. \; \forall x_i, x_j \in CHN(\mathcal{D}), \text{ if } \|x_i - x_j\| < \delta, \text{ then } \|Q(x_i) - Q(x_j)\| < \varepsilon.$$

*Proof.* To simplify the notation, let $X = CHN$, which is compact. For any arbitrary $\varepsilon > 0$, for every $x \in X$, we can find $\delta_x > 0$ such that for all $x' \in B(x, \delta_x)$, $\|Q(x) - Q(x')\| < \varepsilon/2$ (since $Q$ is continuous). The collection $\mathcal{B} = \{B(x, \delta_x) \mid x \in X\}$ is an open cover of $X$. Define $\mathcal{B}_1 = \{B(x, \delta_x/2) \mid x \in X\}$; this is also an open cover of $X$. Since $X$ is compact, there exists a finite subcover of $\mathcal{B}_1$, denoted by $\mathcal{B}_1' = \{B(x_k, \delta_{x_k}/2)\}_{k=1}^n$.

Let $\delta = \min\{\delta_{x_1}, \delta_{x_2}, \ldots, \delta_{x_n}\}/2$. For any $y, z \in X$, if $\|y - z\| < \delta$, without loss of generality, assume $\|y - x_k\| < \delta_{x_k}/2$ for some $k$. By the triangle inequality, $\|z - x_k\| \leq \|z - y\| + \|y - x_k\| < \delta + \delta_{x_k}/2 \leq \delta_{x_k}$. Therefore, $y, z \in B(x_k, \delta_{x_k})$, and by the properties of $Q$ and the triangle inequality, $\|Q(y) - Q(z)\| < \varepsilon$.

Thus, for any $\varepsilon > 0$, there exists $\delta > 0$ such that for all $y, z \in CHN(D)$, if $\|y - z\| < \delta$, then $\|Q(y) - Q(z)\| < \varepsilon$. □

**Proof of Theorem 1.** Assuming that $\max(KL(\pi, \mu), KL(\mu, \pi)) \leq \epsilon$. Then, under the NTK regime, for all $(s, a) \in \mathcal{D}$, with high probability $\geq 1 - \delta$, $\delta \in (0, 1)$.

$$\|\hat{\mathcal{B}}^\pi Q_\theta - \mathcal{B}^\pi Q_\theta\| \leq \underbrace{\frac{C_{r,T,\delta}}{\sqrt{|\mathcal{D}(s,a)|}}}_{\text{sampling error bound}} + \underbrace{\zeta \cdot C \cdot \max_{s'} \left[ \sqrt{\min(\mathbb{E}_{a' \sim \pi} \|(s', a')\|, \mathbb{E}_{a' \sim \mu} \|(s', a')\|)} \sqrt{d} + 2d \right]}_{\text{OOD overestimation error bound}}$$

(21)

where $C_{r,T,\delta} = C_{r,\delta} + \gamma C_{T,\delta} R_{max}/(1 - \gamma)$, $\zeta = \frac{\gamma C_{T,\delta}}{\sqrt{|\mathcal{D}(s,a)|}}$ and $d \leq \frac{\|a_{min}\|^2 + \|a_{max}\|^2}{2} \sqrt{\frac{\epsilon}{2}}$. $C_{r,\delta}$ and $C_{T,\delta}$ is constants dependent on the concentration properties of $r(s, a)$ and $T(s'|s, a)$, $|\mathcal{D}(s,a)|$ is the dataset size, $a_{min}$ and $a_{max}$ denote the minimum and maximum actions. $C$ is a constant. Similar to (Kumar et al., 2020; Auer et al., 2008; Osband & Roy, 2017), we assume concentration properties of the reward function and the transition dynamics.

**Assumption 2.** $\forall (s, a) \in \mathcal{D}$, *with high probability* $\geq 1 - \delta$, *we have,*

$$\|r - r(s, a)\| \leq \frac{C_{r,\delta}}{\sqrt{|\mathcal{D}(s,a)|}}, \quad \|\hat{T}(s'|s, a) - T(s'|s, a)\|_1 \leq \frac{C_{T,\delta}}{\sqrt{|\mathcal{D}(s,a)|}} \tag{22}$$

*where $C_{r,\delta}$ and $C_{T,\delta}$ is constants dependent on the concentration properties of $r(s, a)$ and $T(s'|s, a)$, $|\mathcal{D}(s,a)|$ is the dataset size.*

*Proof.* From Assumption 2, we have,

$$\|\hat{\mathcal{B}}^\pi Q_\theta - \mathcal{B}^\pi Q_\theta\| = \|(r - r(s, a)) + \gamma \sum_{s'} (\hat{T}(s'|s, a) - T(s'|s, a)) \mathbb{E}_\pi Q_{\theta'}(s', a')\|$$

$$\leq \|r - r(s, a)\| + \gamma \|\hat{T}(s'|s, a) - T(s'|s, a)\|_1 \cdot \max_{s'} \|\mathbb{E}_\pi Q_{\theta'}(s', a')\|$$

$$\leq \frac{C_{r,\delta}}{\sqrt{|\mathcal{D}(s,a)|}} + \gamma \frac{C_{T,\delta}}{\sqrt{|\mathcal{D}(s,a)|}} \max_{s'} \|\mathbb{E}_\pi Q_{\theta'}(s', a')\|$$

where $\max_{s'} \|\mathbb{E}_\pi Q_{\theta'}(s', a')\| \leq \max_{s'} \|\mathbb{E}_\pi Q_{\theta'}(s', a') - \mathbb{E}_\mu Q_{\theta'}(s', a')\| + \max_{s', a' \sim \mu} \|Q_{\theta'}(s', a')\|$. For simplicity, we denote $\mathbb{E}_{a' \sim \pi} Q_{\theta'}(s', a')$ as $\mathbb{E}_\pi Q_{\theta'}(s', a')$. From Lemma 2, we have,

$$\|\mathbb{E}_\pi Q_{\theta'}(s', a') - \mathbb{E}_\mu Q_{\theta'}(s', a')\| \leq C \cdot \left[ \sqrt{\min(\mathbb{E}_{a' \sim \pi} \|(s', a')\|, \mathbb{E}_{a' \sim \mu} \|(s', a')\|)} \sqrt{d} + 2d \right]$$

For distance $d$, by applying the Pinsker's Inequality, we have,

$$d = \|\mathbb{E}_{a' \sim \pi}[a'] - \mathbb{E}_{a' \sim \mu}[a']\| = \|\int_A a'(\pi(a'|s') - \mu(a'|s')) \, da'\|$$

$$\leq \int_A \|a'\| \, da' \cdot \underbrace{\sup_{a' \in A} \|\pi(a'|s') - \mu(a'|s')\|}_{\text{Total variation distance}} \leq \frac{\|a_{min}\|^2 + \|a_{max}\|^2}{2} \sqrt{\frac{\epsilon}{2}}$$

Where $a_{min}$ and $a_{max}$ denotes the min and max action, $C$ is a constant. The reward function is bounded, then $\max_{s',a'\sim\mu} \|Q_{\theta'}(s',a')\| \leq R_{max}/(1-\gamma)$. Therefore, we have,

$$\|\hat{\mathcal{B}}^\pi Q_\theta - \mathcal{B}^\pi Q_\theta\| \leq \underbrace{\frac{C_{r,\delta} + \gamma C_{T,\delta} R_{max}/(1-\gamma)}{\sqrt{|\mathcal{D}(s,a)|}}}_{\text{sampling error bound}}$$

$$+ \underbrace{\frac{\gamma C_{T,\delta}}{\sqrt{|\mathcal{D}(s,a)|}} \cdot C \cdot \max_{s'}\left[\sqrt{\min(\mathbb{E}_{a'\sim\pi}\|(s',a')\|, \mathbb{E}_{a'\sim\mu}\|(s',a')\|)}\sqrt{d} + 2d\right]}_{\text{OOD overestimation error bound}}$$

$\square$

**Remark.** One might ask why we can't directly set $\max_{s'} \|\mathbb{E}_\pi Q_{\theta'}(s',a')\| \leq \frac{R_{max}}{1-\gamma}$. The reason is that $\pi$ may generate OOD actions for $Q_{\theta'}(s',a')$, and the neural network could overestimate the $Q$-values for these OOD actions, making it impossible to simply bound them. However, since $\mu$ does not produce OOD actions, we can bound $\max_{s',a'\sim\mu} \|Q_{\theta'}(s',a')\|$ by $\frac{R_{max}}{1-\gamma}$. Therefore, if $\pi$ is not sufficiently close to $\mu$, we cannot bound $\|\hat{\mathcal{B}}^\pi Q_\theta - \mathcal{B}^\pi Q_\theta\|$.

**Proof of Theorem 2.** For the in-sample evaluation, $\mathcal{G}_1$ introduces negligible changes to the empirical Bellman operator $\hat{\mathcal{B}}^\pi Q_\theta$. Under the NTK regime, **given** $(s,a) \in \mathcal{D}$, assuming that $\forall a' \sim \pi(\cdot|s'), \exists a^{in}_{neighbor}, s.t. \|a' - a^{in}_{neighbor}\| < \delta$, we have,

$$\|\widetilde{\mathcal{B}}^\pi Q_\theta(s,a) - \hat{\mathcal{B}}^\pi Q_\theta(s,a)\| \leq C \cdot \gamma\mathbb{E}_{s'}\left[\sqrt{\min(x,y)}\sqrt{\delta} + 2\delta\right] \tag{23}$$

where $x = \mathbb{E}_{a'\sim\pi, \|a'-a^{in}_{neighbor}\|<\delta}\|(s', a^{in}_{neighbor})\|$, $y = \mathbb{E}_{a'\sim\pi}\|(s',a')\|$, $C$ is a constant.

*Proof.* Similar to the proof of Theorem 1, we can derive the inequality from Lemma 2. The distance in this case is bounded by $\delta$ from the assumption. $\square$

**Proof of Theorem 3.** Assuming that for all $(s,a) \in \mathcal{D}$, $Q^k(s,a) \approx Q^\pi(s,a)$. **Given** $(s,a) \notin \mathcal{D}$ and $(s,a) \in CHN$, assuming that $\|a - a^{in}_{neighbor}\| \leq \delta$ and $\|Q^\pi(s,a) - Q^k(s,a^{in}_{neighbor})\| < \varepsilon$, if $\|Q^k(s,a) - Q^k(s,a^{in}_{neighbor})\| > \varepsilon$, by applying the SBO $\widetilde{\mathcal{B}}$ for gradient descent updates (with infinitesimally small learning rate), we have,

$$\|Q^\pi(s,a) - Q^{k+1}(s,a)\| < \|Q^\pi(s,a) - Q^k(s,a)\| \tag{24}$$

If $\|Q^k(s,a) - Q^k(s,a^{in}_{neighbor})\| \leq \varepsilon$, then $\|Q^{k+1}(s,a) - Q^\pi(s,a)\| < 2\varepsilon$.

*Proof.* Given $(s,a) \notin \mathcal{D}$, if $Q^k(s,a) < Q^k(s,a^{in}_{neighbor}) - \varepsilon$, then $Q^k(s,a) < Q^\pi(s,a)$. From the SBO, we define the MSE loss as:

$$\mathcal{L} = [Q^k(s,a) - Q^k(s,a^{in}_{neighbor})]^2$$

for gradient descent updates. Then,

$$Q^{k+1}(s,a) = Q^k(s,a) + 2\alpha[Q^k(s,a^{in}_{neighbor}) - Q^k(s,a)]$$

where the learning rate is infinitesimally small. Consequently, $Q^k(s,a) < Q^{k+1}(s,a) \leq Q^\pi(s,a)$, and we have,

$$\|Q^\pi(s,a) - Q^{k+1}(s,a)\| < \|Q^\pi(s,a) - Q^k(s,a)\|$$

The proof for the case $Q^k(s,a) > Q^k(s,a^{in}_{neighbor}) + \varepsilon$ follows a similar approach.

For $\|Q^k(s,a) - Q^k(s,a^{in}_{neighbor})\| \leq \varepsilon$, if $Q^k(s,a^{in}_{neighbor}) - \varepsilon \leq Q^k(s,a) \leq Q^k(s,a^{in}_{neighbor})$, then by applying the gradient descent method, we have, $Q^k(s,a) \leq Q^{k+1}(s,a) \leq Q^k(s,a^{in}_{neighbor})$. Similarly, if $Q^k(s,a^{in}_{neighbor}) \leq Q^k(s,a) \leq Q^k(s,a^{in}_{neighbor}) + \varepsilon$, then $Q^k(s,a^{in}_{neighbor}) \leq Q^{k+1}(s,a) \leq Q^k(s,a)$. Therefore, $\|Q^{k+1}(s,a) - Q^k(s,a^{in}_{neighbor})\| \leq \varepsilon$ always holds. By the triangle inequality, we have $\|Q^{k+1}_\theta(s,a) - Q^\pi(s,a)\| < 2\varepsilon$.

$\square$

**Proposition 4** (Convergence of SBO). *Assume that the OOD actions generated by $\pi(\cdot|s')$ lie within the CHN. Then the SBO is a $\gamma$- contraction operator in the $\mathcal{L}_\infty$ norm. Any initial Q-function can converge to a unique fixed point by repeatedly applying the SBO.*

**Proof of Proposition 4.** We prove the $\gamma$-contraction property of the SBO operator under the assumption that OOD actions generated by $\pi(\cdot|s')$ lie within the CHN. This assumption can be satisfied with high probability if we incorporate a regularization term into the actor loss to constrain $\pi$ closer to $\mu$, such as using a behavior cloning (BC) loss.

The SBO is a $\gamma$- contraction operator in the $\mathcal{L}_\infty$. Any initial Q-function can converge to a unique fixed point by repeated application of the SBO.

We first recall the definition of smooth Bellman operator,

$$\widetilde{\mathcal{B}}^\pi Q(s,a) = (\mathcal{G}_1 \hat{\mathcal{B}}_2^\pi) Q(s,a) \tag{25}$$

where

$$\mathcal{G}_1 Q(s,a) = \begin{cases} Q(s,a), & \hat{\mu}(a|s) > 0 \\ Q(s,a_{neighbor}^{in}), & \hat{\mu}(a|s) = 0 \text{ and } (s,a) \in CHN \end{cases} \tag{26}$$

$$\hat{\mathcal{B}}_2^\pi Q(s,a) = \begin{cases} \mathbb{E}_{s,a,r,s'\sim\mathcal{D}}[r + \gamma \mathbb{E}_{a'\sim\pi(\cdot|\mathbf{s}')}[Q(s',a')]], & \hat{\mu}(a|s) > 0 \\ Q(s,a), & \hat{\mu}(a|s) = 0 \text{ and } (s,a) \in CHN \end{cases} \tag{27}$$

*Proof.* Given policy $\pi$, let $Q_1$ and $Q_2$ be two arbitrary Q-functions. Since $a \in \mathcal{D}$, we have,

$$\begin{aligned} ||\widetilde{\mathcal{B}}^\pi Q_1 - \widetilde{\mathcal{B}}^\pi Q_2||_\infty &= ||\mathcal{G}_1 \hat{\mathcal{B}}_2^\pi Q_1 - \mathcal{G}_1 \hat{\mathcal{B}}_2^\pi Q_2||_\infty \\ &= \max_{s,a} \mathcal{G}_1 ||r + \gamma \mathbb{E}_{s',a'\sim\pi} Q_1(s',a') - r - \gamma \mathbb{E}_{s',a'\sim\pi} Q_2(s',a')|| \\ &= \max_{s,a} \gamma \mathcal{G}_1 \mathbb{E}_{s',a'\sim\pi} ||Q_1(s',a') - Q_2(s',a')|| \\ &= \max_{s,a} \gamma \mathbb{E}_{s'\sim T, a'\sim\pi} ||\mathcal{G}_1 Q_1(s',a') - \mathcal{G}_1 Q_2(s',a')||. \end{aligned} \tag{28}$$

If $\hat{\mu}(a'|s') > 0$, then $||\mathcal{G}_1 Q_1(s',a') - \mathcal{G}_1 Q_2(s',a')|| = ||Q_1(s',a') - Q_2(s',a')||$, we have,

$$\begin{aligned} ||\widetilde{\mathcal{B}}^\pi Q_1 - \widetilde{\mathcal{B}}^\pi Q_2||_\infty &= \max_{s,a} \gamma \mathbb{E}_{s'\sim T, a'\sim\pi} ||Q_1(s',a') - Q_2(s',a')|| \\ &\leq \gamma \max_{s,a} ||Q_1 - Q_2||_\infty \\ &= \gamma ||Q_1 - Q_2||_\infty \end{aligned} \tag{29}$$

So we can find that $||\widetilde{\mathcal{B}}^\pi Q_1 - \widetilde{\mathcal{B}}^\pi Q_2||_\infty \leq \gamma ||Q_1 - Q_2||_\infty$.

If $\hat{\mu}(a'|s') = 0$ and $(s,a) \in CHN$, then $||\mathcal{G}_1 Q_1(s',a') - \mathcal{G}_1 Q_2(s',a')|| = ||Q_1(s',a_{neighbor}^{in}) - Q_2(s',a_{neighbor}^{in})||$, we have,

$$\begin{aligned} ||\widetilde{\mathcal{B}}^\pi Q_1 - \widetilde{\mathcal{B}}^\pi Q_2||_\infty &= \max_{s,a} \gamma \mathbb{E}_{s'\sim T, a'\sim\pi} ||Q_1(s',a_{neighbor}^{in}) - Q_2(s',a_{neighbor}^{in})||_\infty \\ &\leq \gamma \max_{s,a} ||Q_1 - Q_2||_\infty \\ &= \gamma ||Q_1 - Q_2||_\infty \end{aligned} \tag{30}$$

Combining the results together, we conclude that the smooth Bellman operator is a $\gamma$- contraction operator in the $\mathcal{L}_\infty$. $\square$

# B ADDITIONAL EXPERIMENTS AND EXPERIMENTAL DETAILS

## B.1 RESULTS ON ADDITIONAL DATASETS

Table 2: Normalized average score comparison of SQOG against baseline methods on Maze2d "-v1" and Adroit "-v1" datasets over the final 10 evaluations and 4 random seeds. We bold the highest scores.

| Dataset | BC | CQL | BCQ | TD3+BC | IQL | MCQ | SQOG |
|---|---|---|---|---|---|---|---|
| maze2d-umaze | -3.2 | 18.9 | 49.1 | 25.7±6.1 | 65.3±13.4 | 81.5±23.7 | **126.4±10.4** |
| maze2d-umaze-dense | -6.9 | 14.4 | 48.4 | 39.7±3.8 | 57.8±12.5 | **107.8±3.2** | 100.4±1.9 |
| maze2d-medium | -0.5 | 14.6 | 17.1 | 19.5±4.2 | 23.5±11.1 | 106.8±38.4 | **149.4±2.9** |
| maze2d-medium-dense | 2.7 | 30.5 | 41.1 | 54.9±6.4 | 28.1±16.8 | 112.7±5.5 | **122.7±0.8** |
| Maze2d Average | -2.0 | 19.6 | 38.9 | 35.0 | 37.2 | 102.2 | **124.7** |
| pen-human | 34.4 | 37.5 | 68.9 | 0.0±0.0 | 68.7±8.6 | 68.5±6.5 | **80.0±4.7** |
| door-human | 0.5 | **9.9** | 0.0 | 0.0±0.0 | 3.3±1.3 | 2.3±2.2 | 1.0±1.1 |
| relocate-human | 0.0 | **0.2** | -0.1 | 0.0±0.0 | 0.0±0.0 | 0.1±0.1 | 0.1±0.0 |
| hammer-human | 1.5 | **4.4** | 0.5 | 0.0±0.0 | 1.4±0.6 | 0.3±0.1 | 1.4±0.7 |
| pen-cloned | 56.9 | 39.2 | 44.0 | 0.0±0.0 | 35.3±7.3 | 49.4±4.3 | **66.7±3.4** |
| door-cloned | -0.1 | 0.4 | 0.0 | 0.0±0.0 | 0.5±0.6 | **1.3±0.4** | -0.1±0.0 |
| relocate-cloned | -0.1 | -0.1 | -0.3 | **0.0±0.0** | -0.2±0.0 | **0.0±0.0** | -0.1±0.0 |
| hammer-cloned | 0.8 | 2.1 | 0.4 | 0.0±0.0 | **1.7±1.0** | 1.4±0.5 | 0.6±0.3 |
| Adroit Total | 93.9 | 93.6 | 113.4 | 0.0 | 110.7 | 123.3 | **149.6** |

To further illustrate the effectiveness of the SQOG algorithm, we conduct more experiments on 4 Maze2d "-v1" datasets and 8 Adroit "-v1" datasets from D4RL benchmarks. The Maze2d tasks involve navigating a 2D maze environment from an initial point to a designated goal, presenting intricate pathfinding challenges due to the maze's structural complexity and the presence of obstacles. Conversely, the Adroit environment engages agents in object manipulation within a physics-based simulation, demanding adept control and adaptability across diverse scenarios. The Maze2d datasets feature non-Markovian policies, yielding undirected and multitask data, while Adroit datasets exhibit heightened realism characterized by non-representable policies, narrow data distributions, and sparse rewards.

We compare MCQ against BC, CQL (Kumar et al., 2020), BCQ (Fujimoto et al., 2019), TD3+BC (Fujimoto & Gu, 2021), IQL (Kostrikov et al., 2021b) and MCQ (Lyu et al., 2022). We take the results on these datasets from (Lyu et al., 2022) directly. From Table 2, we observe that SQOG consistently outperforms existing algorithms on the Maze2d datasets, while demonstrating competitiveness with prior methods on Adroit tasks. Remarkably, SQOG achieves the highest average score across all Maze2d and Adroit datasets, underscoring its superior performance. Additional results further corroborate SQOG's efficacy across diverse dataset types, thus attesting to its robust generalization capabilities across varied environments. Moreover, SQOG demonstrates consistent and low runtime on Maze2d and Adroit tasks, with an average runtime of 0.4 hours. Therefore, we believe that its strong performance and computational efficiency make SQOG a noteworthy contribution to the offline RL community.

## B.2 WIDER EVIDENCE ON Q-VALUE ESTIMATION OF SQOG

We have presented an intuitive sanity check to demonstrate that SQOG alleviates the over-constraint issue and achieves more accurate $Q$-value estimation. This sanity check is conducted on the Inverted Double Pendulum task, which features a one-dimensional action space and a suitable level of task complexity. To further validate SQOG's effectiveness, we perform additional experiments on locomotion tasks to show that SQOG maintains accurate $Q$-value estimation in higher-dimensional environments.

The results, shown in Figure 5, are obtained by estimating true $Q$-values using the Monte Carlo method, similar to the sanity check. The critic $Q$-values are computed using SQOG's critic network, and both sets of $Q$-values are normalized for ease of comparison. We compute the difference between these normalized $Q$-values. Across all datasets, the value difference consistently decreases over

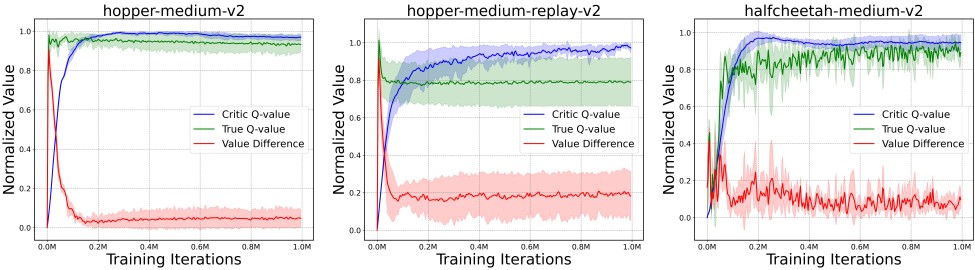

Figure 5: $Q$-value difference between critic $Q$-value and true $Q$-value on hopper-medium-v2, hopper-medium-replay-v2, halfcheetah-medium-v2. The experiments are run for 1M gradient steps over 4 random seeds.

training iterations, indicating that SQOG achieves accurate $Q$-value estimation. Notably, SQOG demonstrates the most stable value estimation on the hopper-medium-v2 dataset, aligning with its stable, SOTA performance on the same task.

## B.3    TRADE-OFF BETWEEN GENERALIZATION AND CONSERVATIVE BEHAVIORS

The trade-off between generalization and conservative behaviors is primarily influenced by two hyperparameters: $\beta$ and $\alpha$. $\beta$ controls the significance of the OOD generalization term in the critic loss, as demonstrated in our ablation study. $\alpha$ governs the relative intensity of the behavioral cloning penalty in the actor loss. Smaller values of $\alpha$ encourage more conservative behaviors. To investigate the impact of $\alpha$, we conduct a series of experiments with varying $\alpha$ values, fixing $\beta = 0.5$. The results are summarized in Table 3.

Table 3: Normalized average score of SQOG over different choices of different $\alpha$ values (with $\beta = 0.5$) on Mujoco "-v2". The results are averaged over 4 different random seeds. We bold the highest scores.

| Dataset | $\alpha = 300$ | $\alpha = 200$ | $\alpha = 150$ | $\alpha = 100$ | $\alpha = 50$ |
|---|---|---|---|---|---|
| halfcheetah-medium | **59.3±1.2** | 57.7±1.6 | 59.2±2.4 | 54.1±1.0 | 51.1±0.4 |
| hopper-medium | 70.9±7.7 | 91.6±6.0 | **100.6±0.7** | 94.1±6.8 | 74.9±4.1 |
| walker2d-medium | 83.6±7.5 | **88.2±1.8** | 82.9±0.8 | 81.7±0.8 | 81.1±0.3 |
| halfcheetah-medium-replay | 41.4±0.6 | 42.5±2.3 | **46.4±1.2** | 37.8±1.6 | 36.1±0.5 |
| hopper-medium-replay | 74.1±5.5 | 86.9±8.5 | **100.9±5.1** | 84.0±9.4 | 82.8±9.9 |
| walker2d-medium-replay | 39.0±12.3 | 42.4±7.4 | **88.3±3.5** | 73.1±8.0 | 68.5±5.3 |
| Mujoco Average | 61.4 | 68.2 | 79.7 | 70.8 | 65.8 |

On datasets like halfcheetah-medium and walker2d-medium, larger $\alpha$ values (e.g., $\alpha = 300$ or $\alpha = 200$) tend to perform better, suggesting that these tasks benefit from more aggressive generalization. The dynamics of these environments may not require highly conservative strategies, allowing the agent to explore beyond the dataset effectively. Across tasks, the results indicate that a moderate value of $\alpha = 150$ provides a strong balance between generalization and conservative behaviors. This balance likely ensures sufficient exploration without risking the instability of over-generalization, making it a robust choice across diverse scenarios.

## B.4    ADDITIONAL EXPERIMENTS ON NOISE SCALE AND CLIP

To further study the effects of the noise scale and clipping range, we conduct additional experiments by systematically varying the scale and clipping parameters to observe their influence on performance across multiple datasets. We present the results in Table 4.

Across most datasets, a scale of 0.6 and clip of 0.5 consistently achieves strong performance, suggesting that moderate noise levels effectively balance exploration and stability. While noise promotes generalization, excessive noise can lead to sampling outside the CHN boundary, resulting

Table 4: Normalized average score of SQOG over different choices of Gaussian noise scale and clip on Mujoco "-v2" and Adroit "-v1" datasets. The results are averaged over 4 different random seeds. We bold the highest scores.

| Dataset | scale=0, clip=0 | scale=0.2, clip=0.3 | scale=0.6, clip=0.5 | scale=1.0, clip=0.7 | scale=2.0, clip=1.0 |
|---|---|---|---|---|---|
| halfcheetah-medium | 51.2±3.9 | **60.6±0.5** | 59.2±2.4 | 53.1±1.9 | 51.7±1.5 |
| hopper-medium | 1.5±0.3 | 67.2±1.6 | **100.6±0.7** | 94.5±3.6 | 79.5±10.3 |
| walker2d-medium | 48.3±1.2 | 58.9±2.3 | 82.9±0.8 | **86.0±1.0** | 82.5±1.1 |
| halfcheetah-medium-replay | **49.2±0.3** | 47.8±2.2 | 46.4±1.2 | 36.6±1.1 | 35.8±0.8 |
| hopper-medium-replay | 22.3±6.6 | 19.4±6.8 | **100.9±5.1** | 24.3±2.7 | 27.1±6.7 |
| walker2d-medium-replay | 12.2±5.5 | 56.8±2.0 | 88.3±3.5 | **90.1±3.6** | 60.6±6.9 |
| Mujoco Average | 30.8 | 51.8 | **79.7** | 64.1 | 56.2 |
| pen-human | 23.3±6.7 | 46.7±4.8 | **80.0±4.7** | 64.9±5.5 | 55.8±5.9 |
| pen-cloned | 9.5±1.9 | 41.2±7.6 | **66.7±3.4** | 43.2±5.1 | 42.3±7.0 |
| Adroit Average | 16.4 | 44.0 | **73.4** | 54.1 | 49.1 |

in OOD $Q$-values that violate *safety guarantees* and degrade performance. Properly scaled noise ensures effective learning while respecting safety constraints.

The sensitivity to the noise distribution parameters (scale and clip) varies across datasets. In halfcheetah-medium-replay, the baseline configuration (scale=0, clip=0) yields the best performance, indicating that the effect of noise on in-sample $Q$-value estimation cannot be ignored. The dataset may already provide sufficient diversity, and adding noise introduces harmful uncertainty, leading to performance degradation. For such datasets, it is crucial to keep noise parameters conservative to avoid disrupting in-sample $Q$-learning and maintain stable performance. In contrast, for halfcheetah-medium, the optimal configuration is (scale=0.2, clip=0.3), and for walker2d-medium and walker2d-medium-replay, (scale=1.0, clip=0.7) achieves the best results. These differences highlight that there is room for improvement in *fine-tuning noise parameters* across various datasets. Combined with the results of experiments on the hyperparameter $\alpha$, we find that larger noise clipping and larger $\alpha$ achieve better performance in the walker2d-medium dataset. This implies that the task benefits more from $Q$-value generalization and aggressive policy extrapolation within the CHN.

The baseline configuration (scale=0, clip=0) significantly underperforms in most datasets, underscoring the necessity of noise injection to enhance exploration and overall performance. Noise injection is essential for improving the $Q$-function's ability to generalize to previously unexplored regions and optimize learning.

## B.5 Additional experiments on noise type

To better analyze the performance differences among noise types, we conducted additional experiments using three different noise settings: normal noise with a scale of 0.6 and clip [-0.5, 0.5], normal noise scaled through a tanh transformation to [-0.5, 0.5], and uniform noise within [-0.5, 0.5]. Results across 4 random seeds are presented in Table 5.

From Table 5, it is evident that uniform noise significantly underperforms on Mujoco datasets, with higher variance compared to normal noise settings. This is likely due to uniform noise generating overly random OOD samples that are *distributed sparsely and equally* across the entire range of [-0.5, 0.5]. While this ensures coverage across the OOD region, it fails to focus sufficiently on some key areas that are critical for $Q$-learning. Consequently, uniform noise may lead to insufficient training in each region, resulting in unstable and inconsistent performance.

Normal noise (both scaled+clip and tanh-transformed) performs better due to its concentrated sampling behavior. For normal+scale+clip noise, samples are densely concentrated at the clip boundaries (e.g., -0.5, 0.5). For normal+tanh noise, most samples are concentrated near the boundaries (e.g., -0.5, 0.5). This boundary-focused sampling behavior is likely to provide relatively adequate training samples and encourage the generalization in some critical OOD regions.

Table 5: Normalized average score of SQOG with different noise types on Mujoco "-v2" and Adroit "-v1" datasets. The results are averaged over 4 different random seeds. We bold the highest scores.

| Dataset | normal+0.6 scale | normal+tanh | uniform |
|---|---|---|---|
| halfcheetah-medium | 59.2±2.4 | **59.3±0.4** | 47.4±11.1 |
| hopper-medium | **100.6±0.7** | 89.7±7.4 | 62.5±36.5 |
| walker2d-medium | **82.9±0.8** | 82.8±3.2 | 65.5±17.9 |
| halfcheetah-medium-replay | **46.4±1.2** | 39.9±1.6 | 40.1±1.2 |
| hopper-medium-replay | **100.9±5.1** | 94.5±5.9 | 37.9±38.3 |
| walker2d-medium-replay | **88.3±3.5** | 62.5±17.4 | 46.8±20.4 |
| Mujoco Average | **79.7** | 71.5 | 49.9 |
| pen-human | **80.0±4.7** | 75.3±5.8 | 75.4±4.1 |
| pen-cloned | **66.7±3.4** | 64.5±8.1 | 61.9±4.7 |
| Adroit Average | **73.4** | 69.9 | 68.7 |

On Adroit datasets, differences between noise types are less pronounced, suggesting that the clip range [-0.5, 0.5] plays a more critical role than the specific noise type. Within this range, all noise types perform reasonably well.

Our observations suggest that the poor performance of uniform noise may result from its overly sparse and evenly spread sampling, which appears to limit its ability to provide sufficient coverage of some critical OOD regions for $Q$-value generalization. In contrast, normal noise with clipping or tanh transformations demonstrates potential advantages due to its boundary-focused sampling, which may facilitate more sufficient learning in those critical OOD regions. While our experiments demonstrate the superiority of normal noise with clipping in this context, we acknowledge that noise type and its influence on OOD action sampling is an important topic deserving deeper exploration. Our primary contribution lies in introducing SQOG, which effectively addresses over-constraint issues in offline RL by improving OOD $Q$-value generalization, delivering superior performance and computational efficiency. In this work, we focused on empirically validating the feasibility of normal noise (with clipping) and conducted preliminary analyses on the effects of noise type and range.

As future work, we plan to conduct a more systematic investigation into the role of noise in OOD sampling and $Q$-function generalization, aiming to establish a clearer theoretical understanding of its impact.

### B.6 ADDITIONAL EVALUATIONS WITH MORE SEEDS

We extended the evaluation of SQOG on Mujoco "-v2" datasets by running it with another 4 random seeds, bringing the total to 8. We believe this is a sufficiently robust number for reliable evaluation. The summarized results are presented in Table 6, where SQOG demonstrates consistent and reliable performance across different settings.

### B.7 COMPUTE INFRASTRUCTURE

In Table 7, we list the compute infrastructure that we use to run all of the baseline algorithms and SQOG experiments.

### B.8 EXPERIMENTAL DETAILS

**The true $Q$-values[7] in sanity check.** In sanity check, we calculate the true $Q$-values using a Monte Carlo estimation method to ensure accuracy. Specifically, we reset the environment to a given state $s$ and execute the action $a$. Starting from $(s, a)$, we simulate full trajectories and calculate the discounted return for each trajectory. To approximate the expected return, we repeat this process for

---

[7]In this context, the term "true $Q$-value" refers to the ground truth $Q$-values computed through Monte Carlo estimation. These values serve as a reference standard for evaluating the accuracy of the predicted $Q$-values generated by the critic networks.

Table 6: Normalized average score of SQOG on Mujoco "-v2" datasets. The results are averaged over 4 and 8 different random seeds

| Dataset | SQOG (4 seeds) | SQOG (8 seeds) |
|---|---|---|
| halfcheetah-random | $25.6 \pm 0.4$ | $25.7 \pm 1.8$ |
| hopper-random | $15.6 \pm 3.3$ | $15.4 \pm 4.1$ |
| walker2d-random | $17.7 \pm 3.5$ | $16.9 \pm 3.1$ |
| halfcheetah-medium | $59.2 \pm 2.4$ | $60.1 \pm 3.0$ |
| hopper-medium | $100.6 \pm 0.7$ | $100.8 \pm 0.6$ |
| walker2d-medium | $82.9 \pm 0.8$ | $82.3 \pm 1.3$ |
| halfcheetah-medium-replay | $46.4 \pm 1.2$ | $45.8 \pm 2.2$ |
| hopper-medium-replay | $100.9 \pm 5.1$ | $99.5 \pm 6.8$ |
| walker2d-medium-replay | $88.3 \pm 3.5$ | $86.5 \pm 5.8$ |
| halfcheetah-medium-expert | $92.6 \pm 0.4$ | $91.6 \pm 1.4$ |
| hopper-medium-expert | $109.2 \pm 2.8$ | $110.1 \pm 2.5$ |
| walker2d-medium-expert | $109.0 \pm 0.3$ | $109.1 \pm 0.3$ |
| Mujoco Average | 70.7 | 70.3 |

Table 7: Compute infrastructure.

| CPU | GPU | Memory |
|---|---|---|
| Intel(R) Xeon(R) CPU E5-2698 | Tesla V100-DGXS-32GB $\times$ 8 | 251G |
| Intel(R) Xeon(R) Silver 4216 | GPU GeForce RTX 3090 | 62G |

1000 sampled trajectories and take the average. We compute the $Q$-values for every 0.01 increment within the action range [-1.0, 1.0] and smooth the values using cubic spline interpolation. Finally, we normalize the values to keep them between 0 and 1 by multiplying by an appropriate constant. Given the computational intensity and rigorous sampling, this process provides a robust approximation of the true $Q$-values.

The true $Q$-function in the Mujoco environment could be irregular or even stepwise. However, we note that the smooth appearance of the $Q$-values in Figure 1 arises from the use of cubic spline interpolation applied to densely sampled data points. The interpolation is used solely for visual clarity and does not affect the underlying accuracy of the $Q$-value computation. Without this interpolation, the individual sampled points would still demonstrate the high accuracy of our SQOG method in estimating $Q$-values while effectively alleviating the issue of over-constraint.

**D4RL benchmarks.** In the main paper, we evaluate SQOG on the D4RL Gym-Mujoco task (Fu et al., 2021), which contains three environments (halfcheetah, hopper, walker2d), and five types of datasets (random, medium, medium-replay, medium-expert, expert). Random datasets are gathered by a random policy. Medium is generated by first training a policy online using Soft Actor-Critic (Haarnoja et al., 2018), early-stopping the training, and collecting 1M samples from this partially-trained policy. Medium-replay datasets are collected during the training process of the "medium" SAC policy. Medium-expert datasets are formed by combining the suboptimal samples and the expert samples. Expert datasets are made up of expert trajectories.

D4RL offers a metric called normalized score to evaluate the performance of the offline RL algorithm, which is calculated by:

$$\text{normalized score} = 100 * \frac{\text{score} - \text{random score}}{\text{expert score} - \text{random score}} \tag{31}$$

If the normalized score equals to 0, that indicates that the learned policy has a similar performance as the random policy, while 100 corresponds to an expert policy. The final results are obtained using a standard evaluation procedure. Specifically, for each experiment, we calculate the average normalized score over the last ten evaluation episodes. We then compute the mean and standard deviation of these scores across different random seeds.

Table 8: Experimental hyperparameters of SQOG.

| Hyperparameter | Value |
|---|---|
| Optimizer | Adam (Kingma & Ba, 2015) |
| Actor network learning rate | $3 \times 10^{-4}$ |
| Critic network learning rate | $3 \times 10^{-4}$ |
| Discount factor $\gamma$ | 0.99 |
| Total training step $T$ | $1 \times 10^6$ |
| Policy noise clipping parameter | 0.5 |
| Mini-batch size | 256 |
| Target network smoothing parameter $\tau$ | 0.005 |
| Actor network update frequency $m$ | 2 |
| $\alpha$ (TD3+BC (Fujimoto & Gu, 2021) Hyperparameter) | 150 (25 for Adroit tasks) |
| $\beta$ | 0.5 (2.5 for Adroit tasks) |
| Noise type | Normal distribution |
| Noise scale | 0.6 |
| Noise clip | 0.5 |

**Hyperparameters.** All experimental hyperparameters of SQOG are documented in Table 8. The hyperparameter $\alpha$ of TD3+BC governs the efficacy of the behavior cloning constraint, while in our algorithm, we introduce $\beta$ to regulate the strength of OOD generalization. Notably, a trade-off exists between $\alpha$ and $\beta$, where a larger $\beta$ corresponds to stronger generalization, while a smaller $\alpha$ implies more stringent constraints. Through empirical exploration, we identify two sets of hyperparameters that yield favorable performance: (150, 0.5) for Mujoco (excluding halfcheetah-medium-replay) and Maze2d tasks, and (25, 2.5) for Adroit tasks and halfcheetah-medium-replay in Mujoco. The Gaussian distribution is a good choice for dataset noise in OOD sampling, but there could be better alternatives such as pink noise (Eberhard et al., 2023), which is left for future work. We did not finely tune the noise scale and noise clip parameters, leaving room for improvement in their optimization.

## C ADDITIONAL DISCUSSION ON CHN

**Why the *state* space needs to be included in the CHN definition?** We study the $Q$-function in this paper, which is also called the state-action value function. The state is necessary when discussing OOD actions, as the state-action pair defines the context in which the action is taken.

An alternative definition of the CHN can be given as follows:

**Definition 3.** *Given state $s$ in the dataset $\mathcal{D}$, let $Conv_s(\mathcal{D})$ denotes the convex hull of the in-sample actions corresponding to the state $s$, $Conv_s(\mathcal{D}) = \{\sum_{i=1}^{n} \lambda_i a_i | \lambda_i \geq 0, \sum_{i=1}^{n} \lambda_i = 1, (s, a_i) \in \mathcal{D}\}$. Define the external neighborhood of the convex hull as: $N_s(Conv_s(\mathcal{D})) = \{a \in A \mid a \notin Conv_s(\mathcal{D}), \min_{a' \in Conv_s(\mathcal{D})} \|a' - a\| \leq r_s\}$, where $r_s$ is a radius chosen to be less than or equal to the diameter of $Conv_s(\mathcal{D})$. Finally, the CHN can be defined as: $CHN_{action}(\mathcal{D}) = \cup_{s \in \mathcal{D}} CHN_s(\mathcal{D})$, where $CHN_s(\mathcal{D}) = Conv_s(\mathcal{D}) \cup N_s(Conv_s(\mathcal{D}))$.*

While the alternative definition provided above introduces more detailed structures, it is less elegant and concise compared to the definition of the CHN in the main text. Furthermore, the alternative definition may lack connectivity, as the $CHN_s$ for different states may be disjoint, failing to form a cohesive and unified region. Importantly, the alternative Definition 3 is encompassed within the broader Definition 1 provided in the main text. (Given state $s \in \mathcal{D}$, $\forall a \in A$, if $a \in CHN_s(\mathcal{D})$, then $(s, a) \in CHN(\mathcal{D})$.) The original definition encompasses a broader region and enables us to offer more comprehensive safety guarantees due to its more general formulation.

## D CONNECTION BETWEEN THEORY AND PRACTICE

In theory, we introduce the CHN and the Smooth Bellman Operator (SBO). In practice, we design SQOG based on the insights from CHN and SBO. The connection between SBO and SQOG's critic loss is summarized in Table 9. For simplicity, we use MSE (mean squared error) to represent the

loss and omit the expectation operator for readability. The complete definition of SBO is provided in Definition 2, while the detailed formulation of SQOG's critic loss can be found in Eq. (13) and (14).

Table 9: Connections between SBO and SQOG's critic loss.

| Connections | In-sample action ($a$) | OOD action within CHN ($a^{ood}$) |
|---|---|---|
| SBO | $Q(s,a) \leftarrow r + \gamma \mathbb{E}_{a' \sim \pi(\cdot|\mathbf{s}')}[Q(s',a')]$ | $Q(s,a^{ood}) \leftarrow Q(s,a^{in}_{neighbor})$ |
| Loss | $\mathrm{MSE}\left(Q_\theta(s,a), \left(r + \gamma \hat{Q}_{\theta'}(s', \pi_\phi(s'))\right)\right)$ | $\mathrm{MSE}\left(Q_\theta(s,a^{ood}), Q(s,a^{in}_{neighbor})\right)$ |

For the $Q$-values of in-sample actions, SBO applies the base Bellman operator $\hat{B}_2^\pi$. Similarly, in practice, SQOG samples batches of data from the dataset, and the first term of its critic loss updates the $Q$-values of in-sample actions using an empirical Bellman operator. This closely aligns with the in-sample component of SBO.

For the $Q$-values of OOD actions within the CHN, SBO uses the smooth generalization operator $\mathcal{G}_1$ to generalize these OOD actions to their in-sample neighbors. Correspondingly, in practice, SQOG actively generates OOD actions from in-sample actions and trains their $Q$-values by leveraging the $Q$-values of their neighboring in-sample actions. This approach aligns well with the OOD component of SBO, enabling smooth generalization for OOD $Q$-values.

However, there is a difference. In SBO, if the policy's action $\pi(s')$ is identified as OOD, $\mathcal{G}_1$ updates its $Q$-value, $Q(s', \pi(s'))$, based on the $Q$-value of a neighboring in-sample action, $Q(s', a^{in}_{neighbor})$. In practice, however, we leave $\hat{Q}_{\theta'}(s', \pi_\phi(s'))$ unchanged, as accurately determining whether an action is OOD is challenging. Nevertheless, as the OOD $Q$-values are iteratively trained through the OG loss term, we expect $\hat{Q}_{\theta'}(s', \pi_\phi(s'))$ to become increasingly accurate over time, mitigating the overestimation and underestimation issue of $\hat{Q}_{\theta'}(s', \pi_\phi(s'))$ as training progresses (empirically shown in Appendix B.2).

Additionally, the BC loss applied to the actor network keeps the learned policy not far from the behavior policy. This ensures that the actions $\pi_\phi(s')$ generated from $\pi_\phi$ are likely to remain close to the dataset. Consequently, $\hat{Q}_{\theta'}(s', \pi_\phi(s'))$ is less prone to overestimation, further stabilizing the learning process.

Finally, the role of CHN is critical in providing safety guarantees and guiding the selection of practicable OOD regions. Without CHN, it would be challenging to identify feasible OOD regions for generalization. Attempting to handle all OOD regions indiscriminately could result in degraded performance or instability. Similarly, without the theoretical analysis provided by SBO, it would be challenging to conceive the idea of detaching the gradient of in-sample $Q$-values, which is essential for constructing meaningful targets for OOD $Q$-value updates.

## E    THE RELATIONSHIP BETWEEN SBO AND BC LOSS

Our main contribution lies in alleviating the over-constraint issue by generalizing $Q$-function estimation to OOD regions within the CHN. While "pessimism" or "constraint" is critical for policy improvement in offline RL, excessive constraints can lead to inaccurate $Q$-value estimation and suboptimal performance. SQOG mitigates this issue by enabling more accurate $Q$-value estimation, which directly benefits the policy improvement step.

To demonstrate this, we adopt TD3+BC as a baseline. However, we emphasize that SQOG's effectiveness is not limited to this specific framework or the use of behavior cloning (BC) loss.

**Theoretical justification.** As shown in Theorem 1, SBO approximates accurate OOD $Q$-values within CHN under the assumption that the learned policy $\pi$ is close to the behavior policy $\mu$. This assumption is satisfied by any policy constraint offline RL algorithm that enforces $\max(\mathrm{KL}(\pi, \mu), \mathrm{KL}(\mu, \pi)) \leq \epsilon$. Therefore, the conclusion that SBO enables gradual approximation of true OOD $Q$-values within CHN is generalizable to other policy constraint methods beyond TD3+BC.

**Empirical validation.** To demonstrate SBO's generalizability, we replace the BC loss in TD3+BC with the KL divergence penalty as used in BRAC's actor loss (Wu et al., 2019). Since the true behavior

Table 10: Normalized average score comparison of BRAC+SBO against baseline method BRAC on Mujoco "-v2" and Adroit "-v1" datasets. The results are averaged over 4 different random seeds. We bold the highest scores.

| Dataset | BRAC | BRAC+SBO |
|---|---|---|
| halfcheetah-medium | 49.8±1.2 | **54.3±1.2** |
| hopper-medium | 3.6±3.1 | **90.9±2.9** |
| walker2d-medium | 7.8±8.1 | **85.6±4.3** |
| halfcheetah-medium-replay | 41.8±6.2 | **47.8±2.0** |
| hopper-medium-replay | 28.8±20.3 | **61.1±11.9** |
| walker2d-medium-replay | 8.5±3.0 | **67.6±11.0** |
| Mujoco Average | 23.4 | **67.9** |
| Improvement | - | **190.2%** |
| pen-human | 19.2±16.3 | **69.7±8.7** |
| pen-cloned | 28.4±23.4 | **69.0±14.8** |
| Adroit Average | 23.8 | **69.4** |
| Improvement | - | **191.6%** |

policy is inaccessible, we adopt behavior cloning (a common and simple method) to approximate it. We then re-implement BRAC and add SBO to it. Table 10 presents the performance of BRAC+SBO across various datasets. We observe a significant performance improvement when SBO is added to BRAC, confirming that SBO is a *versatile* plug-in for policy constraint methods.

Although BRAC+SBO performs well, its performance does not surpass that of SQOG. This is likely because behavior cloning only provides an approximate behavior policy, and inaccuracies in this approximation weaken the KL penalty's ability to ensure $\pi$ is close to $\mu$, which may hurt the effectiveness of SBO. While advanced methods like generative models could improve behavior policy estimation, they are computationally expensive. In contrast, TD3+BC with BC loss strikes a favorable balance between simplicity, computational efficiency, and effectiveness.

In summary, SBO is not restricted to TD3+BC or BC loss. Our empirical results demonstrate its potential to generalize across other policy constraint methods, addressing the over-constraint issue and achieving superior performance.

**Further discussion on the contribution to policy constraint methods.** Policy constraint methods, such as TD3+BC (Fujimoto & Gu, 2021), aim to align the learned policy with the behavior policy to mitigate $Q$-value overestimation. However, these methods often result in $Q$-value underestimation due to limited exploration beyond the behavior policy, leading to the neglect of OOD actions that might correspond to higher $Q$-values. In contrast, SBO mitigates both $Q$-value underestimation and overestimation, particularly in OOD regions, as proven in Theorem 3. This improvement allows the policy to make more informed decisions by considering OOD actions with higher $Q$-values, addressing a key limitation of policy constraint methods. This capability is the primary driver behind the observed experimental improvements.

Furthermore, addressing the offline RL community context, the over-constraint issue remains a significant challenge for policy constraint methods. Theoretical analysis and experiments often neglect the "value part" in critic network, as it risks conflicting with policy constraints in actor network. However, SBO serves as a valuable complement to policy constraint methods. On one hand, SBO overcomes this bottleneck by accurately learning OOD $Q$-values for policy improvement and exploration (proved in Theorem 3, shown in the sanity check and Appendix B.2). On the other hand, SBO is really smooth when integrating into those policy constraint methods, primarily because we treat policy constraints as a precondition (as demonstrated by the significant improvements in Tables 1, 2, and 10). We firmly believe that SBO (SQOG) is a valuable supplement, combining theoretical rigor, strong performance, and computational efficiency.

Finally, while prior value-based offline RL methods often treat OOD regions as inherently risky due to information deficiencies (Xu et al., 2023; Kumar et al., 2020; Kostrikov et al., 2021b). Methods like MCQ (Lyu et al., 2022) explore mild but enough conservatism for offline learning while not

harming generalization. Building on this direction, our work demonstrates the insight that extending $Q$-function generalization to OOD regions within CHN can be beneficial. We believe this approach will attract more attention to $Q$-value estimation in OOD regions and offer new insights for the offline RL community.

