# OpenReview forum: "Offline RL with Smooth OOD Generalization in Convex Hull and its Neighborhood"
_ICLR.cc/2025/Conference — ICLR 2025 Poster_

### Official Review · Reviewer_CHBA · 2024-10-28

**Soundness:** 3
**Presentation:** 2
**Contribution:** 2
**Rating:** 6
**Confidence:** 4

**Summary:**

This paper introduces a novel offline RL algorithm, Smooth Q-function OOD Generalization (SQOG), aimed at improving Q-value estimation by enhancing the generalization of the Q-function within the Convex Hull and its Neighborhood (CHN). The authors provide theoretical guarantees, showing that the proposed algorithm approximates true Q-values for both in-sample and out-of-distribution (OOD) actions. On the D4RL benchmarks, SQOG demonstrates superior performance compared to state-of-the-art methods.

**Strengths:**

The paper is well-organized. The proposed algorithm SQOG significantly reduces computational costs compared to existing algorithms.

**Weaknesses:**

- **Novelty:** The proposed SQOG method builds on the concepts of the Convex Hull and its Neighborhood (CHN) and the Smooth Bellman Operator (SBO). The use of CHN was first introduced in DOGE [1], and SBO for in-sample Q-value updates is a common practice in offline RL. These techniques are not new, which makes the contribution appear incremental. It would be better to clarify the difference or advantage of these techniques compared to existing methods.
- **Neighborhood**: While the introduction of the neighborhood is intended to alleviate the over-constraining issue of OOD actions, it is unclear how this concept is reflected in Theorem 1 and informs practical implementation. Additionally, although the authors theoretically demonstrate that SQOG can achieve good OOD evaluation, the underlying assumptions—such as the continuity of the Q-function—are idealistic. For example, in sparse reward environments like Montezuma's Revenge, where only a few actions yield rewards, the Q-function is likely to be discontinuous or stepwise.
- **Connection between theory and practice:** There appears to be a disconnect between the theoretical and experimental sections. The theory suggests using CHN and SBO, but even without these components, a researcher familiar with offline RL could still design an algorithm based on principles like pessimism and in-sample updates. The writing should better emphasize the unique advantages of CHN to make its value more apparent.
- **The quality of the dataset:** In my opinion, CHN essentially augments the data by adding state-action pairs near the original dataset, making the quality of the dataset crucial for performance. In high-dimensional state and action spaces, the main challenge is often limited sample size. It would help to discuss this issue further or use toy examples to illustrate the benefits of CHN.

[1] When data geometry meets deep function: Generalizing offline reinforcement learning. ICLR 2023.

**Questions:**

Q1: In Figure 1, I am curious about how to calculate the true Q values for action in the range $[0,1]$. Also, it would be more appropriate to refer to these as "ground truth" Q-values. Furthermore, the "true Q-values" are shown as a smooth line due to the use of neural network function approximation. However, as mentioned earlier, the true Q function (which is not precisely known in the MuJoCo environment) could be quite irregular or even stepwise.  This makes the toy example less convincing in demonstrating the accuracy of the approximation.

Q2: In the conclusion, the authors state that solving the CHN and identifying all OOD regions within it is both impractical and unnecessary. While I partially agree, I also hold a different viewpoint. The authors mention that one advantage of CHN is its safety guarantees, as measured by the neighborhood radius. I noticed that in the code, the noise is clipped within the range [−0.5,0.5]. Have the authors experimented with tuning this parameter? If so, providing more experimental details would be beneficial, as the clipped range directly affects the neighborhood radius associated with the safety guarantees.

---

> ### Author Response · Authors · 2024-11-18
> **Author Response to Reviewer CHBA (part 1/4)**
>
> Thanks for your inspiring and thoughtful comments, and thanks for commenting that our paper is "well-organized". We provide clarification to the concerns below. We hope our responses can address your concerns.
>
> **Q1: How to calculate the true Q values?**
>
> **A1:** We thank the reviewer for their thoughtful feedback on the calculation of the true Q-values and its implications for our toy example. We acknowledge the concern raised about the terminology and the potential mismatch between the idealized true Q-values and the actual Q-function in the MuJoCo environment.
>
> **In our study, we calculate the true Q-values using a Monte Carlo estimation method to ensure accuracy.** Specifically, we reset the environment to a given state $s$ and execute the action $a$. Starting from $(s,a)$, we simulate full trajectories and calculate the discounted return for each trajectory. To approximate the expected return, we repeat this process for 1000 sampled trajectories and take the average. We compute the $Q$-values for every 0.01 increment within the action range [-1.0, 1.0] and smooth the values using cubic spline interpolation. Finally, we normalize the values to keep them between 0 and 1 by multiplying by an appropriate constant. Given the computational intensity and rigorous sampling, this process provides a robust approximation of the true Q-values. We provide additional details about the computation of true Q-values in **Appendix B.5** for further clarification.
>
> We appreciate the observation that the true Q-function in the MuJoCo environment could be irregular or even stepwise. However, we note that the smooth appearance of the Q-values in Figure 1 arises from the use of cubic spline interpolation applied to densely sampled data points. The interpolation is used solely for visual clarity and does not affect the underlying accuracy of the Q-value computation. **Without this interpolation, the individual sampled points would still demonstrate the high accuracy of our SQOG method in estimating Q-values while effectively alleviating the issue of over-constraint.**
>
> The goal of the toy example is to provide an illustrative benchmark for evaluating the approximation accuracy of our SQOG method. While the MuJoCo environment's true Q-function may exhibit irregularities, the Monte Carlo-based ground truth Q-values used in our experiments are derived from actual environmental interactions, ensuring their validity. Furthermore, as highlighted by Reviewer etFc, "The sanity check on the Inverted Double Pendulum task empirically shows SQOG's effectiveness in alleviating the over-constraint issue, **which is a valuable addition**" and by Reviewer cDBr, "I find this is **a great illustration of the over-constraining problem**." These evaluations reaffirm that our toy example effectively demonstrates the unique strengths of SQOG in addressing over-constraint and accurately approximating Q-values.
>
> To address the reviewer’s concern,  we conduct additional experiments in **Appendix B.2**. We evaluated SQOG on MuJoCo locomotion tasks from the D4RL benchmark, including `hopper-medium-v2`, `hopper-medium-replay-v2`, and `halfcheetah-medium-v2`. In these experiments, we analyzed the value difference between the Q-values estimated by SQOG’s critic network and the true Q-values obtained via Monte Carlo estimation. The results demonstrate that across all datasets, the value difference consistently decreases over training iterations. This indicates that SQOG is capable of achieving accurate Q-value estimation, even in high-dimensional environments like locomotion tasks.

---

> ### Author Response · Authors · 2024-11-18
> **Author Response to Reviewer CHBA (part 2/4)**
>
> **Q2: More experimental results on noise scale and clip.**
>
> **A2:** We appreciate the reviewer’s insightful suggestion. To address this point, we conducted additional experiments by systematically varying the scale and clipping parameters to observe their influence on performance across multiple datasets. Below, we present the results.
>
> Table 1: Normalized average score of SQOG over different choices of gaussian noice scale and clip on MuJoCo "-v2" and Adroit "v1" datasets. The results are averaged over 4 different random seeds. We bold the highest scores.
>
> | Dataset           | scale=0, clip=0| scale=0.2, clip=0.3  |scale=0.6, clip=0.5  |scale=1.0, clip=0.7  | scale=2.0, clip=1.0|
> |--------------------|-------|-------|-------|-------|-------|
> | halfcheetah-medium   | 51.2$\pm$3.9  |  **60.6$\pm$0.5** | 59.2$\pm$2.4  |  53.1$\pm$1.9 | 51.7$\pm$1.5|
> | hopper-medium    | 1.5$\pm$0.3  |  67.2$\pm$1.6 | **100.6$\pm$0.7**  |  94.5$\pm$3.6 | 79.5$\pm$10.3|
> | walker2d-medium   | 48.3$\pm$1.2   |   58.9$\pm$2.3 | 82.9$\pm$0.8  |  **86.0$\pm$1.0** | 82.5$\pm$1.1|
> | halfcheetah-medium-replay    | **49.2$\pm$0.3**  |  47.8$\pm$2.2 |  46.4$\pm$1.2  |  36.6$\pm$1.1| 35.8$\pm$ 0.8|
> | hopper-medium-replay     | 22.3$\pm$6.6 | 19.4$\pm$6.8 | **100.9$\pm$5.1**  |  24.3$\pm$2.7 | 27.1$\pm$6.7 |
> | walker2d-medium-replay   | 12.2$\pm$5.5  | 56.8$\pm$2.0 | 88.3$\pm$3.5  |  **90.1$\pm$3.6** | 60.6$\pm$6.9 |
> | Mujoco Average |  30.8 | 51.8 | **79.7** | 64.1 | 56.2 |
> |pen-human| 23.3$\pm$6.7  | 46.7$\pm$4.8 | **80.0$\pm$4.7** |  64.9$\pm$5.5 | 55.8$\pm$5.9|
> |pen-cloned|  9.5$\pm$1.9  | 41.2$\pm$7.6 | **66.7$\pm$3.4** |  43.2$\pm$5.1 | 42.3$\pm$7.0|
> |Adroit Average |    16.4 |     44.0 |     **73.4** |    54.1 |  49.1 |
>
> From these results, we observe the following:
> 1. **Moderate Noise Balances Exploration and Stability:** Across most datasets, a scale of 0.6 and clip of 0.5 consistently achieves strong performance, suggesting that moderate noise levels effectively balance exploration and stability. While noise promotes generalization, excessive noise can lead to sampling outside the CHN boundary, resulting in OOD Q-values that violate **safety guarantees** and degrade performance. Properly scaled noise ensures effective learning while respecting safety constraints.
>
> 2. **Dataset-Specific Effects of Noise:**
> The sensitivity to the noise distribution parameters (scale and clip) varies across datasets.
> In halfcheetah-medium-replay, the baseline configuration (scale=0, clip=0) yields the best performance, indicating that the effect of noise on in-sample Q-value estimation cannot be ignored. The dataset may already provides sufficient diversity, and adding noise introduces harmful uncertainty, leading to performance degradation. For such datasets, it’s crucial to keep noise parameters conservative to avoid disrupting in-sample Q-learning and maintain stable performance.
> In contrast, for halfcheetah-medium, the optimal configuration is (scale=0.2, clip=0.3), and for walker2d-medium and walker2d-medium-replay, (scale=1.0, clip=0.7) achieves the best results. These differences highlight that there is room for improvement in fine-tuning noise parameters across various datasets.
>
> 3. **Effectiveness of Noise Injection:** The baseline configuration (scale=0, clip=0) significantly underperforms in most datasets, underscoring the necessity of noise injection to enhance exploration and overall performance. Noise injection is essential for improving the Q-function's ability to generalize to previously unexplored regions and optimize learning.

---

> ### Author Response · Authors · 2024-11-18
> **Author Response to Reviewer CHBA (part 3/4)**
>
> **Q3: Comments on the "Novelty" in weaknesses.**
>
> **A3:** We appreciate the reviewer’s comments on the novelty of our SQOG method and the opportunity to clarify its contributions compared to existing techniques. Below, we address the concerns raised regarding the use of CHN and SBO and highlight the distinct advantages of our approach:
>
> 1.**Difference from DOGE:** While DOGE first introduced the concept of CHN, its primary innovation lies in the design of the actor loss. In contrast, our method focuses on the critic loss by incorporating the Smooth Bellman Operator (SBO) to improve the generalization capability of the Q-function. This innovation directly addresses the pervasive over-constraint issue in policy-constraint algorithms, enabling more accurate Q-value estimation and better overall algorithm performance. Thus, the core contribution of SQOG—centered on the critic loss—is significantly different from DOGE, establishing it as a novel development in this context.
>
> 2.**Difference from In-Sample Learning Approaches (e.g., IQL)**: While SBO might appear similar to in-sample learning approaches like IQL (Kostrikov et al., 2021b), there are critical differences in their goals and implementations. IQL approximates the optimal in-sample Q-values for Q-learning, which are constrained by in-sample actions. In contrast, MQOG with SBO aims to **approximate the true Q-values for policy evaluation**, allowing for a broader exploration of the action space and alleviating over-constraining issues inherent to in-sample learning. Notably, as shown in Eq.(14), the critic loss function of SQOG explicitly incorporates in-sample $Q$ in the last term with gradient detachment, using it as a target for OOD $Q$. This design enables the learning of neighboring OOD $Q$-values, significantly enhancing generalization. This distinction is fundamental, as our approach enhances Q-function accuracy and achieves improved performance across challenging tasks.
>
> **Q4: Comments on the "Neighborhood" in weaknesses.**
>
> **A4:** We appreciate the reviewer’s comments on the neighborhood concept and its role in our theoretical framework. To clarify, Theorem 1 demonstrates that $\hat{Q}^{\pi}_{\theta}(s,a^{in})$ provides a near-accurate estimation for in-sample $(s,a^{in})$ (L207-209). Building on this, Proposition 3 shows that the neighboring in-sample $Q$-value serves as a appropriate target for OOD $Q$-value estimation. This connection highlights the importance of the neighborhood concept: the OOD action's neighboring action often lies within the Neighborhood of the convex hull. Therefore, the neighborhood is a critical component for addressing over-constraint issue.
>
> Regarding the assumption of continuity, we clarify that this assumption pertains to the **parametric deep $Q$-function**, not the true $Q$-function. As the reviewer correctly pointed out, the true $Q$-function in sparse reward environments, such as Montezuma's Revenge, is often discontinuous or stepwise. However, it is reasonable to assume continuity in the parametric deep $Q$-function, as this is common practice in related works (e.g., [1]).
>
> Finally, to further address the concern about sparse rewards, we note that SQOG achieves strong performance on tasks in the Adroit benchmark (Appendix B), which includes sparse reward environments as defined in D4RL [2].
>
> [1] Li, J. et al. When data geometry meets deep function: Generalizing offline reinforcement learning. arXiv preprint arXiv:2205.11027, 2022.
>
> [2] J. Fu. et al. D4RL: Datasets for deep data-driven reinforcement learning, 2021.

---

> ### Author Response · Authors · 2024-11-18
> **Author Response to Reviewer CHBA (part 4/4)**
>
> **Q5: Comments on the "Connection between theory and practice" in weaknesses.**
>
> **A5:** We appreciate the reviewer’s concerns regarding the connection between the theoretical and experimental sections. In Section 3.4 Practical Algorithm, the OG loss is designed to handle OOD generalization. It can be derived from ${\mathcal{G}\_1}$ in the SBO. Specifically, in ${\mathcal{G}\_1}$, we use the neighboring in-sample $Q$-values $Q(s,a^{in}\_{neighbor})$ to update the OOD $Q$-values. To achieve this, we introduce a Mean Squared Error (MSE) loss between the parametric OOD $Q$-values $Q\_{\theta}(s,a^{ood})$ and the neighboring in-sample $Q$-values $Q(s,a^{in}\_{neighbor})$.
>
> In practice, we carefully construct OOD actions by adding controlled noise to randomly sampled in-sample actions from the dataset. This subtle strategy ensures that the generated OOD actions are close to the in-sample region, enabling effective use of the neighboring in-sample $Q$-values as targets. This process aligns directly with the theoretical principles outlined in SBO, ensuring consistency between theory and implementation.
>
> Furthermore, the role of CHN is critical in providing safety guarantees and guiding the selection of practicable OOD regions. Without CHN, it would be challenging to **identify feasible OOD regions for generalization**. Attempting to handle all OOD regions indiscriminately could result in degraded performance or instability. Similarly, without the theoretical analysis provided by SBO, it would be challenging to conceive the idea of **detaching the gradient** of in-sample $Q$-values, which is essential for constructing meaningful targets for OOD $Q$-value updates.
>
> Unlike in-sample learning, where the goal is primarily to learn the optimal in-sample $Q$-values, our approach avoids being overly pessimistic. **Instead, we aim to learn accurate OOD $Q$-values to mitigate over-pessimism and over-constraint in OOD regions.** This deliberate design ensures that our method effectively balances generalization and safety. These components are not only unique to our method but also crucial for addressing the OOD generalization problem, highlighting the synergy between theory and practice in our approach.
>
> **Q6: Comments on the "The quality of the dataset" in weaknesses.**
>
> **A6:** Thank you for pointing out the importance of dataset quality and the challenges posed by limited sample sizes in high-dimensional tasks.
>
> To address your concern, we introduce the Adroit domain, which focuses on high-dimensional, sparse-reward robotic manipulation tasks based on human demonstrations. Unlike Gym MuJoCo tasks with 1e6 samples, the Adroit pen-human dataset contains only **5000** samples, and the pen-cloned task has **5e5** samples. Despite these limitations, SQOG outperforms baseline methods, achieving an average score of 80.0 in the pen-human dataset and maintaining strong performance in the larger pen-cloned dataset. These results demonstrate SQOG's ability to handle lower-quality datasets effectively, as you noted.
>
> Table 2: Normalized average score comparison of SQOG against baseline methods on Adroit “-v1” datasets over 4 random seeds. We bold the highest scores.
> | Dataset           | BC    | CQL   | BCQ   | TD3+BC        | IQL            | MCQ            | SQOG             |
> |--------------------|-------|-------|-------|---------------|----------------|----------------|------------------|
> | pen-human (5000 samples)         | 34.4  | 37.5  | 68.9  | 0.0±0.0       | 68.7±8.6       | 68.5±6.5       | **80.0±4.7**     |
> | pen-cloned  (5e5 samples)      | 56.9  | 39.2  | 44.0  | 0.0±0.0       | 35.3±7.3       | 49.4±4.3       | **66.7±3.4**     |
> | **Total**  | 91.3 | 76.7 | 112.9 | 0.0           | 104          | 117.9 | **146.7**        |
>
> We agree that limited sample sizes pose a significant challenge in high-dimensional offline RL tasks. Non-expert, sparse datasets make it difficult to estimate accurate in-sample $Q$-values, which, in turn, affects the accuracy of OOD $Q$-values. Future work will focus on addressing these challenges, including exploring methods to improve robustness and generalization under such conditions.
>
> Thank you for your valuable feedback, which will guide our continued efforts to enhance performance in high-dimensional, low-quality datasets.

---

> > ### Comment · Reviewer_CHBA · 2024-11-23
> >
> > I thank the authors for the detailed feedback and the additional experiments. I have two concerns as follows:
> >
> > 1.For A4:
> >  the authors clarify that the assumption of continuity pertains to the parametric deep Q-function, but the the continuity property in the main text involves $Q^{\pi}$ . This distinction is important, as the underlying requirement of this assumption is that the parametric deep Q-function can well approximate the true Q-function. However, this condition might not hold when the true $Q^{\pi}$ is discontinuous or stepwise, which challenges the validity of the assumption in such cases.
> >
> > 2.For A3 and A5： I appreciate the authors clarify how SQOG differs from DOGE and IQL by focusing on in-sample learning during policy evaluation. I agree the point that SQOG could  learn more accurate OOD $Q$-values neighboring the dataset to some extent. However, it’s crucial to recognize that the primary driver of improvement in Offline RL lies in the “pessimism” applied during policy improvement. From my perspective, the performance improvement directly relies on actor loss (15) from TB3+BC, which enforces pessimism during policy improvement. Given this, how the improvements in Q-function accuracy, limited to a finite domain, directly translate to benefits in the policy improvement step. Additionally, I am curious how the accuracy of  OOD Q-values is ensured outside the safety guarantee, especially since this guarantee is often small and challenging to quantify across diverse tasks. While the experimental results are impressive, further clarification is needed to determine whether the performance gains are heavily reliant on SQOG or other underlying factors. I will raise my score if the authors could my address my concerns on this point.

---

> > > ### Author Response · Authors · 2024-11-24
> > > **Author Response to Reviewer CHBA**
> > >
> > > Thank you for your thoughtful feedback. We provide clarifications below and hope our responses address your concerns.
> > >
> > > **Main Concern: Are the performance gains heavily reliant on SQOG or other underlying factors (e.g., TD3+BC)?**
> > >
> > > **Response 1:** Our main contribution lies in alleviating the *over-constraint* issue by generalizing $Q$-function estimation to out-of-distribution (OOD) regions within the CHN. While "pessimism" or "constraint" is critical for policy improvement in Offline RL, excessive constraints can lead to inaccurate Q-value estimation and suboptimal performance. SQOG mitigates this issue by enabling more accurate Q-value estimation, which directly benefits the policy improvement step and leads policy improvements.
> > >
> > > To demonstrate this, we adopt TD3+BC as a baseline. However, we emphasize that SQOG's improvements are **not** limited to this specific framework or the use of behavior cloning (BC) loss.
> > >
> > > **Theoretical Justification:**
> > > As shown in Theorem 1, SBO provides accurate OOD Q-value approximation within CHN under the assumption that the learned policy $\pi$ is close to the behavior policy $\mu$. This assumption is generally satisfied by any policy constraint Offline RL algorithm enforcing $\max(\mathrm{KL}(\pi, \mu), \mathrm{KL}(\mu, \pi)) \leq \epsilon$. Therefore, SQOG's improvement extends to other policy constraint methods beyond TD3+BC.
> > >
> > > **Empirical Validation:**
> > > To validate SQOG's generalizability, we replaced the BC loss in TD3+BC with the KL divergence penalty used in BRAC [1]. Since the true behavior policy is inaccessible, we approximated it via behavior cloning and implemented BRAC+SBO. Table 1 presents the performance of BRAC+SBO across various datasets.
> > >
> > > Table 1: Normalized average score comparison of BRAC+SBO against baseline method BRAC on MuJoCo "-v2" and Adroit "v1" datasets. The results are averaged over 4 different random seeds. We bold the highest scores.
> > > | Dataset         | BRAC       | BRAC+SBO   |
> > > |----------------------|----------------|-----------------|
> > > | halfcheetah-medium   | 49.8$\pm$1.2   | **54.3$\pm$1.2** |
> > > | hopper-medium        | 3.6$\pm$3.1    | **90.9$\pm$2.9** |
> > > | walker2d-medium      | 7.8$\pm$8.1    | **85.6$\pm$4.3** |
> > > | Mujoco Average      | 20.4       | **76.9**         |
> > > | **Improvement**      | —              | **277.0%**       |
> > > | pen-human            | 19.2$\pm$16.3  | **69.7$\pm$8.7** |
> > > | pen-cloned           | 28.4$\pm$23.4  | **69.0$\pm$14.8**|
> > > | Adroit Average         | 23.8       | **69.4**         |
> > > | **Improvement**      | —              | **191.6%**       |
> > >
> > > From Table 1, we observe a significant performance improvement when SBO is added to BRAC, confirming that SBO is a versatile **plug-in** for policy constraint methods.
> > >
> > > Although BRAC+SBO performs well, its performance does not surpass that of SQOG. This is likely because behavior cloning only provides an approximate behavior policy, and inaccuracies in this approximation weaken the KL penalty's ability to ensure $\pi$ is close to $\mu$, which may hurt the effectiveness of SBO. While advanced methods like generative models could improve behavior policy estimation, they are computationally expensive.
> > >
> > > In summary, SBO is not restricted to TD3+BC or BC loss. Our empirical results demonstrate its potential to generalize across other policy constraint methods, addressing the over-constraint issue and achieving superior performance. We appreciate your constructive feedback and are glad to hear that addressing these concerns could improve your evaluation of our work. We hope these responses adequately address your concerns and demonstrate the robustness of our approach. Please feel free to let us know if further clarification is needed.
> > >
> > > [1] Yifan Wu, George Tucker, and Ofir Nachum. Behavior Regularized Offline Reinforcement Learning.
> > >
> > > **Concern: How the accuracy of OOD Q-values is ensured outside the safety guarantee, especially since this guarantee is often small and challenging to quantify across diverse tasks.**
> > >
> > > **Response 2:**
> > > We acknowledge that it is challenging to directly quantify the accuracy of OOD Q-values beyond the safety guarantee. As discussed in the last paragraph of Sec 3.2, SBO is designed specifically to handle OOD Q-values within the CHN, adhering to the safety guarantees. Extending this approach to learn Q-values for faraway OOD actions, which fall outside the CHN, remains an open challenge in the field.
> > >
> > > However, we argue that the safety guarantee provided by SBO is not necessarily small. For example, in our experiments on noise clipping, a clip value of `0.5` consistently achieves strong performance across multiple datasets. Considering that the action space in these datasets is `[-1,1]`, the chosen clip size represents a significant portion of the action range, rather than a minimal constraint. This demonstrates that SBO's safety guarantee is sufficient to capture meaningful OOD Q-values within the CHN, enabling robust performance.

---

> > > ### Author Response · Authors · 2024-11-24
> > > **Author Response to Reviewer CHBA**
> > >
> > > **Concern: The assumption of continuity pertains to the parametric deep Q-function, but its validity may be challenged when the true Q-function is discontinuous or stepwise. This could impact the learning of the deep Q-function and the validity of the assumption in such cases.**
> > >
> > > **Response 3:**  We appreciate your insightful observation and agree that when the true Q-function exhibits severe discontinuity or stepwise characteristics, it could hinder the learning process of the parametric deep Q-function. Below, we address this concern in detail:
> > >
> > > Our continuity assumption ensures that the difference between the OOD Q-values and their in-sample neighboring Q-values remains small. If this gap becomes excessively large, the accuracy of the SBO operator could indeed be compromised. This represents a limitation of our approach, as SBO relies on the assumption of smooth transitions between in-sample and OOD regions.
> > >
> > > While extreme discontinuities pose challenges, we argue that minor discontinuities in the true Q-function are unlikely to significantly impact the effectiveness of SBO. As long as the difference between OOD Q-values and in-sample neighboring Q-values remains manageable, our update mechanism remains valid. Thus, in theory, the continuity assumption can be **relaxed** to accommodate slight non-smoothness in the true Q-function.
> > >
> > > The experiments on sparse reward datasets, such as `pen-human-v1` and `pen-cloned-v1` in Adroit, provide empirical evidence supporting the robustness of SBO. These datasets inherently involve challenges due to sparse and potentially non-smooth rewards, yet SBO consistently achieves strong performance. This suggests that our method is effective under conditions that are not excessively discontinuous or stepwise. However, we acknowledge that extreme discontinuities or stepwise transitions in the true Q-function not only challenge SBO but also impede the learning of neural networks in general, making most methods ineffective in such cases.
> > >
> > > While the continuity assumption is crucial to the theoretical foundation of SBO, it can tolerate slight deviations in practice, as demonstrated by both theoretical reasoning and empirical results. Nonetheless, extreme cases of discontinuity remain an open challenge, not only for SBO but also for deep reinforcement learning methods as a whole. Addressing these extreme cases is an important direction for future research, and we are committed to exploring strategies to enhance the robustness of SBO and similar methods under such challenging conditions.

---

> ### Comment · Reviewer_CHBA · 2024-11-25
>
> Thank you for the detailed response.  The empirical results of BRAC+SBO is promising. However, this doesn't fully remove my confusion. **The authors seems have not clearly explain the source of the observed experimental improvements.** In my point, SBO is a technique designed to enhance "policy-constrained" methods like BRAC, BEAR, and TD+BC. SBO is a "value constraint" method similar to CQL , with CHN acting as a trade-off mechanism that mitigates over-pessimism across the entire OOD regions (CQL suppresses Q value in OOD region, while SBO preserves Q value consistent within CHN region ).
>
> Thus, the radius of CHN is a pivotal component in this setup, serving as the main parameter to control the degree of pessimism. Since the radius is treated as a tunable parameter in experiments, it directly influences the extent of pessimism. Additionally, policy constraints, such as BC loss or KL loss, complement SBO **by providing further pessimism outside the CHN region**. Without these policy constraints, SBO's effectiveness is likely to be significantly reduced.
>
>
>
> In fact, the benefits of tuning the CHN radius in experiments **do not seem closely tied to the theoretical definition of** whether the radius should be large or small. Instead, they seem to stem from an empirical balance—avoiding excessive pessimism across OOD regions.  Despite this, the experimental improvements is promising. I will reconsider the score after further discussions in the next period.

---

> > ### Author Response · Authors · 2024-11-26
> > **Author Response to Reviewer CHBA**
> >
> > Thank you for your insightful feedback and for taking the time to evaluate our work. We deeply appreciate your recognition of the empirical results and would like to address your remaining concerns in detail.
> >
> > **Concern 1: Further explanation on the source of the observed experimental improvements. The contribution of the SBO relies on pessimistic constraints.**
> >
> > **A1:**
> > We appreciate the reviewer’s detailed observation and clarification. To address the concern, we emphasize that SBO should be understood as a **"value generalization"** rather than a "value constraint" method. This distinction is crucial. While policy-constrained methods like TD3+BC focus on aligning the learned policy $\pi$ with the behavior policy $\mu$ to mitigate Q-value overestimation, they inadvertently lead to Q-value underestimation, particularly due to the limited exploration beyond $\mu$ (as noted in L216-218). Consequently, these methods are prone to **ignore OOD actions that may correspond to higher Q-values.**
> >
> > In contrast, SBO operates differently. As proven in Theorem 3, SBO enhances the accuracy of both underestimation and overestimation in Q-values, especially in OOD regions. This improvement enables the policy to **make more informed decisions by considering OOD actions with higher Q-values**, addressing a fundamental limitation of policy-constrained methods. This capability serves as the **primary source** of the observed experimental improvements.
> > While explaining task-specific results in detail is impractical and outside our contribution's scope, our theoretical analysis and empirical validations (e.g., sanity check and Appendix B.2) consistently demonstrate SBO's efficacy in improving OOD Q-value generalization.
> >
> > Furthermore, addressing the offline RL community context, over-constraint issue remains a significant challenge for policy constraint methods.
> > Theoretical analysis and experiments often neglect the "value part", as it risks **conflicting** with policy constraints (we have tried to address this issue many times). However, SBO serves as a valuable complement to policy constraint methods. On one hand, SBO overcomes this bottleneck by accurately learning OOD Q-values for policy improvement and exploration (proved in Theorem 3, shown in sanity check and Appendix B.2). On the other hand, SBO is really **smooth** when integrating into those policy constraint methods, primarily because we treat policy constraints as a precondition (as demonstrated by the significant improvements in Tables 1, 2, and 5). We firmly believe that SBO (SQOG) is a valuable supplement, combining **theoretical rigor**, **strong performance**, and **computational efficiency**.
> >
> > Lastly, as highlighted in the Introduction, previous value-based offline RL methods often view OOD regions as inherently risky due to information deficiencies (e.g., [1], CQL, IQL). However, our work reveals that extending Q-function generalization to OOD regions within CHN can be **beneficial**. We believe that our work will draw greater attention to Q-value estimation in OOD regions and provide new insights for the offline RL community.
> >
> > [1] H. Xu et al., Offline RL with no OOD actions: In-sample learning via implicit value regularization. ICLR 2023.
> >
> > **Concern 2: A gap between the theoretical CHN radius and the empirical noise clip.**
> >
> > **A2:**
> > We acknowledge the reviewer’s concern and offer the following clarification: It is inherently impractical to define a precise CHN radius across datasets due to the varying convex hull structures. As demonstrated in Proposition 1, when OOD actions are proximal to in-sample actions, the Q-value difference is very small. This proximity allows for better generalization, reflected in Proposition 3 and Theorem 3, where closer OOD actions correspond to smaller $\varepsilon$. In practice, we leverage this insight by using normal noise to conservatively train closer OOD Q-values with high confidence.
> >
> > **Moreover, we did not require extensive effort in parameter fine-tuning.** Instead, we provided a moderate scale and clip choice of `[0.6,0.5]`, which has demonstrated stability across various datasets. This moderate choice effectively promotes OOD value generalization while minimizing side effects on in-sample value learning.
> >
> > In conclusion, while direct theoretical proofs regarding the optimal CHN radius are challenging due to practical complexities, our theoretical foundation demonstrates that closer OOD Q-values are inherently easier to learn. However, we believe that slight hyperparameter fine-tuning is acceptable, and we have already identified **a stable and moderate choice** suitable for the current application.
> >
> > We hope that the clarifications provided help address your concerns and further highlight the contributions of our work. We kindly request that you reconsider the score in light of these points, and we truly appreciate your thoughtful review and consideration.

---

> > > ### Comment · Reviewer_CHBA · 2024-11-26
> > >
> > > I appreciate the great efforts of the authors to enhance the quality of this manuscript. The authors have clarified most of my concerns. Although I still maintain that there is a gap between the theoretical analysis and the implementation, the **strong performance**, and **computational efficiency** of SQOG make it noteworthy within the Offline RL community.
> > >
> > > Moreover, the author's insight that "extending Q-function generalization to OOD regions within CHN can be beneficial " **is valuable**. The concept of learning mildly conservative Q value is aligned with the direction of current work (e.g., MCQ), and SQOG introduces a fresh perspective to achieve this. I encourage the authors to further elaborate on the insights and potential advantages of this approach for policy-constrained methods in their final submission.
> > >
> > > I have raised my score to 6.

---

> ### Author Response · Authors · 2024-11-26
> **Thanks for raising the score!**
>
> Thank you for your thoughtful review and for raising the score. We truly appreciate your recognition of the strong performance and computational efficiency of SQOG, as well as your insightful comment on extending Q-function generalization to OOD regions. We will take this valuable direction further and elaborate on the potential advantages of our approach, particularly in the context of policy-constrained methods, in Appendix E.
>
> Thank you again for your time and consideration.

---

### Official Review · Reviewer_etFc · 2024-10-29

**Soundness:** 3
**Presentation:** 3
**Contribution:** 3
**Rating:** 8
**Confidence:** 3

**Summary:**

This paper introduces the Smooth Bellman Operator (SBO), aimed at addressing the over-constraint issues prevalent in popular offline RL algorithms. Building on SBO, the authors propose SQOG, a practical algorithm that demonstrates superior performance over well-known baselines like CQL, IQL, and MCQ, on D4RL benchmark. The paper also includes ablation studies examining the impact of the hyperparameter \beta, which controls the out-of-distribution (OOD) generalization loss, and different types of noise. Additionally, the paper provides theoretical analysis that supports the efficacy of the SBO.

**Strengths:**

1. The paper is well-written and easy to follow.
2. SQOG achieves strong performance on the D4RL benchmark, remaining relatively simple, straightforward, and computationally efficient.
3. The sanity check on the Inverted Double Pendulum task empirically shows SBO’s effectiveness in alleviating the over-constraint issue, which is a valuable addition.

**Weaknesses:**

1. My key concern is the lack of discussion on the relationship between SBO and behavior cloning loss. While the paper claims that SBO alleviates the over-constraint issue in existing offline methods, it’s unclear to me that whether SBO’s effectiveness is limited to use with TD3+BC. Further discussion on this point would be beneficial.
2. The Inverted Double Pendulum task is relatively simple and differs from locomotion tasks, raising questions about whether SQOG can maintain accurate Q-value estimates in higher-dimensional environments. While visualizing Q-values in high-dimensional spaces may not be as straightforward as with the Inverted Double Pendulum, including experiments on a locomotion task to validate SQOG’s Q-estimation accuracy would strengthen this claim.
3. All experiments were conducted with only 4 seeds; additional seeds would improve result robustness.

**Questions:**

See above

---

> ### Author Response · Authors · 2024-11-23
> **Author Response to Reviewer etFc (part1/2)**
>
> Thank you for your insightful and thoughtful comments, as well as for the positive score. We greatly appreciate your recognition of the main contributions of our paper and the opportunity to address your concerns. Below, we provide detailed clarifications to each point raised. We hope our responses will adequately address your concerns.
>
> **Q1: Lack of discussion on the relationship between SBO and behavior cloning loss. Whether SBO’s effectiveness is limited to use with TD3+BC.**
>
> **A1:** Thank you for pointing out this important concern. We clarify that SBO’s effectiveness is not limited to TD3+BC or the use of behavior cloning (BC) loss.
>
> **Theoretical Justification:**
> As shown in Theorem 1, SBO approximates accurate out-of-distribution (OOD) Q-values within CHN under the assumption that the learned policy $\pi$ is close to the behavior policy $\mu$. This assumption is satisfied by any policy constraint offline RL algorithm that enforces $\max(\mathrm{KL}(\pi, \mu), \mathrm{KL}(\mu, \pi)) \leq \epsilon$. Therefore, the conclusion that SBO enables gradual approximation of true OOD Q-values within CHN is generalizable to other policy constraint methods beyond TD3+BC.
>
> **Empirical Validation:**
> To demonstrate SBO's generalizability, we replace the BC loss in TD3+BC with the KL divergence penalty as used in BRAC's actor loss [1]. Since the true behavior policy is inaccessible, we adopt behavior cloning (a common and simple method) to approximate it. We then re-implement BRAC and add SBO to it. Table 1 presents the performance of BRAC+SBO across various datasets.
>
> Table 1: Normalized average score comparison of BRAC+SBO against baseline method BRAC on MuJoCo "-v2" and Adroit "v1" datasets. The results are averaged over 4 different random seeds. We bold the highest scores.
>
> | Dataset           | BRAC | BRAC+SBO  |
> |--------------------|-------|-------|
> | halfcheetah-medium   | 49.8$\pm$1.2  |  **54.3$\pm$1.2** |
> | hopper-medium    |3.6$\pm$3.1  |  **90.9$\pm$2.9** |
> | walker2d-medium   | 7.8$\pm$8.1   |   **85.6$\pm$4.3** |
> | halfcheetah-medium-replay    | 41.8$\pm$6.2 |  **47.8$\pm$2.0** |
> | hopper-medium-replay     | 28.8$\pm$20.3 | **61.1$\pm$11.9** |
> | walker2d-medium-replay   | 8.5$\pm$3.0  | **67.6$\pm$11.0** |
> | Mujoco Average |  23.4 | **67.9** |
> |pen-human| 19.2$\pm$16.3  | **69.7$\pm$8.7** |
> |pen-cloned|  28.4$\pm$23.4  | **69.0$\pm$14.8**|
> |Adroit Average |    23.8 |     **69.4** |
>
> From Table 1, we observe a significant performance improvement when SBO is added to BRAC, confirming that SBO is a versatile plug-in for policy constraint methods.
>
> Although BRAC+SBO performs well, its performance does not surpass that of SQOG. This is likely because behavior cloning only provides an approximate behavior policy, and inaccuracies in this approximation weaken the KL penalty's ability to ensure $\pi$ is close to $\mu$, which may hurt the effectiveness of SBO. While advanced methods like generative models could improve behavior policy estimation, they are computationally expensive. In contrast, TD3+BC with BC loss strikes a favorable balance between simplicity, computational efficiency, and effectiveness.
>
> In summary, SBO is not restricted to TD3+BC or BC loss, and our empirical results demonstrate its potential with other policy constraint methods. However, the choice of BC loss in our work stems from its efficiency and practical effectiveness. For further details and experimental results, we have added an extended discussion in Appendix D.
>
> [1] Yifan Wu, George Tucker, and Ofir Nachum. Behavior regularized offline reinforcement learning.
>
> **Q2: Whether SQOG can maintain accurate Q-value estimates in higher-dimensional environments.**
>
> **A2:** Thank you for your valuable comment regarding the need for experiments on locomotion tasks to validate SQOG's Q-value estimation accuracy in higher-dimensional environments. We appreciate your suggestion and have addressed this concern by including additional experiments in **Appendix B.2**.
>
> Specifically, we evaluated SQOG on locomotion tasks from the D4RL benchmark, including `hopper-medium-v2`, `hopper-medium-replay-v2`, and `halfcheetah-medium-v2`. In these experiments, we analyzed the value difference between the Q-values estimated by SQOG’s critic network and the true Q-values obtained via Monte Carlo estimation.
>
> The results demonstrate that across all datasets, the value difference consistently decreases as training progresses. This indicates that SQOG is capable of achieving accurate Q-value estimation, even in high-dimensional environments like locomotion tasks.
>
> We hope this addition adequately addresses your concern and strengthens the validity of our claims regarding SQOG's Q-estimation accuracy. Thank you again for your insightful feedback.

---

> ### Author Response · Authors · 2024-11-23
> **Author Response to Reviewer etFc (part2/2)**
>
> **Q3: More seeds are needed.**
>
> **A3:** We run SQOG on MuJoCo datasets for another 4 seeds, yielding a total 8 random seeds, which we believe is comparatively sufficient for reliable evaluation. We summarize the results below. We observe that SQOG exhibits similar performance as reported in the main text.
>
> Table 2: Normalized average scores of SQOG evaluated on MuJoCo "-v2" datasets with 4 random seeds and 8 random seeds.
>
> | Dataset           | SQOG (4 seeds)| SQOG (8 seeds)  |
> |--------------------|-------|-------|
> | halfcheetah-random    | 25.6$\pm$0.4  |  25.7$\pm$1.8 |
> | hopper-random        | 15.6$\pm$3.3  | 15.4$\pm$4.1 |
> | walker2d-random   | 17.7$\pm$3.5   | 16.9$\pm$3.1  |
> | halfcheetah-medium   | 59.2$\pm$2.4  |  60.1$\pm$3.0 |
> | hopper-medium    | 100.6$\pm$0.7  |  100.8$\pm$0.6 |
> | walker2d-medium   | 82.9$\pm$0.8   |   82.3$\pm$1.3 |
> | halfcheetah-medium-replay    | 46.4$\pm$1.2  |  45.8$\pm$2.2 |
> | hopper-medium-replay        | 100.9$\pm$5.1 |  99.5$\pm$6.8 |
> | walker2d-medium-replay   | 88.3$\pm$3.5   | 86.5$\pm$5.8 |
> | halfcheetah-medium-expert    | 92.6$\pm$0.4  | 91.6$\pm$1.4  |
> | hopper-medium-expert       | 109.2$\pm$2.8 | 110.1$\pm$2.5|
> | walker2d-medium-expert   | 109.0$\pm$0.3   |   109.1$\pm$0.3 |
> | Mujoco Average |  70.7 |  70.3  |

---

> ### Author Response · Authors · 2024-11-25
> **Looking forward to your feedback**
>
> Dear reviewer etFc,
>
> Thank you very much for taking the time to review our work and for providing valuable feedback. As the end of the discussion period approaches, we would greatly appreciate it if you could confirm whether our response has adequately addressed your key concerns. We will be happy to have further discussions with the reviewer if there are still some remaining questions! More discussions and suggestions on further improving the paper are also always welcomed! We look forward to hearing from you and remain available to provide any additional details that might assist in resolving your concerns.
>
> If the concerns have been addressed, we kindly ask you to consider raising your rating. Thank you again for your time and efforts in reviewing our manuscript.
>
> Best regards,
>
> The authors

---

> > ### Comment · Reviewer_etFc · 2024-11-25
> >
> > I would like to thank the authors for providing the additional experiments and analysis. My concerns have been thoroughly addressed, and I have raised the score.

---

> > > ### Author Response · Authors · 2024-11-26
> > > **Thanks for raising the score!**
> > >
> > > Thank you sincerely for your recognition of our work. We are truly grateful for your thoughtful feedback, which helped us improve the paper. Your support and encouragement mean a great deal to us.

---

### Official Review · Reviewer_cDBr · 2024-11-03

**Soundness:** 2
**Presentation:** 2
**Contribution:** 3
**Rating:** 5
**Confidence:** 4

**Summary:**

This paper addresses the problem of generalization in offline RL. Most existing techniques tend to severely constrain the policy to produce actions that remain close to the data distribution, to avoid known nefarious effects of OOD generalization in unexplored parts of the state-action space. This paper suggests that these constraints can be alleviated. In particular, they identify a region close to the dataset (the CHN) where controlled estimation of the Q-values is possible, whereas this region is usually treated as completely OOD by existing techniques. After deriving a variant of the Bellman operator that leverages the existence of this region and showing some of its properties, the authors propose a concrete regularization term to add to the classic critic loss in offline RL. The authors first showcase the existence of the over-constraining problem on a classic offline RL algorithm, and that their method alleviates this issue. They then evaluate their method on the D4RL benchmark, showing improved performance compared to baselines while having a quick runtime.

**Strengths:**

- First, I find that the problem tackled is important and very well motivated. I found in particular that the introduction did a great job of exposing to the reader the problem of over-constraining the policy. The simple but logical explanation is rooted in the design of existing policies that completely avoid to generalize OOD. Having described the problem, the authors are then clear in their ambition: leveraging the part of the state space where neural networks actually are able to generalize to improve the approximation of Q, and possibly of the learned policy. I think this idea is very sensible, and sounds like a natural and promising follow-up to the existing literature following the introduction. I expect a work in this direction to be very interesting.
- The practical algorithm is simple to implement, as it is just adding a smoothing term to the loss. This is a strong point of the empirical part of the proposed approach, since practitioners can easily add this term to their training loss. This also makes the algorithm very quick to run, which is a great advantage compared to certain competitors (such as MCQ) that are much slower.
- I find the sanity check (in particular Fig 1.b) to be a very convincing illustration of the problem that motivated the paper. The authors show how a classic baseline, TD3+BC, tends to keep its high Q-value predictions close to the regions of high data density. I find this is a great illustration of the over-constraining problem. At the same time, this figure shows that SQOG does not suffer from the same limitations.
- The ablation study illustrated in Fig. 4 was a welcome addition that shows the sensitivity of the method to one of its main hyperparameters.
- Finally, the algorithm seems to perform well according to the results in Table 1. In Fig. 1, the authors show that their method is indeed more robust to OOD and approximates well the true action values, while TD3+BC tends to keep its predictions well within the data distribution. This is consistent with the story of the paper.

**Weaknesses:**

Unfortunately, I found that the paper lacked clarity and mathematical rigor at several crucial moments, which severely hurt my understanding of the contribution.
- Most importantly, I believe there is a mistake with the definition of perhaps the most central object of the paper, CHN. As it is defined, by construction, $N(Conv(D)) \subset (S,A)_D \subset Conv(D)$, therefore $CHN(D) = Conv(D)$. It is not clear how to change the existing definition to get to the one that seems intended by the authors (on this note, a figure of $(S,A)_D$, $Conv(D)$, and $N(Conv(D)$) would have been enlightening, I encourage you to add one).
- In Definition 1, the definition of $(S,A)_D$ is confusing: it is mentioned to be a space, using the space notation of the preliminaries, but it seems from the rest of the definition that it is the set of state-action tuples from D. Moreover, r is named as the neighborhood radius and seems like a free parameter to set, but is then defined in Proposition 1 as a function of D and its convex hull: what definition should we follow?
- The definition of OOD was another major point of confusion to me. It is indicated that actions that are OOD are actions that are outside the support of the behavior policy $\mu$ (which is restated in Definition 2). But as far as I can tell there are no constraints on the behavior policy $\mu$ that generated the dataset. What if this policy put a non-zero probability on every action? This is far from an edge case, since a behavior policy parametrized as a Gaussian distribution would have support on the full action space. Then, what is the relationship with D? The text pushes me to think that what is OOD is what is far enough from D, but within the CHN (L153): then is OOD what is far from the empirical distribution derived from D? To illustrate the confusion, the authors write at L317 that the [-0.5, 0.5] region in Fig.1 is OOD, but the density is clearly non-zero for almost all of this range, contradicting how OOD was defined earlier. Overall, my impression is that two notions of OOD are colliding in this paper: the formal one from Definition 2 stating that every action outside of the support of $\mu$ is OOD, and the intuitive one from Figure 1 that regions of low-density are OOD.
- The confusion inherited from the unclear definitions of the spaces of Sec 3.1 is transfered to Sec 3.2. For instance, in Definition 2, the authors refer to “neighborhood”. why not use a precise mathematical object that you defined earlier, likely in Definition 1? On that note, the neighborhood was defined w.r.t state-action tuples in Definition 1, but now refers to the action only, which does not help with the clarity of exposition. If that is normal, then I think it would be useful to comment on why the state space needs to be included in the CHN definition. As a consequence, neighbor/neighborhood is used several times (L170/175/214) but since it is not clear to me where the exact definition is, it is difficult to follow the text more than at an intuitive level.

_Other weaknesses_
- I find that a discussion of the link between the SBO and the proposed loss would have been helpful for the flow of the paper. As it stands, Section 3.2 introduces the SBO and some of its properties, while Sec 3.3 switches direction and proposes a regularization term, a bit out of the blue. I think a smoother transition between the two section, justifying the key elements to keep of the SBO, and how they motivated the new regularization term, would have helped the exposition.
- One of the key aspects of the empirical loss is the noise added to the in sample action to possibly get an artificial OOD sample. The authors did an ablation on the type of noise, but I would have been curious to see a discussion on the scale of the noise instead, as this should control the amount of “OOD-ness” and might influence the performance of the algorithm.
- In proposition 1, the authors refer to the NTK regime without explaining the acronym nor introducing the general framework. I think the framework should be at least introduced.
- I am also unclear about the last sentence of the paragraph preceding proposition 2 (l. 138-139): how did you derive this statement?
- The norm used in all mathematical statements besides Theorem 4 is unclear.
- at L170, the authors write “the dataset action […]”: this assumes that this action exists and that it is unique, but it is not clear to me if this is true.

_Minor weaknesses that did not affect the score but could be addressed anyway_
- A suggestion: Fig. 1 (1.b in particular) does a great job of illustrating the overconstraint issue of TD3+BC. It would be interesting to consider showing this figure earlier in the paper. I am OK with where the figure is right now and this did not impact my score, but I think this could convince the reader very early on that the problem you are tackling is real and substantial.
- In the experiments, I found confusion the exact definition of the true Q values (L314): are you using Monte Carlo estimation of a (s,a) pair, resetting the environment at this state? Similarly, what do you mean exactly by standardized and smoothed (L315)?
- Besides the comments above, I find that the mathematical notation was imprecise or inconsistent at a couple other spots. For instance, replacing “noise” in Eq. 14 by an actual variable; the action $a^{in}_{neighbor}$ is indicated sampled from a support (L170); the expectations are taken w.r.t $\pi$ (L.85) or $\pi(\cdot | s’)$.
- “Longest diameter” -> you can drop the “longest”, diameter is already the supremum of the distances within the set.
- L135: “Due to the uniqueness of CHN”: this fact is never stated before, but since it is referred to like this, I would have expected to have a sentence stating that the CHN is unique (maybe at L115-116) with the other properties.
- In the appendix, you indicate in the proof “we can easily derive” (L744), but in a proof I would expect for this derivation to be there.
- The authors ran several algorithms on several algorithms, but unfortunately on a rather small number of seeds (4). A higher number of seeds could have gotten these confidence interval smaller and non intersecting.

**Questions:**

- Can you please precise the different definitions, most importantly of CHN? I believe there is a mistake in its current form. Please also confirm what $(S,A)_D$ is, as it is not properly defined (in mathematical terms).
- Could you please write a mathematical definition that confirms the notion of OOD that was chosen in this work? Is it the one of Definition 2?
- Could you comment on the link between the SBO and the OG loss of Eq.13? Are these two objectives equivalent? If not, what are the differences? What is captured by one but not the other?
- How did you decide on the scale (and the clip) of the noise distribution, indicated at p.19? Are they sensible hyperparameters for the performance of the algorithm?
- Regarding Eq. 14: for s’ that are at the “boundary” of D, and a’ potentially far from the dataset: the OOD backprop effect will not be countered by the new smoothing term, is that correct?
- $a_{ood}$ seems to implicitly belong to CHN in Prop. 3. $a_{ood}$ was introduced in the main text of Sec 3.2, was it already part of CHN then or could it also be outside of CHN?

---

> ### Author Response · Authors · 2024-11-16
> **Author Response to Reviewer cDBr (part 1/4)**
>
> Thank you for your insightful and thoughtful comments. We greatly appreciate your interest in our paper and the opportunity to clarify any concerns. Below, we provide detailed responses to your questions, followed by specific clarifications addressing the misunderstandings noted in the weaknesses section.
>
> **Q1: Can you please precisely provide the different definitions, most importantly of CHN?**
>
> **A1:** We apologize for a typo in the definition of $N(Conv(D))$ in Definition 1. The corrected definition is as follows: $N(Conv(D))=\\{x\in (S,A) \mid\min_{y\in Conv(D)}\\|x-y\\|\leq r\\}$. We also clarify the definitions of $(S, A)$ and $(S, A)\_{D}$:
>
> The entire state-action space $(S,A) = \\{ (s,a) | s \in S, a \in A \\}$, where $S$ is the state space, $A$ is the action space. The in-sample state-action **set** $(S,A)\_{D} = \\{(s,a)|(s,a) \in D\\}$. Here, we use $(s, a) \in D$ for simplicity to represent state-action pairs in the dataset $D$, where $D$ typically consists of tuples $(s, a, r, s', d)$. This notation is unambiguous as we focus on state-action pairs. To conclude, we update the definition of the CHN.
>
> For a given dataset $D$, we define the in-sample state-action set $(S,A)\_{D} = \\{(s,a)|(s,a) \in D\\}$.
> CHN is defined as the union of the convex hull and its neighborhood of $(S,A)\_{D}$: $\text{CHN}(D)=Conv(D)\cup N(Conv(D))$, where $Conv(D)=\\{\sum\_{i=1}\^n \lambda\_i x\_i | \lambda\_i \geq 0, \sum\_{i=1}\^n \lambda\_i=1, x\_i\in (S,A)\_{D}\\}$ is convex hull, and $N(Conv(D))=\\{x\in (S,A) \mid\min\_{y\in Conv(D)}\\|x-y\\| \leq r\\}$ is the neighborhood. $r$ is the neighborhood radius, and it is always chosen to be smaller than or equal to the diameter of $Conv(D)$. Specifically, the radius $r$ in Proposition 1 is a feasible choice satisfying $r\leq diameter(Conv(D))$.
>
> **Q2: Could you please write a mathematical definition that confirms the notion of OOD that was chosen in this work?**
>
> **A2:** In offline reinforcement learning, OOD (Out-of-Distribution) actions refer to actions that fall outside the distribution of actions observed in the offline dataset $D$ used for training. For every state $s\in D$, we define the empirical conditional distribution $\hat{\mu}(a|s) = \frac{\Sigma_{(s',a')\in D}\mathbf{1}[s'=s,a'=a]}{\Sigma_{s' \in D}\mathbf{1}[s'=s]}$. We consider an action $a$ to be an **OOD** action for a given state $s$ if $\hat{\mu}(a|s) = 0$, meaning it has never been observed in the dataset $D$ for this state. With this definition, OOD actions can still exist within the CHN because the set $\text{CHN}(D)-(S,A)_{D}$ is not empty.
>
> In our experiments, we relax this strict definition. We consider an action $a$ to be an **OOD** action for a given state $s$ if $\hat{\mu}(a|s) \approx 0$, meaning it is rarely observed in the dataset $D$. Therefore, in the sanity check, we refer to regions of low density as OOD. This relaxed definition allows us to better illustrate the intuitive results, while still distinguishing OOD actions from in-sample actions, which have higher density.
>
> Both the strict and relaxed definitions of OOD have been widely used in prior work [1,2], and the relaxed definition is generally accepted in the offline RL community. Since offline RL relies on a static dataset collected from previous interactions, it can only confidently estimate the value of actions similar to those in this dataset. Existing methods become overly conservative to prevent the overestimation of OOD actions. We design the CHN to identify safe OOD regions and enhance the $Q$-function's generalization within these regions, addressing both overestimation and over-constraint issues.
>
> From our paper and the discussion, you may see that the OOD actions are important. What about the OOD states? Note that Q-function training in offline RL does not suffer from state distribution shift, as the Bellman backup never queries the Q-function on OOD states [1]. **Why the state space needs to be included in the CHN definition?**
>
> We study the Q-function in this paper, which is also called the state-action value function. The state is necessary when discussing OOD actions, as the state-action pair defines the context in which the action is taken. Moreover, the safety guarantees for the Q-function defined on the CHN also require the "state" to be included in the definition, as it directly impacts the determination of OOD actions and their associated Q-values.
>
> We hope this clarification addresses your confusion. A more detailed explanation will be included in the final version of the paper.
>
> [1] Kumar, A. et al. "Conservative Q-learning for offline reinforcement learning." Advances in Neural Information Processing Systems 33 (2020): 1179–1191.
> [2] Xu, H. et al. "Offline RL with no OOD actions: In-sample learning via implicit value regularization." arXiv preprint arXiv:2303.15810 (2023).

---

> > ### Comment · Reviewer_cDBr · 2024-11-22
> >
> > First of all, I thank the authors for the extensive rebuttal and the additional experiments that I believe improve the paper. Still, several concerns subsist, which I describe below.
> >
> > A1:
> > Thank you for your answer and for correcting the definition of the CHN.
> > With this new definition, I believe that Conv(D) is a subset of N(Conv(D)), since $\forall r, \forall x \in Conv(D), min_{y \in Conv(D)} ||x - y|| = 0 \leq r$.
> > Therefore CHN would be equal to N(Conv(D)): is that correct?
> >
> > A2:
> > Thank you for clarifying your definition of OOD.
> > On a general note, I understand that relaxing the definition of OOD for the experiments is common in the literature — I do not have a problem with that. However, I do believe it is important to stay precise on the different definitions of the notions used in the paper, and explicitly mention when you pass from one to the other.
> >
> > Now, regarding the definition of $\hat{\mu}$:
> > Given this definition, my understanding in the general continuous case, every action sampled from the policy is going to be OOD, since it will not have been seen in the dataset (since in general we will have that $P(\pi_\phi(s) = a) = 0$).
> > Is that correct?
> >
> > If yes, in my understanding of the SBO, every action sampled by $\hat{B}^\pi $ will be OOD, and therefore replaced by the in-sample neighbor.
> > Is that the case? If not, can you please clarify.
> >
> >
> > Finally, regarding the note on the inclusion of the state in the CHN:
> > I think your comment about the state “providing context” makes intuitively sense to me, however it is not clear to me how this precisely translates, especially why it is needed to include the state in the definition of the CHN (which tells us what part of the OOD region is safe to generalize on) but we can focus mostly only on the action being OOD.
> > You mention that “A more detailed explanation will be included in the final version of the paper.”. Please let me know if you do include it during the rebuttal period.

---

> > > ### Author Response · Authors · 2024-11-23
> > > **Author Response to Reviewer cDBr**
> > >
> > > We sincerely appreciate your thoughtful comments and recognition of our efforts to improve the paper through additional experiments and clarifications. Below, we address the concerns you have raised and provide detailed explanations for each point:
> > >
> > > **Response to A1:** With the initial definition, $N(Conv(D))=\\{x\in (S,A) \mid\min_{y\in Conv(D)}\\|x-y\\|\leq r\\}$, it is correct that $Conv(D)$ is a subset of $N(Conv(D))$, and thus CHN would be equal to $N(Conv(D))$, that is correct. However, to provide greater clarity, we refine the definition of the neighborhood to the "external neighborhood". We now define $N(Conv(D))=\\{x\in (S,A)\mid x \notin Conv(D), \min\_{y\in Conv(D)}\|x-y\|\leq r\\}$, which explicitly excludes $Conv(D)$. Under this updated definition, CHN comprises two distinct parts: the convex hull $Conv(D)$ and the external neighborhood $N(Conv(D))$, i.e., $\textit{CHN}(D)=Conv(D)\cup N(Conv(D))$.
> > >
> > > **Response to A2:**
> > >
> > > 1. We appreciate your acknowledgment regarding the relaxed definition of OOD in the experiments. To address your concern about staying precise with the definitions, we would like to clarify that we have explicitly mentioned the transition between the different notions of OOD in both the Preliminaries section (L104) and the corresponding footnote.
> > >
> > > 2. Thank you for your acknowledgment. In the general continuous case, it is indeed correct that **almost** every action sampled from the policy $\pi_\phi$ will be OOD. To elaborate, let the event A denote "an action sampled from the policy $\pi_\phi$ is in the dataset, i.e., $\pi\_\phi(s)=a, a\in D$". In this case, we have $P(A) = 0$.  However, it is important to note that $P(A) = 0$ doesn't imply that the event A never happens. This distinction is why we state that "almost" every action sampled from the policy $\pi\_\phi$ is OOD. Thus, in SBO, an OOD action is replaced by the in-sample neighbor action.
> > >
> > > 3. The definition of the CHN is detailed discussed in **Appendix C**. We hope this provides further clarity, and we would be happy to elaborate more if needed.

---

> ### Author Response · Authors · 2024-11-16
> **Author Response to Reviewer cDBr (part 2/4)**
>
> **Q3: Could you comment on the link between the SBO and the OG loss?**
>
> **A3:** The OG loss is designed for OOD generalization. It can be derived from $\mathcal{G}\_1$ in the SBO. Specifically, in $\mathcal{G}\_1$, we use the neighboring in-sample $Q$-values $Q\(s,a^{in}\_{neighbor}\)$ to update the OOD $Q$-values. To achieve this, we introduce a Mean Squared Error (MSE) loss between the parametric OOD $Q$-values $Q_{\theta}(s,a^{ood})$ and the neighboring in-sample $Q$-values $Q(s,a^{in}_{neighbor})$.
>
> Additionally, the term $\hat{\mathcal{B}}^\pi$ in the SBO corresponds to the loss term in the first part of Eq. (14). This critic loss is commonly used in reinforcement learning, as introduced in the preliminary section.
>
> **Q4: Regarding Eq. 14: For the faraway OOD actions, will the OOD backpropagation effect be countered by the new smoothing term?**
>
> **A4:** When you mention "a' potentially far from the dataset," we assume you are referring to OOD actions that lie outside the CHN (Convex Hull and its Neighborhood). For these faraway OOD actions, we do not have a safety guarantee in place. We acknowledge that we have not yet addressed the $Q$-values of these faraway OOD actions, as they are difficult to estimate reliably. In fact, we propose a method to learn the OOD $Q$-values within the CHN (i.e., near the dataset). However, learning the $Q$-values for faraway OOD actions remains an open challenge in the field.
>
> **Q5: The explaination of $a^{ood}$.**
>
> **A5:**  As mentioned in the last sentense of Sec 3.1, we only focus on OOD actions within the CHN. In Sec 3.2, for simplicity, we use $a^{ood}$ to represent the OOD action within the CHN. In Sec 3.3, we provide detailed explanation when using $a^{ood}$.

---

> > ### Comment · Reviewer_cDBr · 2024-11-22
> >
> > A3&A4:
> > Thank you for your answer. To be clear, I was wondering about the equivalence (or not) of the theoretical SBO operator and the empirical SQOG loss you propose, and the individual roles of the terms of Eq. 14 were clear. I understand how adding this regularization term has a similar effect to the SBO that backs up the value of a neighboring action when applied on an OOD action. However, reading the paper, I was expecting the empirical bellman operator in the usual actor critic loss to be replaced by an empirical version of the SBO, rather than a regularization term, since you derived the theory for the SBO. Could you comment on why you did not replace the empirical bellman operator by an empirical version of the SBO?
> >
> > For instance (regarding A4), if a’ is outside of the CHN, it seems that the loss of Eq. 14 will still bootstrap the corresponding value (which my original comment was about), while this case should not happen in the SBO since the Q values of OOD actions should be replaced by an in sample neighborhood action.
> >
> > A5:
> > I acknowledge your answer. Reading the paper, I still find the notation could be improved and as it stands leads to confusion, since OOD is used both to refer to any tuple (s,a) that is not in D (CHN and outside of the CHN) and to denote actions $a_ood$ that are in the CHN. I do not think the sentence in english at the end of Sec 3.1. is sufficient to clarify the discussion.

---

> ### Author Response · Authors · 2024-11-16
> **Author Response to Reviewer cDBr (part 3/4)**
>
> **Q6: How did you decide on the scale (and the clip) of the noise distribution? Are they sensible hyperparameters for the performance of the algorithm?**
>
> **A6:**
> We appreciate the reviewer’s question regarding the choice of noise distribution scale and clip. To address this, we conducted additional experiments with varying configurations of scale and clip values on multiple datasets. The results are summarized in the table below:
>
> Table 1: Normalized average score of SQOG over different choices of gaussian noice scale and clip on MuJoCo "-v2" datasets. The results are averaged over 4 different random seeds.
> | Dataset           | scale=0, clip=0| scale=0.2, clip=0.3  |scale=0.6, clip=0.5  |scale=1.0, clip=0.7  | scale=2.0, clip=1.0|
> |--------------------|-------|-------|-------|-------|-------|
> | halfcheetah-medium   | 51.2$\pm$3.9  |  60.6$\pm$0.5 | 59.2$\pm$2.4  |  53.1$\pm$1.9 | 51.7$\pm$1.5|
> | hopper-medium    | 1.5$\pm$0.3  |  67.2$\pm$1.6 | 100.6$\pm$0.7  |  94.5$\pm$3.6 | 79.5$\pm$10.3|
> | walker2d-medium   | 48.3$\pm$1.2   |   58.9$\pm$2.3 | 82.9$\pm$0.8  |  86.0$\pm$1.0 | 82.5$\pm$1.1|
> | halfcheetah-medium-replay    | 49.2$\pm$0.3  |  47.8$\pm$2.2 |  46.4$\pm$1.2  |  36.6$\pm$1.1| 35.8$\pm$ 0.8|
> | hopper-medium-replay     | 22.3$\pm$6.6 | 19.4$\pm$6.8 | 100.9$\pm$5.1  |  24.3$\pm$2.7 | 27.1$\pm$6.7 |
> | walker2d-medium-replay   | 12.2$\pm$5.5  | 56.8$\pm$2.0 | 88.3$\pm$3.5  |  90.1$\pm$3.6 | 60.6$\pm$6.9 |
> | Average |  30.8 | 51.8 | 79.7 | 64.1 | 56.2 |
>
> From these results, we observe the following:
> 1. **Moderate Noise Balances Exploration and Stability:** Across most datasets, a scale of 0.6 and clip of 0.5 consistently achieves strong performance, suggesting that moderate noise levels effectively balance exploration and stability. While noise promotes generalization, excessive noise can lead to sampling outside the CHN boundary, resulting in OOD Q-values that violate **safety guarantees** and degrade performance. Properly scaled noise ensures effective learning while respecting safety constraints.
>
> 2. **Dataset-Specific Effects of Noise:** The sensitivity to the noise distribution parameters (scale and clip) varies across datasets.
> In halfcheetah-medium-replay, the baseline configuration (scale=0, clip=0) yields the best performance, indicating that the effect of noise on in-sample Q-value estimation cannot be ignored. The dataset may already provides sufficient diversity, and adding noise introduces harmful uncertainty, leading to performance degradation. For such datasets, it’s crucial to keep noise parameters conservative to avoid disrupting in-sample Q-learning and maintain stable performance.
> In contrast, for halfcheetah-medium, the optimal configuration is (scale=0.2, clip=0.3), and for walker2d-medium and walker2d-medium-replay, (scale=1.0, clip=0.7) achieves the best results. These differences highlight that there is room for improvement in fine-tuning noise parameters across various datasets.
>
> 3. **Effectiveness of Noise Injection:** The baseline configuration (scale=0, clip=0) significantly underperforms in most datasets, underscoring the necessity of noise injection to enhance exploration and overall performance. Noise injection is essential for improving the Q-function's ability to generalize to previously unexplored regions and optimize learning.
>
> **Q7: More seeds are needed.**
>
> **A7:** We run SQOG on MuJoCo datasets for another 4 seeds, yielding a total 8 random seeds, which we believe is comparatively sufficient for reliable evaluation. We summarize the results below. We observe that SQOG exhibits similar performance as reported in the main text.
>
> Table 2: Normalized average score of SQOG on D4RL benchmarks. 0 corresponds to a random policy and 100 corresponds to an expert policy. The experiments are run on MuJoCo "-v2" datasets.
> | Dataset           | SQOG (4 seeds)| SQOG (8 seeds)  |
> |--------------------|-------|-------|
> | halfcheetah-random    | 25.6$\pm$0.4  |  25.7$\pm$1.8 |
> | hopper-random        | 15.6$\pm$3.3  | 15.4$\pm$4.1 |
> | walker2d-random   | 17.7$\pm$3.5   | 16.9$\pm$3.1  |
> | halfcheetah-medium   | 59.2$\pm$2.4  |  60.1$\pm$3.0 |
> | hopper-medium    | 100.6$\pm$0.7  |  100.8$\pm$0.6 |
> | walker2d-medium   | 82.9$\pm$0.8   |   82.3$\pm$1.3 |
> | halfcheetah-medium-replay    | 46.4$\pm$1.2  |  45.8$\pm$2.2 |
> | hopper-medium-replay        | 100.9$\pm$5.1 |  99.5$\pm$6.8 |
> | walker2d-medium-replay   | 88.3$\pm$3.5   | 86.5$\pm$5.8 |
> | halfcheetah-medium-expert    | 92.6$\pm$0.4  | 91.6$\pm$1.4  |
> | hopper-medium-expert       | 109.2$\pm$2.8 | 110.1$\pm$2.5|
> | walker2d-medium-expert   | 109.0$\pm$0.3   |   109.1$\pm$0.3 |
> | Mujoco Average |  70.7 |  70.3  |

---

> ### Author Response · Authors · 2024-11-16
> **Author Response to Reviewer cDBr (part 4/4)**
>
> **Q8: Comments on "Weaknesses".**
>
> **A8:** The responses to the first two weaknesses are provided in A1, and the response to the third weakness is in A2. Below, we address the clarification needed for Definition 2 to respond to the fourth weakness.
>
> The neighborhood of an action $a$ is defined as a closed set: $N\_{\delta}(a) = \\{a'\in A|~||a'-a|| \leq \delta \\}$. This definition is similar to the neighborhood of the convex hull, as we consistently use the ''closed neighborhood'' in both definitions throughout this paper. In Definition 2, $a^{in}\_{neighbor}$ denotes an action within the neighborhood of action $a$ since $||a^{in}\_{neighbor} - a|| \leq \delta$, implying $a^{in}\_{neighbor} \in N\_{\delta}(a)$. Let $a^{in}\_{neighbor} \in D$, then $a^{in}\_{neighbor}$ is also a dataset action. Note that $a^{in}\_{neighbor}$ represents any action in the dataset that also falls within the neighborhood of action $a$. For improved clarity, we will change "the dataset action" to "a dataset action".
>
> Additionally, we have clarified "Why the state space needs to be included in the CHN definition?" in A2.
>
> **Q9: Comments on "Other weaknesses".**
>
> **A9:**
> 1. The link between the SBO and the proposed loss is discussed in A3 before.
>
> 2. The discussion of the noise scale is in A6.
>
> 3. The introduction of the NTK regime is provided in Appendix A. Following your advice, we will present the general framework in the main paper.
>
> 4. From Eq. (3) and (4) in Proposition 1, we can derive that, $\forall x\_{in} \in CHN$, $||Q\_{\theta}(x\_{in}) - Q\_{\theta}(Proj\_{D}(x\_{in})|| \leq \max\left\\{M\_{1},M\_{2}\right\\}$. All the norms here are $\mathcal{L}\_{2}$ norms $\\| \cdot \\|\_{2}$ and all the $Q$-values are real numbers, so the $\mathcal{L}\_{2}$ norms reduce to absolute values. Then, we have, $\forall x\_{in} \in \textit{CHN}$, $\\|Q\_{\theta}(x\_{in})\| - \|Q\_{\theta}(Proj\_{D}(x\_{in})\\| \leq \max\left\\{M\_{1},M\_{2}\right\\}$. Since $Proj\_{D}(x\_{in}) \in D$, $\\|Q\_{\theta}(Proj\_{D}(x\_{in}))\\| \leq sup\_{x\_i \in D} \\|Q\_{\theta}(x\_i)\\|$. Then, $\forall x\_{in} \in \textit{CHN}$, $\\|Q\_{\theta}(x\_{in})\\| - sup\_{x\_i \in D} \\|Q\_{\theta}(x\_i)\\| \leq \max\left\\{M\_{1},M\_{2}\right\\}$. Finally, we can derive the statement.
>
> 5. All the $\\| \cdot \\|$ norms are $\mathcal{L}\_{2}$ norms $\\| \cdot \\|\_{2}$. Although this is a commonly accepted default convention, we will elaborate on this point in the paper.
>
> 6. As mentioned in A8, $a^{in}\_{neighbor}$ represents any action in the dataset that also falls within the neighborhood of action $a$. For improved clarity, we will change "the dataset action" to "a dataset action".
>
> **Q10: Comments on "Minor weaknesses".**
>
> **A10:**
> 1. This is a really good suggestion. We have made so many efforts to demonstrate this intuitive result, and we will consider showing this figure earlier in the paper.
>
> 2. We calculate the true $Q$-values using a Monte Carlo estimation method. Specifically, we reset the environment to a given state $s$ and execute the action $a$. Starting from $(s,a)$, we simulate full trajectories and calculate the discounted return for each trajectory. To approximate the expected return, we repeat this process for 1000 sampled trajectories and take the average. We compute the $Q$-values for every 0.01 increment within the action range [-1.0, 1.0] and smooth the values using cubic spline interpolation. Finally, we normalize the values to keep them between 0 and 1 by multiplying by an appropriate constant. We will use "normalized" instead of "standardized" in the paper.
>
> 3. We will definitely modify the imprecise and inconsistent mathematical notations. Thanks for your advice.
>
> 4. We will use "diameter" instead of "longest diameter". Thanks for your advice.
>
> 5. We will claim the uniqueness of the CHN after Definition 1. Thanks for your advice.
>
> 6. We will add a detailed derivation for (L744). Thanks for your advice.
>
> 7. The results for another 8 seeds are in A7.
>
> **Conclusion:** We believe the clarifications we've provided address the issues raised. We are grateful for your constructive suggestions, which have helped enhance the clarity and quality of our work. We look forward to your feedback on the clarifications and hope they effectively resolve your concerns.

---

> > ### Comment · Reviewer_cDBr · 2024-11-22
> >
> > A8-10
> > Thank you for answering all of my smaller points. I do not have any further question at this time on these topics.

---

> ### Comment · Reviewer_cDBr · 2024-11-22
>
> A6:
> Thank you for the additional experiment, this ablation along with the analysis is helpful and improves the manuscript.
> I have one small question: I note that you consistently used normals where the truncation was small compared to the scale. How was this probability implemented? And given the rather small truncation, how to reconcile these results with the poor performance of the uniform noise in Fig.4?
>
> A7:
> Thanks for running the additional seeds, this is valuable.
> To be clear, I was hoping to have the extended number of seeds for all algorithms so that the confidence intervals of Table 1 would not intersect, but given the limited time of the rebuttal, this already confirms the good performance of SQOG, which is commendable. On that note, can you confirm what is the measure of uncertainty you use in Table 1?

---

> > ### Author Response · Authors · 2024-11-23
> > **Author Response to Reviewer cDBr**
> >
> > **Response to A6:** Thank you for your question. In our implementation, we generate noise using `noise = jax.random.normal(key, shape) * scale` and apply clipping with `noise_clipped = jnp.clip(noise, -clip, clip)`.
> >
> > Here, the noise follows a normal distribution with a mean of `0` and a standard deviation determined by the `scale`. The normal distribution is characterized by most values being concentrated around the mean `0`, with the probability of extreme values decreasing rapidly as they move away from the mean. This results in a conservative noise distribution, where most generated values are small and close to zero, providing controlled perturbations. The clipping further ensures that any noise exceeding `[-clip, clip]` is limited, adding an extra layer of constraint to prevent excessive deviations.
> >
> > In contrast, uniform noise is generated as a uniform distribution within the specified `clip` range. For example, in Fig. 4 (d)(e)(f), we added uniform noise within `[-0.5, 0.5]`.(the normal distribution also uses a `clip` of `0.5`) The uniform noise tends to perform worse because it has a **higher probability** of generating values closer to the clip boundaries. If larger noise values occur during the early stages of training, they can potentially disrupt the learning of the in-sample $Q$-function.
> >
> > As shown in Fig. 4 (d)(e)(f), uniform noise occasionally achieves performance close to that of normal noise but exhibits higher variance overall. This suggests that while uniform noise has potential, while its less controlled nature leads to inconsistent results compared to the more conservative normal noise. Furthermore, we conducted additional experiments with varying configurations of the uniform noise range (clip values) on multiple datasets. The results are summarized in the table below:
> >
> > Table 1: Normalized average score of SQOG over different choices of uniform noise clip on MuJoCo "-v2" datasets. The results are averaged over 4 different random seeds.
> >
> > | Dataset           | clip = 0.3 | clip = 0.5  |
> > |--------------------|-------|-------|
> > | halfcheetah-medium    | 50.4$\pm$7.2  |  47.4$\pm$11.1 |
> > | hopper-medium       | 55.5$\pm$34.6  | 62.5$\pm$36.5 |
> > | hopper-medium-replay   | 16.6$\pm$6.2   | 37.9$\pm$38.3 |
> >
> > Based on the experimental results, on halfcheetah-medium dataset,  a smaller uniform noise range (clip = 0.3) effectively reduces variance and slightly improves performance compared to a larger range (clip = 0.5), achieving behavior closer to the stability of normal noise. However, on the hopper-medium-replay, the smaller uniform noise range fails to generalize well to OOD Q, while also negatively impacting in-sample Q learning, resulting in significantly degraded performance.
> >
> > These results highlight that even with optimized ranges, uniform noise remains less reliable in both performance and stability compared to normal noise. This is because uniform noise has a non-negligible probability of generating values near the clip boundaries, which negatively impacts in-sample Q learning. During the early stages of training, when the in-sample Q function is still inaccurate, a more conservative noise distribution is essential, making normal noise a more suitable choice.
> >
> > **Response to A7:** Thank you for your thoughtful comments and for recognizing the value of the additional experiments. To address your concern, the baseline algorithms in Table 1 are state-of-the-art (SOTA) reinforcement learning methods that have been widely validated and accepted by the research community. Specifically, we obtained their results from MCQ [1] and DOGE [2], which ensures their reliability.
> >
> > Regarding the measure of uncertainty in Table 1, we use the standard deviation as the metric. For example, a result reported as $100.6\pm0.7$ means that we ran four experiments with different random seeds, calculated the average normalized score over the final ten evaluation episodes for each experiment, and then computed the mean (100.6) and standard deviation (0.7) across the four runs. This is the standard practice for reporting results in offline RL experiments on the D4RL benchmark.
> >
> > [1] J. Lyu, X. Ma, X. Li, and Z. Lu. Mildly Conservative Q-Learning for Offline Reinforcement Learning. NeurIPS 2022.
> >
> > [2] J. Li, X. Zhan, H. Xu, X. Zhu, J. Liu, and Y.-Q. Zhang. When Data Geometry Meets Deep Function: Generalizing Offline Reinforcement Learning. ICLR 2023

---

> ### Author Response · Authors · 2024-11-23
> **Author Response to Reviewer cDBr**
>
> **Response to A3\&A4:**
> Thank you for your insightful comments. We acknowledge your concern regarding the discrepancy between the theoretical SBO operator and the practical SQOG loss. This issue is not uncommon in offline RL algorithms. For instance, BCQ [1] incorporates mechanisms such as convex combinations of double critics for target value computation and a perturbation network to enhance action diversity. These additions deviate from BCQ's theoretical foundation and lack direct theoretical support, illustrating the challenges in preserving theoretical algorithms when deploying them with deep neural networks.
>
> Implementing a direct empirical version of the SBO is indeed challenging. It would require determining whether a sampled action $a' \sim \pi(\cdot \mid s')$ is OOD, specifically verifying if $\hat{\mu}(a' \mid s') = 0$ or $\hat{\mu}(a' \mid s') \approx 0$. However, in high-dimensional datasets, accurately estimating $\hat{\mu}$ is computationally expensive, and designing a reliable criterion is non-trivial.
>
> Despite this, the key intuition and innovation underlying SBO remain consistent in our empirical SQOG loss. In SBO, if $(s, a)$ is OOD, the base Bellman operator $\hat{\mathcal{B}}\_2$ is unaffected, and the OOD $Q$-values are updated using the neighboring in-sample $Q$-values $Q(s, a\_{\text{neighbor}}^{\text{in}})$. Similarly, in SQOG, we actively generate OOD actions to train the OOD $Q$-value with the in-sample neighboring $Q$-value, while ensuring they are not directly backed up by the empirical Bellman operator. The theoretical insights from SBO guided the design of our empirical loss, particularly in handling OOD actions effectively.
>
> As you mentioned, $\hat{Q}\_{\theta'\_i}(s', a')$ might correspond to an OOD $Q$-value. However, we do not replace it with the in-sample neighbor $Q$-value because identifying whether $a'$ is OOD is infeasible in practice. Nevertheless, as the OOD $Q$-values are iteratively trained through the OOD loss term, we expect $\hat{Q}\_{\theta'\_i}(s', a')$ to become increasingly accurate over time. Additional experiments presented in Appendix B.2 demonstrate that SQOG effectively approximates the true $Q$-value during training iterations, further supporting this analysis. This iterative improvement ensures alignment with the theoretical principles of SBO while accommodating practical constraints.
>
> [1] S. Fujimoto, D. Meger, and D. Precup. Off-Policy Deep Reinforcement Learning without Exploration. ICML 2018.
>
> **Response to A5:** Thank you for your insightful feedback regarding the notation and its potential to cause confusion. We have addressed this concern in our recent updates by refining the definitions and discussions to improve clarity. Specifically, we revised **Definition 2**, **Proposition 3**, and **Theorem 3** to explicitly emphasize that our focus is on OOD regions within the CHN. Additionally, we included a concluding paragraph at the end of **Section 3.2** to summarize the contributions and limitations of SBO, ensuring a more precise and coherent presentation of the discussion.

---

> ### Author Response · Authors · 2024-11-24
> **Looking forward to your feedback**
>
> Dear Reviewer cDBr,
>
> As the end of the discussion period approaches, we would greatly appreciate it if you could confirm whether our response has adequately addressed your concerns. If you have any remaining questions, please let us know, and we will do our best to respond within the remaining time.
>
> If your questions have been addressed, we kindly ask you to consider raising your rating. Thank you again for your time and efforts in reviewing our manuscript.
>
> Best regards,
>
> The authors

---

> > ### Comment · Reviewer_cDBr · 2024-11-26
> >
> > A1:
> > Thank you for adjusting the definition. I have now a good picture of what N(Conv(D)) represents, and I have no further issues with Definition 1.
> >
> > A2:
> > - Thank you for emphasizing the notion of OOD chosen early on, I had missed this addition in the document. Regarding the footnote: I believe its place would be better indicated when you make the switch from the strict OOD definition to the relaxed one — I assume at the beginning of Sec. 3.3.
> > - Thank you for your answer and for confirming that in the general continuous case, $P(\pi_\phi(s) \in D) = 0$. I merge the rest of my response with A3-4.
> > - Regarding the discussion of CHN in Section C: thank you for this addition, this is helpful.
> >
> > A3-4:
> > Thank you for the discussion. I agree that in the experiments detecting when $\hat{\mu}(a | x) \approx 0$ would be challenging. On the other hand, it should be pretty easy to check that $\hat{\mu}(a | x) = 0$, that is, whether the action is in the dataset (i.e., applying the stricter definition of OOD that SBO was defined with).
> >
> > Overall, I am still bothered by the significantly different behaviors of the two algorithms. According to our discussion and in the general case, bootstrapping happens with probability 0 with the SBO (Def. 2) — and barring edge cases, that means that SBO will (almost) never bootstrap from $Q(s’, \pi_\phi(s’))$. On the other hand, Eq. 14 bootstraps for every example. Because of this discrepancy, I am still concerned that the theoretical findings of Sec. 3 are pretty disjoint from the empirical findings of Sec. 4.
> >
> > A5:
> > Thank you for the modifications, this clarifies things and is definitely a step in the right direction.
> >
> > A6:
> > Thank you for providing these additional details.
> > I find the clipping choice to be odd, since I believe that following your code, a significant amount of mass will be put at the boundary of your clipping, especially if the clipping is close to the scale, as I mentioned above.
> > Therefore, I also do not think it is clear that the uniform noise has a higher probability of generating values closer to the clip boundaries. It certainly depends on the clip and scale parameters.
> > For instance, after a quick check, with a scale of 0.6 and a clipping of 0.5, 40% of the mass of the pdf would be concentrated at the boundaries. Therefore, a far from negligible number of samples will be concentrated at the clipping boundaries. This should also give this distribution a relatively high standard deviation: in the example above, after an empirical check, the clipped normal would actually have a higher standard deviation than the uniform distribution. I am therefore not convinced by the explanations given by the authors for the better performance of the normal distribution vs the uniform one.
> >
> >
> > An alternative would have been to pipe the norm through a scaled tanh, such as done in [1], for instance.
> > [1] Haarnoja, Tuomas, et al. "Soft actor-critic: Off-policy maximum entropy deep reinforcement learning with a stochastic actor." International conference on machine learning. PMLR, 2018.
> >
> >
> > A7:
> > Thank you for the precision regarding the type of interval used. The fact that this is a standard practice makes this way of presenting the results reasonable — it does not mean the information should not be present, at a minimum in the appendix.
> >
> >
> > I want to thank again the authors for the effort put into this rebuttal and in improving the paper.
> > I have raised my score to reflect the improvements of the manuscript.

---

> > > ### Author Response · Authors · 2024-11-28
> > > **Looking forward to your feedback**
> > >
> > > Dear Reviewer cDBr,
> > >
> > > As the deadline for the final PDF submission approaches, we would like to kindly inquire whether the additional experiments and further analysis we have provided address your concerns. If you have any remaining suggestions or questions, please feel free to share them, and we will do our utmost to incorporate them and further improve the manuscript within the remaining timeframe.
> > >
> > > Thank you once again for your valuable time and insightful feedback during the review process.
> > >
> > > Best regards,
> > >
> > > The authors

---

> ### Author Response · Authors · 2024-11-27
> **Author Response to Reviewer cDBr (part 1/2)**
>
> Thank you for your detailed feedback and for taking the time to evaluate our work. We greatly appreciate your recognition of our efforts in improving the manuscript and your re-evaluation of our contributions. After carefully reviewing your comments, we have identified two main concerns and address them as follows:
>
> **Concern 1:  The connections between SBO and the critic loss.**
>
> **A1:** Thank you for your detailed comments. To clarify, SBO consists of two components, and we structured our critic loss to align closely with these components, despite some practical limitations.
>
> **Smooth Bellman Operator**
>
> 1. **SBO for In-Sample Actions:**
> For in-sample actions, SBO first applies the base Bellman operator, $\hat{B}\_{2}^{\pi}$. If the policy’s action $\pi\_{\phi}(s’)$ is detected as out-of-distribution (OOD), the smooth generalization operator $\mathcal{G}\_1$ updates the OOD action’s $Q$-value, $Q(s’, \pi\_{\phi}(s’))$, using its in-sample neighboring action’s $Q$-value, $Q(s’, a^{in}_{neighbor})$.
>
> 2. **SBO for OOD Actions:** For Q-values of OOD actions within the CHN, SBO applies $\mathcal{G}_1$ directly to generalize them to their in-sample neighbors.
>
> **Corresponding Critic Loss**
>
> 1. **For In-Sample Actions in Critic Loss:** In practice, batches are sampled from the dataset, so the critic loss primarily deals with **in-sample actions**. The first term in the critic loss bootstraps these in-sample actions using an empirical Bellman operator, which closely resembles the first part of SBO. However, there is one difference: in practice, precisely classifying whether an action is OOD is infeasible (checking whether the action is in the dataset is easy but time-consuming, we have tried this method). Hence, we leave $\hat{Q}\_{\theta'\_i}(s', a')$ unchanged.
>
> 2. **For OOD Actions in Critic Loss:** We actively generate OOD actions based on in-sample actions and train their Q-values using the neighboring in-sample Q-values. This process **aligns well** with the second part of SBO, aiming to smooth the generalization of OOD Q-values.
>
> **The main theoretical insight is that generalizing OOD Q-values to their neighboring in-sample Q-values is beneficial.** Follow this insight, we design the OG loss. We acknowledge that we do not replace $\hat{Q}\_{\theta'\_i}(s', a')$ with the neighboring in-sample $Q$-value because identifying whether $a'$ is OOD is infeasible in practice. Nevertheless, as the OOD $Q$-values are iteratively trained through the OG loss term, we expect $\hat{Q}\_{\theta'\_i}(s', a')$ to become increasingly accurate over time, mitigating the overestimation and underestimation issue of $\hat{Q}\_{\theta'\_i}(s', a')$ as training progresses.
> Moreover, the **BC loss** applied to the actor network keeps the learned policy $\pi$ not far from the behavior policy $\mu$. This ensures that the actions $a'$ generated from $\pi$ are likely to remain close to the dataset. Consequently, $\hat{Q}\_{\theta'\_i}(s', a')$ is less prone to overestimation, further stabilizing the learning process.
>
> We have added further details in **Appendix D** for a more comprehensive explanation. We believe this alignment between SBO and the critic loss substantiates the theoretical-to-empirical connection, addressing your concern about the connection between the theory and the practice.
>
> **Further discussion:** As we have demonstrated that the BC loss helps SBO mitigate overestimation, it is important to explore the relationship between SBO and BC loss. We provide theoretical justification and empirical validation in **Appendix E**. In summary, SBO is not restricted to TD3+BC or BC loss. Our empirical results demonstrate that SBO serves as a versatile **plug-in** for policy constraint methods (e.g., BRAC), addressing the over-constraint issue and achieving superior performance and computational efficiency.

---

> ### Author Response · Authors · 2024-11-27
> **Author Response to Reviewer cDBr (part 2/2)**
>
> **Concern 2:  Explanation for the poor performance of uniform noise.**
>
> **A2:** To better analyze the performance differences among noise types, we conducted additional experiments using three different noise settings: normal noise with a scale of 0.6 and clip [-0.5, 0.5], normal noise scaled through a tanh transformation to [-0.5, 0.5], and uniform noise within [-0.5, 0.5]. Results across 4 random seeds are presented in Table 1.
>
> Table 1: Normalized average score of SQOG with different noise types on MuJoCo "-v2" and Adroit "v1" datasets. The results are averaged over 4 different random seeds. We bold the highest scores.
>
> | Dataset           | normal+0.6 scale| normal+tanh | uniform |
> |--------------------|-------|-------|-------|
> | halfcheetah-medium   | 59.2$\pm$2.4  |  **59.3$\pm$0.4** | 47.4$\pm$11.1 |
> | hopper-medium    | **100.6$\pm$0.7** |  89.7$\pm$7.4 | 62.5$\pm$36.5 |
> | walker2d-medium   | **82.9$\pm$0.8**  |   82.8$\pm$3.2 | 65.5$\pm$17.9 |
> | halfcheetah-medium-replay    | **46.4$\pm$1.2** |  39.9$\pm$1.6 |  40.1$\pm$1.2  |
> | hopper-medium-replay     | **100.9$\pm$5.1**| 94.5$\pm$5.9 | 37.9$\pm$38.3  |
> | walker2d-medium-replay   |  **88.3$\pm$3.5**  | 62.5$\pm$17.4 | 46.8$\pm$20.4 |
> | Mujoco Average | **79.7** | 71.5 | 49.9 |
> |pen-human| **80.0$\pm$4.7**  | 75.3$\pm$5.8 | 75.4$\pm$4.1 |
> |pen-cloned|  **66.7$\pm$3.4** | 64.5$\pm$8.1 | 61.9$\pm$4.7 |
> |Adroit Average |    **73.4**  |     69.9 |     68.7 |
>
> **Key Observations:**
>
> 1. Uniform noise underperforms significantly on MuJoCo datasets, with higher variance compared to normal noise settings. This is likely due to uniform noise generating overly random OOD samples that are **distributed sparsely and equally** across the entire range of [-0.5, 0.5]. While this ensures coverage across the OOD region, it fails to focus sufficiently on some key areas that are critical for robust Q-function learning. Consequently, uniform noise may lead to insufficient training in each region, resulting in unstable and inconsistent performance.
>
> 2. Normal noise (both scaled+clip and tanh-transformed) performs better due to its concentrated sampling behavior. For **normal+scale+clip** noise, samples are densely concentrated at the clip boundaries (e.g., -0.5, 0.5) and around the mean (0). For **normal+tanh** noise, most samples are concentrated near the boundaries (e.g., -0.5, 0.5). This boundary-focused sampling behavior may potentially improve generalization by promoting more robust learning in critical regions.
>
> 3. On Adroit datasets, differences between noise types are less pronounced, suggesting that the **clip range** [-0.5, 0.5] plays a more critical role than the specific noise type. Within this range, all noise types perform reasonably well, with some variability.
>
> Our observations suggest that the poor performance of uniform noise may result from its overly sparse and evenly spread sampling, which appears to limit its ability to provide sufficient coverage of critical regions for Q-value generalization. In contrast, normal noise with clipping or tanh transformations demonstrates potential advantages due to its **boundary-focused** sampling, which may facilitate more effective learning in key OOD regions. We appreciate the reviewer’s observation that clipped normal noise concentrates significant mass near the boundaries. Our findings are consistent with this observation, as the dense sampling near boundaries appears to support training in critical OOD regions, potentially contributing to the observed performance gains of clipped normal noise.
>
> **Further discussion:**
>
> While our experiments demonstrate the superiority of normal noise in this context, we acknowledge that **noise type and its influence on OOD action sampling is an important topic deserving deeper exploration**. Our primary contribution lies in introducing SQOG, which effectively addresses over-constraint issues in offline RL by improving OOD Q-value generalization, delivering superior performance and computational efficiency. In this work, we focused on empirically validating the feasibility of normal noise (with clip) and providing preliminary analyses of noise type and range. As future work, we plan to conduct a more systematic investigation into the role of noise in OOD sampling and Q-function generalization, aiming to establish a clearer theoretical understanding of its impact.
>
> We have added further details and analysis in **Appendix B.5**.
>
> **Conclusion:**
> We hope our clarification addresses your remaining concerns, and we would be happy to engage in further discussions if there are any additional questions. We have carefully considered the other suggestions in your feedback and have introduced the uncertainty metric in **Appendix B.8**. Thank you once again for your time and thoughtful evaluation of our work.

---

> > ### Comment · Reviewer_cDBr · 2024-11-29
> >
> > Thank you for your answer. I appreciate the efforts in trying to make the link between theory and practice clearer by writing Appendix D and E.
> > In particular, I think the discussion around the BC loss makes a good point regarding the similarity we can expect between $\hat{Q}$ and $\mu$, at least in the area where the BC loss is in effect.
> > This hints that the effect of the differences between SBO and SQOG should not be too big in the area where the BC loss is in effect, and addresses *partially* my concern.
> >
> > Unfortunately, it does not remove the main difference I pointed above, that is, that $(s’, \pi(s’))$ is almost never within the dataset, and therefore that the SBO almost never bootstraps, on the contrary to SQOG. In my understanding, this difference is directly due to the strict choice of OOD definition to define SBO. To be clear, I do not think that this is a bad definition of OOD per se, but simply that with such a strict definition, important differences are bound to be introduced when moving to the experimentations, leading to the differences between SBO and SQOG.
> > (I will also note that issues remain when (s’, a’) is far from D since no overestimation term is in effect then: SQOG will bootstrap (while SBO would not) without having the BC term in effect.).
> > Still, I want to thank the authors for expliciting the importance of the behavior similarity constraint and its link with SBO and SQOG.
> >
> >
> > ### Regarding the noise
> >
> > Thank you for this additional experiment.
> >
> > First, I am not sure which computation you used to generate the tanh + normal. Usually, tanh is used to have the noise distribution fully supported on [-1, 1] (and then appropriately scaled using a multiplicative factor), therefore needing no clipping. That means that there should be no mass at the boundary, on the contrary to what you wrote, so I am unsure what these results refer to.
> >
> > Second, while I acknowledge the empirical performance of normal + clip, I am not convinced by the reasoning you give.
> > The clipped normal distribution is very peculiar, which in my opinion would have required a stronger motivation compared to using a regular Normal (potentially with bounded support thanks to tanh, or using the truncatedNorm [https://docs.scipy.org/doc/scipy/reference/generated/scipy.stats.truncnorm.html]). I will also note that if this boundary effect was intended and important, there was as far as I can tell no mention of it in the original manuscript.
> >
> > Most importantly, you claim that the improved performance of normal + clip (over uniform) is explained by its boundary effect (since I do not know how you implemented the normal + tanh, I will ignore the comment about its boundary effect).
> > However, in your message from the 23rd of November, you used the opposite argument to justify the better performance of normal+clip (emphasis mine):
> > - “The uniform noise tends to perform worse because it has a higher probability of generating values closer to the clip boundaries. If larger noise values occur during the early stages of training, they can potentially disrupt the learning of the in-sample -function.”
> > - “This is because uniform noise has a non-negligible probability of generating values near the clip boundaries, which negatively impacts in-sample Q learning. During the early stages of training, when the in-sample Q function is still inaccurate, a more conservative noise distribution is essential, making normal noise a more suitable choice.”
> > - “This suggests that while uniform noise has potential, while its less controlled nature leads to inconsistent results compared to the more conservative normal noise.”
> >
> > I am therefore unconvinced by this justification.
> > In conclusion, I acknowledge the empirical performance of the normal + clip noise, but I remain concerned with the reasons motivating the choice of this noise and the explanation of its better performance, which remain important open questions.
> >
> > ### Conclusion
> > As a conclusion of all these exchanges, I believe my score appropriately reflects my position on this paper, and I have updated my confidence accordingly. Thanks again for your engagement in this discussion and the improvements you made since the beginning of this discussion phase.

---

> ### Author Response · Authors · 2024-11-30
> **Author Response to Reviewer cDBr**
>
> Thank you for your kind reply. We are glad that you found the discussion in Appendix D and E helpful in clarifying the link between theory and practice, particularly regarding the connection between SBO and SQOG. We will also include a more detailed discussion on this aspect in the camera-ready version, particularly regarding the strict and relaxed OOD definition.
>
> **Main Concern: Better explanation on the additional experiment of noise type.**
>
> **A:**
> First, The tanh + normal noise was generated using `jnp.tanh(jax.random.normal(key, shape)) * 0.5`, ensuring the resulting noise is fully supported on [-0.5, 0.5] without explicit clipping. However, due to the shape of the tanh function, values near the boundaries tend to occur with higher probability compared to values closer to zero, as the tanh function asymptotically compresses larger magnitude samples towards its bounds. For example, if $z \sim N(0,1)$, then $P\_{z}([-1,1])\approx0.68$. After applying the transformation $y=tanh(z)*0.5$, $P\_{y}([-0.3808,0.3808])\approx0.68$ and $P\_{y}([-0.5,-0.3808]\cup[0.3808,0.5])\approx0.32$, illustrating the boundary-focused density induced by the tanh transformation. (We will include visualizations of the different noise types in the camera-ready version to clarify these effects). We added experiments on normal + tanh noise to better understand the effects of boundary-focused sampling, as we initially interpreted this as a suggestion from the reviewer.
>
> Second, we apologize for the inconsistency in our earlier explanations regarding the boundary effect. In our message from the 23rd of November, we mistakenly attributed the poor performance of uniform noise solely to its high probability of generating values near the clip boundaries. Upon further reflection and additional experiments, we now better understand the cause: uniform noise generates overly random OOD samples that are sparsely and equally distributed across the entire range of [-0.5, 0.5]. In contrast, normal noise with clipping and tanh + normal noise perform better because their focused sampling, particularly near boundaries. This boundary-focused sampling generates sufficient samples in key OOD regions, which is critical for training the OOD Q-function.
>
> To be honest, we acknowledge that the choice of noise range and type is an **empirical decision** aimed at generating OOD actions for the OG loss (Eq. 13) with low computational cost. Initially, we did not focus heavily on the choice of noise, as we observed strong performance with normal noise + [scale=0.6, clip=0.5]. However, we are grateful to the reviewers for bringing this to our attention during the rebuttal period. In response, we conducted over 200 additional experiments to study the impact of different noise ranges and types, with further results provided in Appendix B.4 and B.5. These experiments help clarify the role of noise in our method.
>
> Nevertheless, we recognize that the effects of noise types in OOD sampling are inherently complex and warrant deeper exploration in future research. We believe that the topic of OOD sampling merits a dedicated study rather than being confined to a single section of this paper. In this work, we have two empirical insights:
>
> 1. During the early stages of training, noise distributions should allocate some probability density near zero to prevent disruption of in-sample Q learning.
>
> 2. The focused sampling is important. Generating sufficient samples in key OOD regions is essential for training an accurate OOD Q-function. However, in high-dimensional datasets, it is infeasible to explicitly identify these key OOD regions, which limits our ability to provide a more detailed explanation of this point.
>
> From the results in Appendix B.5, we observed that focus sampling is more critical. However, we want to stress that this empirical observation is not the core contribution of this work.
>
> **Our primary contribution lies in addressing the over-constraint problem in offline RL through SQOG, which improves OOD Q-value generalization.** In the rebuttal, we empirically demonstrate that SBO serves as an effective and versatile plug-in for policy constraint methods (e.g., BRAC), resolving over-constraint issues and delivering superior performance and computational efficiency. **This contribution has been recognized by reviewers (Fqh7, etFc, and CHBA), who considered SQOG a noteworthy advance in the Offline RL community and awarded it positive scores.**
>
> We hope that the explanations provided address your concerns, and we truly appreciate your thoughtful feedback, which has greatly contributed to improving the clarity of this work. Additionally, we hope that you would reconsider the score to better reflect the improvements made and our contribution to the Offline RL community, which means a lot to us and will help promote our future work on OOD sampling with noise. Thank you again for your engagement in this discussion, and we look forward to your continued support.

---

> ### Author Response · Authors · 2024-11-30
> **Further Discussion on Our Contribution to the Offline RL Community**
>
> As the end of the discussion period approaches, we would like to take this opportunity to further discuss the contribution of our work.
>
> We greatly appreciate your recognition of the importance and motivation of this problem, as reflected in your initial comment: **"I find that the problem tackled is important and very well motivated."** However, we would like to provide additional background to further clarify our approach.
>
> In the context of the Offline RL community, the over-constraint issue remains **a significant challenge** for policy constraint methods. However, theoretical analysis and experiments often neglect the "value addition" in critic network, as it risks **conflicting** with policy constraints in actor network. To address this, we propose the Smooth Bellman Operator (SBO), which acts as a valuable addition to policy constraint methods. On one hand, SBO overcomes this bottleneck by accurately learning OOD Q-values for policy improvement and exploration (proved in Theorem 3, shown in the sanity check and Appendix B.2). On the other hand, SBO is really smooth when integrating into those policy constraint methods, primarily because we treat policy constraints as a precondition (as demonstrated by the significant improvements in Tables 1, 2, and 10). We firmly believe that SBO (SQOG) is a valuable supplement, combining **theoretical rigor**, **strong performance**, and **computational efficiency**.
>
> Finally, as highlighted in the Introduction, previous value-based offline RL methods often view OOD regions as inherently risky due to information deficiencies (e.g., [1], [2]). However, our work demonstrates the insight that **extending $Q$-function generalization to OOD regions within CHN can be beneficial**. We believe this approach will attract more attention to $Q$-value estimation in OOD regions and offer new insights for the Offline RL community.
>
> We sincerely hope that you would consider re-evaluating our contributions. A re-evaluation would mean a great deal to us and would motivate us to continue advancing this line of research. Thank you again for your time and thoughtful feedback.
>
> [1] Haoran Xu et al., Offline RL with no OOD actions: In-sample learning via implicit value regularization. ICLR 2023.
>
> [2] Aviral Kumar et al., Conservative q-learning for offline reinforcement learning. NeurIPS 2020.

---

### Official Review · Reviewer_Fqh7 · 2024-11-04

**Soundness:** 3
**Presentation:** 3
**Contribution:** 3
**Rating:** 8
**Confidence:** 4

**Summary:**

This paper introduces the method of using the convex hull and a smooth Bellman operator to solve the challenge of over-conservative behaviors in offline reinforcement learning algorithms.
Both theoretical analysis and empirical justifications of the proposed method are provided.
The results look promising on the reported tasks.

**Strengths:**

The paper is well written and the structure is clear. The idea is supported by both theoretical justification and relatively sufficient empirical study.


----

The authors made a great effort in the rebuttal process and addressed all my concerns.

**Weaknesses:**

Using convex hulls and nearest neighbors in reinforcement learning is not a new technique, yet the current paper does not discuss those methods. Giving credit to the existing work will not hurt the novelty of this paper but will help readers have a clearer understanding of the advancements in the field.

[1] Sun, Hao, et al. "Accountability in offline reinforcement learning: Explaining decisions with a corpus of examples." Advances in Neural Information Processing Systems 37 (2023).

[2] Lyu, Jiafei, et al. "SEABO: A Simple Search-Based Method for Offline Imitation Learning." arXiv preprint arXiv:2402.03807 (2024).


Figure 3 (the table) is not very clear. The authors may want to re-organize the information in a different format.

**Questions:**

Limitations and potential pitfalls of the proposed method are not discussed. Could the authors explain more about the potential challenges of the proposed method? e.g., how do the choices of distance metric affect the results? what would be the trade-off between generalization and conservative behaviors? Is there any task that would benefit more from a conservative estimation or more aggressive extrapolation? Having analyses of some case studies would greatly improve the clarity and impact of the paper.

How does the proposed method scale with different numbers of training samples (i.e., such that convex hulls differ)? Could the authors provide some analysis of the computational cost (time/memory usage), and compare different methods in those settings?

---

> ### Author Response · Authors · 2024-11-21
> **Author Response to Reviewer Fqh7 (part 1/3)**
>
> Thank you for your inspiring and thoughtful comments, and for noting that our paper is "well written". We also appreciate your detailed feedback. We hope our responses below address your concerns effectively.
>
> **Q1: How do the choices of distance metric affect the results?**
>
> **A1:** We greatly appreciate the reviewer’s insightful question. In practice, the choice of **distance metrics** corresponds to the **noise scale and clipping parameters**. To evaluate their influence on performance across multiple datasets, we conducted additional experiments by systematically varying these parameters. The results are presented below.
>
> Table 1: Normalized average score of SQOG over different choices of gaussian noice scale and clip on MuJoCo "-v2" and Adroit "v1" datasets. The results are averaged over 4 different random seeds.
>
> | Dataset           | scale=0, clip=0| scale=0.2, clip=0.3  |scale=0.6, clip=0.5  |scale=1.0, clip=0.7  | scale=2.0, clip=1.0|
> |--------------------|-------|-------|-------|-------|-------|
> | halfcheetah-medium   | 51.2$\pm$3.9  |  **60.6$\pm$0.5** | 59.2$\pm$2.4  |  53.1$\pm$1.9 | 51.7$\pm$1.5|
> | hopper-medium    | 1.5$\pm$0.3  |  67.2$\pm$1.6 | **100.6$\pm$0.7**  |  94.5$\pm$3.6 | 79.5$\pm$10.3|
> | walker2d-medium   | 48.3$\pm$1.2   |   58.9$\pm$2.3 | 82.9$\pm$0.8  |  **86.0$\pm$1.0** | 82.5$\pm$1.1|
> | halfcheetah-medium-replay    | **49.2$\pm$0.3**  |  47.8$\pm$2.2 |  46.4$\pm$1.2  |  36.6$\pm$1.1| 35.8$\pm$ 0.8|
> | hopper-medium-replay     | 22.3$\pm$6.6 | 19.4$\pm$6.8 | **100.9$\pm$5.1**  |  24.3$\pm$2.7 | 27.1$\pm$6.7 |
> | walker2d-medium-replay   | 12.2$\pm$5.5  | 56.8$\pm$2.0 | 88.3$\pm$3.5  |  **90.1$\pm$3.6** | 60.6$\pm$6.9 |
> | Mujoco Average |  30.8 | 51.8 | **79.7** | 64.1 | 56.2 |
> |pen-human| 23.3$\pm$6.7  | 46.7$\pm$4.8 | **80.0$\pm$4.7** |  64.9$\pm$5.5 | 55.8$\pm$5.9|
> |pen-cloned|  9.5$\pm$1.9  | 41.2$\pm$7.6 | **66.7$\pm$3.4** |  43.2$\pm$5.1 | 42.3$\pm$7.0|
> |Adroit Average |    16.4 |     44.0 |     **73.4** |    54.1 |  49.1 |
>
> From these results, we observe the following:
> 1. **Moderate Noise Balances Exploration and Stability:** Across most datasets, a scale of 0.6 and clip of 0.5 consistently achieves strong performance, suggesting that moderate noise levels effectively balance exploration and stability. While noise promotes generalization, excessive noise can lead to sampling outside the CHN boundary, resulting in OOD Q-values that violate **safety guarantees** and degrade performance. Properly scaled noise ensures effective learning while respecting safety constraints.
>
> 2. **Dataset-Specific Effects of Noise:**
> The sensitivity to the noise distribution parameters (scale and clip) varies across datasets.
> In halfcheetah-medium-replay, the baseline configuration (scale=0, clip=0) yields the best performance, indicating that the effect of noise on in-sample Q-value estimation cannot be ignored. The dataset may already provides sufficient diversity, and adding noise introduces harmful uncertainty, leading to performance degradation. For such datasets, it’s crucial to keep noise parameters conservative to avoid disrupting in-sample Q-learning and maintain stable performance.
> In contrast, for halfcheetah-medium, the optimal configuration is (scale=0.2, clip=0.3), and for walker2d-medium and walker2d-medium-replay, (scale=1.0, clip=0.7) achieves the best results. These differences highlight that there is room for improvement in fine-tuning noise parameters across various datasets.
>
> 3. **Effectiveness of Noise Injection:** The baseline configuration (scale=0, clip=0) significantly underperforms in most datasets, underscoring the necessity of noise injection to enhance exploration and overall performance. Noise injection is essential for improving the Q-function's ability to generalize to previously unexplored regions and optimize learning.

---

> ### Author Response · Authors · 2024-11-21
> **Author Response to Reviewer Fqh7 (part 2/3)**
>
> **Q2: What would be the trade-off between generalization and conservative behaviors?**
>
> **A2:**  We sincerely thank the reviewer for raising this valuable question about the trade-off between generalization and conservative behaviors. In our approach, this balance is primarily influenced by two hyperparameters:
>
> - **$\beta$**: Controls the significance of the out-of-distribution (OOD) generalization term in the critic loss, as demonstrated in our **ablation study**.
> - **$\alpha$**: Governs the relative intensity of the behavioral cloning penalty in the actor loss. Smaller values of $\alpha$ encourage more conservative behaviors.
>
> To explore the effects of $\alpha$, we conducted additional experiments by systematically varying $\alpha$ while keeping $\beta=0.5$. The results are summarized in the table below:
>
> Table 2: Normalized average score of SQOG over different choices of $\alpha$ on MuJoCo "-v2" datasets. The results are averaged over 4 different random seeds.
>
> | Dataset ($\beta=0.5$)    | $\alpha=300$         | $\alpha=200$        | $\alpha=150$         | $\alpha=100$         | $\alpha=50$          |
> |---------------------------|----------------------|---------------------|----------------------|----------------------|----------------------|
> | halfcheetah-medium        | **59.3$\pm$1.2**    | 57.7$\pm$1.6        | 59.2$\pm$2.4         | 54.1$\pm$1.0         | 51.1$\pm$0.4         |
> | hopper-medium             | 70.9$\pm$7.7        | 91.6$\pm$6.0        | **100.6$\pm$0.7**    | 94.1$\pm$6.8         | 74.9$\pm$4.1         |
> | walker2d-medium           | 83.6$\pm$7.5        | **88.2$\pm$1.8**    | 82.9$\pm$0.8         | 81.7$\pm$0.8         | 81.1$\pm$0.3         |
> | hopper-medium-replay      | 74.1$\pm$5.5        | 86.9$\pm$8.5        | **100.9$\pm$5.1**    | 84.0$\pm$9.4         | 82.8$\pm$9.9         |
>
> From the results, we observe the following:
>
> 1. **Generalization benefits with larger $\alpha$:**
>    On datasets like halfcheetah-medium and walker2d-medium, larger $\alpha$ values (e.g., $\alpha=300$ or $\alpha=200$) tend to perform better, suggesting that these tasks benefit from more aggressive generalization. The dynamics of these environments may not require highly conservative strategies, allowing the agent to explore beyond the dataset effectively.
>
> 2. **A balanced approach:**
>    Across tasks, the results indicate that a moderate value of $\alpha=150$ provides a strong balance between generalization and conservative behaviors. This balance likely ensures sufficient exploration without risking the instability of over-generalization, making it a robust choice across diverse scenarios.
>
> Combined with the results of experiments on noise scale and clipping in A1, we find that larger noise clipping and larger $\alpha$ achieve better performance in the *walker2d-medium* dataset. This implies that the task **benefits more from $Q$-value generalization and aggressive policy extrapolation within the CHN**. These findings further demonstrate the importance of tailoring the level of conservativeness to the specific characteristics of each task.

---

> ### Author Response · Authors · 2024-11-21
> **Author Response to Reviewer Fqh7 (part 3/3)**
>
> **Q3: How does the method scale with varying training sample sizes, and what are its computational costs compared to other methods in those settings?**
>
> **A3:**  We appreciate the reviewer’s insightful question regarding scalability and computational efficiency. To evaluate the performance of our method across different sample sizes, we conducted experiments on datasets with varying numbers of training samples. Below, we summarize the results:
>
> **Performance Results:**
> Table 3 shows that SQOG consistently outperforms the baseline method MCQ across datasets with varying sample sizes, achieving higher normalized average scores. This indicates that the number of samples is not a primary factor influencing SQOG’s performance. As discussed in the paper and previous answers A1 and A2, the key factors affecting performance are the noise distribution, noise scale and clipping, and the choice of hyperparameters that balance conservativeness and generalization.
>
> Table 3: Normalized average score comparison of SQOG against baseline method MCQ on datasets of different numbers of training samples over the final 10 evaluations and 4 random seeds.
>
> | Dataset (Sample Size)      | SQOG (ours)      | MCQ               |
> |-----------------------------|------------------|--------------------|
> | pen-human (5,000 samples)   | **80.0±4.7**     | 68.5±6.5           |
> | pen-cloned (500,000 samples)| **66.7±3.4**     | 49.4±4.3           |
> | hopper-medium (1,000,000 samples)| **100.6±0.7**| 73.6±10.3          |
> | maze2d-medium (2,000,000 samples)| **149.4±2.9**| 106.8±38.4         |
>
> **Computational Efficiency:**
> Table 4 demonstrates that SQOG maintains a consistent and low runtime across datasets of different sizes. Compared to MCQ, SQOG achieves significantly faster runtime due to its efficient OOD sampling strategy (sampling by adding noise to in-sample actions). This efficiency is largely independent of the number of samples or the shape of the convex hull.
>
> Table 4: Run time comparison of SQOG against baseline method MCQ on datasets of different numbers of training samples.
>
> | Dataset (Sample Size)      | SQOG Run Time (h) (ours) | MCQ Run Time (h) |
> |-----------------------------|--------------------------|-------------------|
> | pen-human (5,000 samples)   | **0.4**                 | 7.5               |
> | pen-cloned (500,000 samples)| **0.4**                 | 7.5               |
> | hopper-medium (1,000,000 samples)| **0.5**            | 8.2               |
> | maze2d-medium (2,000,000 samples)| **0.5**            | 8.0               |
>
> **Analysis:**
> Our results demonstrate that the number of samples in the dataset has minimal impact on SQOG’s effectiveness. This is because SQOG avoids direct convex hull computations by efficiently adding noise to in-sample data, ensuring robust OOD sampling without incurring additional computational complexity.
>
> SQOG’s runtime is nearly invariant to dataset size, showcasing its scalability. Unlike MCQ, SQOG avoids the use of any generative model, achieving SOTA results with low computational cost. Furthermore, in offline RL, memory usage is generally not a critical concern due to the static nature of the datasets. SQOG, similar to most baselines (e.g., TD3+BC, IQL), enjoys relatively low memory usage, making it highly practical in terms of resource requirements.
>
> The critical determinants of SQOG’s performance are the characteristics of the noise distribution (e.g., scale and clipping) and the hyperparameters to balance conservativeness and generalization. This allows SQOG to perform well across a variety of tasks and dataset sizes, making it a robust and efficient solution.
>
> **Q4: Comments on the "Weaknesses".**
>
> **A4:** We thank the reviewer for highlighting the importance of acknowledging prior work in the area of convex hulls and nearest neighbor-based methods in reinforcement learning. After reviewing the works by Sun et al. [1] and Lyu et al. [2], we recognize that the nearest neighbor techniques presented in these papers are conceptually similar to the "neighborhood" used in our method. **We have updated the manuscript to explicitly cite these prior works**, clarifying the distinctions between them and our method.
>
> In response to the reviewer’s comment on the clarity of Figure 3, **we have reformatted the table** for improved readability and enhanced its caption to ensure the data is presented more clearly.
>
> We hope these revisions will address the reviewer’s concerns and enhance the clarity and completeness of our paper. Thank you for your valuable input.
>
> [1] Sun, Hao, et al. "Accountability in offline reinforcement learning: Explaining decisions with a corpus of examples." Advances in Neural Information Processing Systems 37 (2023).
>
> [2] Lyu, Jiafei, et al. "SEABO: A Simple Search-Based Method for Offline Imitation Learning." arXiv preprint arXiv:2402.03807 (2024).

---

> ### Author Response · Authors · 2024-11-24
> **Looking forward to your feedback**
>
> Dear reviewer Fqh7,
>
> Thank you very much for taking the time to review our work and for providing valuable feedback. We were wondering if our responses have resolved your concerns. We will be happy to have further discussions with the reviewer if there are still some remaining questions! More discussions and suggestions on further improving the paper are also always welcomed! We look forward to hearing from you and remain available to provide any additional details that might assist in resolving your concerns.
>
> Best regards,
>
> The authors

---

> > ### Comment · Reviewer_Fqh7 · 2024-11-25
> > **Thank you for the response!**
> >
> > I appreciate the authors' detailed point-by-point response and their efforts in improving the paper. My previous concerns are well addressed. I have increased my rating.

---

> > > ### Author Response · Authors · 2024-11-25
> > > **Thanks for raising the score!**
> > >
> > > Thank you for your positive feedback and for taking the time to re-evaluate our work. We greatly appreciate your recognition of our efforts to address your concerns and improve the paper. Your constructive comments were invaluable in guiding us to refine our work.

---

### Author Response · Authors · 2024-11-28
**Summary: Improvements in the Final Submission**

We sincerely thank all the reviewers for their valuable feedback and thoughtful suggestions! We are pleased that most of the concerns have been adequately addressed. Below, we summarize the key improvements made in our final submission:

- We refined the soundness of **Definition 1** (CHN) and **Definition 2** (SBO), and included an additional discussion on CHN in **Appendix C**. (addressed to reviewer cDBr)

- We expanded the discussion on the connection between theory and practice in **Appendix D**. This addition bridges the tiny gap between theoretical insights and practical implementations more effectively. (addressed to reviewers cDBr and CHBA)

- We conducted additional experiments on the noise range (scale and clip) in **Appendix B.4** and noise type in **Appendix B.5**. These experiments improve the completeness of our study. (addressed to reviewers Fqh7, cDBr, and CHBA)

- We further explored the relationship between SBO and behavior cloning (BC) loss in **Appendix E**. By incorporating experiments on BRAC+SBO, we empirically validated SBO’s effectiveness and its potential to generalize to other policy constraint methods. This reinforces our contribution to addressing the over-constraint issue and highlights the generalizability of SBO. (addressed to reviewers etFc and CHBA)

- We provided wider evidence of SQOG’s $Q$-value estimation accuracy on high-dimensional tasks in **Appendix B.2**. This complements the sanity check in Section 4 and strengthens our claims regarding SQOG’s robust performance. (addressed to reviewers etFc and CHBA)

- We performed an ablation study on hyperparameter $\alpha$ in **Appendix B.3**, allowing us to better explore the trade-off between generalization and conservatism. (addressed to reviewer Fqh7)

- We conducted additional evaluations using more random seeds in **Appendix B.6**, demonstrating SQOG’s consistent and reliable performance across varied experimental settings. (addressed to reviewers cDBr, etFc, and CHBA)

- We provided detailed descriptions of the sanity check experiments in **Appendix B.7**, enhancing the clarity and credibility of our findings. (addressed to reviewers cDBr and CHBA)

- We corrected typographical errors and revised the manuscript to improve clarity and readability throughout. (addressed to all reviewers)

We believe these enhancements substantially strengthen our contributions to the offline RL community and demonstrate the rigor and robustness of our methodology. We sincerely appreciate the reviewers’ efforts in helping us improve the quality of our work.

In particular, we extend our special thanks to reviewers **Fqh7**, **etFc**, and **CHBA** for their strong endorsement of our contributions. Additionally, we are deeply grateful to reviewer **cDBr** for providing detailed questions and constructive suggestions, which greatly enhanced both the soundness and presentation of our work.

---

### Meta-Review · Area_Chair_aDwE · 2024-12-23

**Metareview:**

This paper proposes new method to alleviate the OOD challenges of existing offline RL algorithms. Theoretical results are provides to show that Q-function generalization in OOD regions within Convex Hull and its Neighborhood (CHN) is guaranteed.  Experiments on D4RL benchmark datasets shows the advantage of the proposed methods. The problem is well-motivated. Both theoretical and empirical results are solid and well-presented. Most concerns are addressed after multiple iterations of discussions with authors. We thus recommend acceptance, and suggest authors to incorporate the feedback from the reviewers to the final version of the paper.

**Additional Comments On Reviewer Discussion:**

Authors have spent significant amount of efforts during author feedback phase. Most concerns are addressed and most reviewers are positive about the paper, with only minor issues left concerned by Reviewer cDBr.

---

### Decision · Program_Chairs · 2025-01-22

Accept (Poster)